# Efficient Reinforcement Learning in Factored MDPs with Application to Constrained RL

**Xiaoyu Chen**    **Jiachen Hu**
Key Laboratory of Machine Perception, MOE,
School of EECS, Peking University
{cxy30, NickH}@pku.edu.cn

**Lihong Li**
Amazon
llh@amazon.com

**Liwei Wang**
Key Laboratory of Machine Perception, MOE,
School of EECS, Peking University
Center for Data Science, Peking University
wanglw@cis.pku.edu.cn

## Abstract

Reinforcement learning (RL) in episodic, factored Markov decision processes (FMDPs) is studied. We propose an algorithm called FMDP-BF, whose regret is exponentially smaller than that of optimal algorithms designed for non-factored MDPs, and improves on the previous FMDP result of Osband & Van Roy (2014b) by a factor of $\sqrt{nH|\mathcal{S}_i|}$, where $|\mathcal{S}_i|$ is the cardinality of the factored state subspace, $H$ is the planning horizon and $n$ is the number of factored transitions. We also provide a lower bound, which shows near-optimality of our algorithm w.r.t. timestep $T$, horizon $H$ and factored state-action subspace cardinality. Finally, as an application, we study a new formulation of constrained RL, RL with knapsack constraints (RLwK), and provide the first sample-efficient algorithm based on FMDP-BF.

## 1 Introduction

Reinforcement learning (RL) is concerned with sequential decision making problems where an agent interacts with a stochastic environment and aims to maximize its cumulative rewards. The environment is usually modeled as a Markov Decision Process (MDP) whose transition kernel and reward function are unknown to the agent. A main challenge of the agent is efficient exploration in the MDP, so as to minimize its regret, or the related sample complexity of exploration.

Extensive study has been done on the *tabular* case, in which almost no prior knowledge is assumed on the MDP dynamics. The regret or sample complexity bounds typically depend polynomially on the cardinality of state and action spaces (e.g., Strehl et al., 2009; Jaksch et al., 2010; Azar et al., 2017; Dann et al., 2017; Jin et al., 2018; Dong et al., 2019; Zanette & Brunskill, 2019). Moreover, matching lower bounds (e.g., Jaksch et al., 2010) imply that these results cannot be improved without additional assumptions. On the other hand, many RL tasks involve large state and action spaces, for which these regret bounds are still excessively large.

In many practical scenarios, one can often take advantage of specific structures of the MDP to develop more efficient algorithms. For example, in robotics, the state may be high-dimensional, but the subspaces of the state may evolve independently of others, and only depend on a low-dimensional subspace of the previous state. Formally, these problems can be described as factored MDPs (Boutilier et al., 2000; Kearns & Koller, 1999; Guestrin et al., 2003). Most relevant to the present work is Osband & Van Roy (2014b), who proposed a posterior sampling algorithm and a

UCRL-like algorithm that both enjoy $\sqrt{T}$ regret, where $T$ is the maximum timestep. Their regret bounds have a linear dependence on the time horizon and each factored state subspace. It is unclear whether this bound is tight or not.

In this work, we tackle this problem by proposing algorithms with improved regret bounds, and developing corresponding lower bounds for episodic FMDPs. We propose a sample- and computation-efficient algorithm called FMDP-BF based on the principle of *optimism in the face of uncertainty*, and prove its regret bounds. We also provide a lower bound, which implies that our algorithm is near-optimal with respect to the timestep $T$, the planning horizon $H$ and factored state-action subspace cardinality $|\mathcal{X}[Z_i]|$.

As an application, we study a novel formulation of constrained RL, known as *RL with knapsack constraints (RLwK)*, which we believe is natural to capture many scenarios in real-life applications. We apply FMDP-BF to this setting, to obtain a statistically efficient algorithm with a regret bound that is near-optimal in terms of $T$, $S$, $A$, and $H$.

Our contributions are summarized as follows:

1. We propose an algorithm for FMDP, and prove its regret bound that improves on the previous result of Osband & Van Roy (2014b) by a factor of $\sqrt{nH|\mathcal{S}_i|}$.
2. We prove a regret lower bound for FMDP, which implies that our regret bound is near-optimal in terms of timestep $T$, horizon $H$ and factored state-action subspace cardinality.
3. We apply FMDP-BF in RLwK, a novel constrained RL setting with knapsack constraints, and prove a regret bound that is near-optimal in terms of $T$, $S$, $A$ and $H$.

## 2 PRELIMINARIES

We consider the setting of a tabular episodic Markov decision process (MDP), $(\mathcal{S}, \mathcal{A}, H, \mathbb{P}, R)$, where $\mathcal{S}$ is the set of states, $\mathcal{A}$ is the action set, $H$ is the number of steps in each episode. $\mathbb{P}$ is the transition probability matrix so that $\mathbb{P}(\cdot|s, a)$ gives the distribution over states if action $a$ is taken on state $s$, and $R(s, a)$ is the reward distribution of taking action $a$ on state $s$ with support $[0, 1]$. We use $\bar{R}(s, a)$ to denote the expectation $\mathbb{E}[R(s, a)]$.

In each episode, the agent starts from an initial state $s_1$ that may be arbitrarily selected. At each step $h \in [H]$, the agent observes the current state $s_h \in \mathcal{S}$, takes action $a_h \in \mathcal{A}$, receives a reward $r_h$ sampled from $R(s_h, a_h)$, and transits to state $s_{h+1}$ with probability $\mathbb{P}(s_{h+1}|s_h, a_h)$. The episode ends when $s_{H+1}$ is reached.

A policy $\pi$ is a collection of $H$ policy functions $\{\pi_h : \mathcal{S} \to \mathcal{A}\}_{h \in [H]}$. We use $V_h^\pi : \mathcal{S} \to \mathbb{R}$ to denote the value function at step $h$ under policy $\pi$, which gives the expected sum of remaining rewards received under policy $\pi$ starting from $s_h = s$, i.e. $V_h^\pi(s) = \mathbb{E}\left[\sum_{h'=h}^H R(s_{h'}, \pi_{h'}(s_{h'})) \mid s_h = s\right]$. Accordingly, we define $Q_h^\pi(s, a)$ as the expected Q-value function at step $h$: $Q_h^\pi(s, a) = \mathbb{E}\left[R(s_h, a_h) + \sum_{h'=h+1}^H R(s_{h'}, \pi_{h'}(s_{h'})) \mid s_h = s, a_h = a\right]$. We use $V_h^*$ and $Q_h^*$ to denote the optimal value and Q-functions under optimal policy $\pi^*$ at step $h$.

The agent interacts with the environment for $K$ episodes with policy $\pi_k = \{\pi_{k,h} : \mathcal{S} \to \mathcal{A}\}_{h \in [H]}$ determined before the $k$-th episode begins. The agent's goal is to maximize its cumulative rewards $\sum_{k=1}^K \sum_{h=1}^H r_{k,h}$ over $T = KH$ steps, or equivalently, to minimize the following expected regret:

$$\text{Reg}(K) \stackrel{\text{def}}{=} \sum_{k=1}^K \left[V_1^*(s_{k,1}) - V_1^{\pi_k}(s_{k,1})\right],$$

where $s_{k,1}$ is the initial state of episode $k$.

### 2.1 FACTORED MDPs

A factored MDP is an MDP whose rewards and transitions exhibit certain conditional independence structures. We start with the formal definition of factored MDP (Boutilier et al., 2000; Osband & Van Roy, 2014b; Xu & Tewari, 2020; Lu & Van Roy, 2019). Let $\mathcal{P}(\mathcal{X}, \mathcal{Y})$ denote the set of functions that map $x \in \mathcal{X}$ to the probability distribution on $\mathcal{Y}$.

**Definition 1.** *(Factored set) Let $\mathcal{X} = \mathcal{X}_1 \times \cdots \times \mathcal{X}_d$ be a factored set. For any subset of indices $Z \subseteq \{1, 2, \ldots, d\}$, we define the scope set $\mathcal{X}[Z] := \otimes_{i \in Z} \mathcal{X}_i$. Further, for any $x \in \mathcal{X}$, define the scope variable $x[Z] \in \mathcal{X}[Z]$ to be the value of the variables $x_i \in \mathcal{X}_i$ with indices $i \in Z$. If $Z$ is a singleton, we will write $x[i]$ for $x[\{i\}]$.*

**Definition 2.** *(Factored reward) The reward function class $\mathcal{R} \subset \mathcal{P}(\mathcal{X}, \mathbb{R})$ is factored over $\mathcal{S} \times \mathcal{A} = \mathcal{X} = \mathcal{X}_1 \times \cdots \times \mathcal{X}_d$ with scopes $Z_1, \cdots, Z_m$ if for all $R \in \mathcal{R}, x \in \mathcal{X}$, there exist functions $\{R_i \in \mathcal{P}(\mathcal{X}[Z_i], [0, 1])\}_{i=1}^m$ such that $r \sim R(x)$ is equal to $\frac{1}{m} \sum_{i=1}^m r_i$ with each $r_i \sim R_i(x[Z_i])$ individually observed. We use $\bar{R}_i$ to denote the expectation $\mathbb{E}[R_i]$.*

**Definition 3.** *(Factored transition) The transition function class $\mathcal{P} \subset \mathcal{P}(\mathcal{X}, \mathcal{S})$ is factored over $\mathcal{S} \times \mathcal{A} = \mathcal{X} = \mathcal{X}_1 \times \cdots \times \mathcal{X}_d$ and $\mathcal{S} = \mathcal{S}_1 \times \cdots \times \mathcal{S}_n$ with scopes $Z_1, \cdots, Z_n$ if and only if, for all $\mathbb{P} \in \mathcal{P}, x \in \mathcal{X}, s \in \mathcal{S}$, there exist functions $\{\mathbb{P}_j \in \mathcal{P}(\mathcal{X}[Z_j], \mathcal{S}_j)\}_{j=1}^n$ such that $\mathbb{P}(s \mid x) = \prod_{j=1}^n \mathbb{P}_j(s[j] \mid x[Z_j])$.*

A factored MDP is an MDP with factored rewards and transitions. A factored MDP is fully characterized by $M = \left( \{\mathcal{X}_i\}_{i=1}^d ; \{Z_i^R\}_{i=1}^m ; \{R_i\}_{i=1}^m ; \{\mathcal{S}_j\}_{j=1}^n ; \{Z_j^P\}_{j=1}^n ; \{\mathbb{P}_j\}_{j=1}^n ; H \right)$, where $\mathcal{X} = \mathcal{S} \times \mathcal{A}$, $\{Z_i^R\}_{i=1}^m$ and $\{Z_j^P\}_{j=1}^n$ are the scopes for the reward and transition functions, which we assume to be known to the agent.

An excellent example of factored MDP is given by Osband & Van Roy (2014), about a large production line with $d$ machines in sequence with $S_i$ possible states for machine $i$. Over a single time-step each machine can only be influenced by its direct neighbors. For this problem, the scopes $Z_i^R$ and $Z_i^P$ of machine $i \in \{2, ..., d-1\}$ can be defined as $\{i-1, i, i+1\}$, and the scopes of machine 1 and machine $d$ are $\{1, 2\}$ and $\{d-1, d\}$ respectively. Another possible example to explain factored MDP is about robotics. For a robot, the transition dynamics of its different parts (e.g. its legs and arms) may be relatively independent. In that case, the factored transition can be defined for each part separately.

For notation simplicity, we use $\mathcal{X}[i : j]$ and $\mathcal{S}[i : j]$ to denote $\mathcal{X}[\cup_{k=i,\ldots,j} Z_k]$ and $\otimes_{k=i}^j \mathcal{S}_k$ respectively. Similarly, We use $\mathbb{P}_{[i:j]}(s'[i : j] \mid s, a)$ to denote $\prod_{k=i}^j \mathbb{P}(s'[k]|(s, a)[Z_k^P])$. For every $V : \mathcal{S} \to \mathbb{R}$ and the right-linear operators $\mathbb{P}$, we define $\mathbb{P}V(s, a) \stackrel{\text{def}}{=} \sum_{s' \in \mathcal{S}} \mathbb{P}(s' \mid s, a)V(s')$. A state-action pair can be represented as $(s, a)$ or $x$. We also use $(s, a)[Z]$ to denote the corresponding $x[Z]$ for notation convenience. We mainly focus on the case where the total time step $T = KH$ is the dominant factor, and assume that $T \geq |\mathcal{X}_i| \geq H$ during the analysis.

## 3 RELATED WORK

**Exploration in Reinforcement Learning**    Recent years have witnessed a tremendous of work for provably efficient exploration in reinforcement learning, including tabular MDP (e.g., Dann et al., 2017; Azar et al., 2017; Jin et al., 2018; Zanette & Brunskill, 2019), linear RL (e.g., Jiang et al., 2017; Yang & Wang, 2019; Jin et al., 2020; Zanette et al., 2020), and RL with general function approximation (e.g., Osband & Van Roy, 2014a; Ayoub et al., 2020; Wang et al., 2020). For algorithms in tabular setting, the regret bounds inevitably depend on the cardinality of state-action space, which may be excessively large. Based on the concept of eluder dimension (Russo & Van Roy, 2013), many recent works proposed efficient algorithms for RL with general function approximation (Osband & Van Roy, 2014a; Ayoub et al., 2020; Wang et al., 2020). Since eluder dimension of the function class for factored MDPs is at most $O\left(\sum_{i=1}^m |\mathcal{X}[Z_i^R]| + \sum_{i=1}^n |\mathcal{X}[Z_i^P]|\right)$, it is possible to apply their algorithms and regret bounds to our setting, though the direct application of their algorithms leads to a loose regret bound.

**Factored MDP**    Episodic FMDP was studied by Osband & Van Roy (2014b), in which they proposed both PSRL and UCRL style algorithm with near-optimal Bayesian and frequentist regret bound. In non-episodic scenarios, Xu & Tewari (2020) recently generalizes the algorithm of Osband & Van Roy (2014b) to the infinite horizon average reward setting. However, both their results suffer from linear dependence on the horizon (or the diameter) and factored state space's cardinality.

Concurrent with our work is the recent paper by Tian et al. (2020), which also applies UCBVI and EULER to factored MDPs. Compared with their results, we propose a more refined variance

decomposition theorem for factored Markov chains (Theorem 1), which results in a better regret by a factor of $\sqrt{n}$; the theorem is also of independent interest with potential use in other problems in factored MDPs. Furthermore, we formulate the RLwK problem, and provide a sample-efficient algorithm based on our FMDP algorithm.

**Constrained MDP and knapsack bandits**  The knapsack setting with hard constraints has already been studied in bandits with both sample-efficient and computational-efficient algorithms (Badanidiyuru et al., 2013; Agrawal et al., 2016). This setting may be viewed as a special case of RLwK with $H = 1$. In constrained RL, there is a line of works that focus on *soft constraints* where the constraints are satisfied in expectation or with high probability (Brantley et al., 2020; Zheng & Ratliff, 2020), or a violation bound is established (Efroni et al., 2020; Ding et al., 2020). RLwK requires stronger constraints that is almost surely satisfied during the execution of the agents. A more related setting is proposed by Brantley et al. (2020), which studies a sample-efficient algorithm for knapsack episodic setting with hard constraints on all $K$ episodes. However, we require the constraints to be satisfied *within each episode*, which we believe can better describe the real-world scenarios. The setting of Singh et al. (2020) is closer to ours since they are focusing on "every-time" hard constraints, although they consider the non-episodic case.

## 4    MAIN RESULTS

In this section, we introduce our FMDP-BF algorithm, which uses empirical variance to construct a Bernstein-type confidence bound for value estimation. Besides FMDP-BF, we also propose a simpler algorithm called FMDP-CH with a slightly worse regret, which follows the similar idea of UCBVI-CH (Azar et al., 2017). The algorithm and the corresponding analysis are more concise and easy to understand; details are deferred to Section B.

### 4.1    ESTIMATION ERROR DECOMPOSITION

Our algorithm will follow the principle of "optimism in the face of uncertainty". Like ORLC (Dann et al., 2019) and EULER (Zanette & Brunskill, 2019), our algorithm also maintains both the optimistic and pessimistic estimates of state values to yield an improved regret bound. We use $\overline{V}_{k,h}$ and $\underline{V}_{k,h}$ to denote the optimistic estimation and pessimistic estimation of $V_h^*$, respectively. To guarantee optimism, we need to add confidence bonus to the estimated value function $\overline{V}_{k,h}$ at each step, so that $\overline{V}_{k,h}(s) \geq V_h^*(s)$ holds for any $k \in [K]$, $h \in [H]$ and $s \in \mathcal{S}$. Suppose $\hat{R}_{k,i}$ and $\hat{\mathbb{P}}_{k,j}$ denote the estimated value of each expected factored reward $R_i$ and factored transition probability $\mathbb{P}_j$ before episode $k$ respectively. By the definition of the reward $R$ and the transition $\mathbb{P}$, we use $\hat{R} \overset{\text{def}}{=} \frac{1}{m} \sum_{i=1}^m \hat{R}_{k,i}$ and $\hat{\mathbb{P}}_k \overset{\text{def}}{=} \prod_{j=1}^n \hat{\mathbb{P}}_{k,j}$ as the estimation of $\bar{R}$ and $\mathbb{P}$. Following the previous framework, this confidence bonus needs to tightly characterize the estimation error of the one-step backup $\bar{R}(s,a) + \mathbb{P}V_h^*(s,a)$; in other words, it should compensate for the estimation errors, $\left(\hat{R}_k - \bar{R}\right)(s,a)$ and $\left(\hat{\mathbb{P}}_k - \mathbb{P}\right)V_h^*(s,a)$, respectively.

For the estimation error of rewards $\left(\hat{R}_k - \bar{R}\right)(s_{k,h}, a_{k,h})$, since the reward is defined as the average of $m$ factored rewards, it is not hard to decompose the estimation error of $\bar{R}(s,a)$ to the average of the estimation error of each factored rewards. In that case, we separately construct the confidence bonus of each factored reward $\bar{R}_i$. Suppose $CB_{k,Z_i^R}^R(s,a)$ is the confidence bonus that compensates for the estimation error $\hat{R}_{k,i} - \bar{R}_i$, then we have $CB_k^R(s,a) \overset{\text{def}}{=} \frac{1}{m} \sum_{i=1}^m CB_{k,Z_i^R}^R(s,a)$.

For the estimation error of transition $\left(\hat{\mathbb{P}}_k - \mathbb{P}\right)V_{h+1}^*(s_{k,h}, a_{k,h})$, the main difficulty is that $\hat{\mathbb{P}}_k$ is the multiplication of $n$ estimated transition dynamics $\hat{\mathbb{P}}_{k,i}$. In that case, the estimation error $\left(\hat{\mathbb{P}}_k - \mathbb{P}\right)V_{h+1}^*(s_{k,h}, a_{k,h})$ may be calculated as the multiplication of $n$ estimation error for each factored transition $\hat{\mathbb{P}}_{k,i}$, which makes the analysis much more difficult. Fortunately, we have the following lemma to address this challenge.

**Lemma 4.1.** *(Informal)  Let the transition function class $\mathbb{P} \in \mathcal{P}(\mathcal{X}, \mathcal{S})$ be factored over $\mathcal{X} = \mathcal{X}_1 \times \cdots \times \mathcal{X}_d$, and $\mathcal{S} = \mathcal{S}_1 \times \cdots \times \mathcal{S}_n$ with scopes $Z_1^P, \cdots, Z_n^P$. For a given function $V : \mathcal{S} \to \mathbb{R}$, the estimation error of one-step value $|(\hat{\mathbb{P}}_k - \mathbb{P})V(s, a)|$ can be decomposed by:*

$$|(\hat{\mathbb{P}}_k - \mathbb{P})V(s, a)| \leq \sum_{i=1}^n \left| (\hat{\mathbb{P}}_{k,i} - \mathbb{P}_i) \left( \prod_{j \neq i, j=1}^n \mathbb{P}_j \right) V(s, a) \right| + \beta_{k,h}(s, a)$$

Here, $\beta_{k,h}(s, a)$, formally defined in Lemma E.1, are higher order terms that do not harm the order of the regret. This lemma allows us to decompose the estimation error $\left( \hat{\mathbb{P}}_k - \mathbb{P} \right) V_{h+1}^*(s_{k,h}, a_{k,h})$ into an *additive* form, so we can construct the confidence bonus for each factored transition $\mathbb{P}_j$ separately.  Let $CB_{k,Z_j^P}^P(s, a)$ be the confidence bonus for the estimation error $(\hat{\mathbb{P}}_{k,j} - \mathbb{P}_j) \left( \prod_{t \neq j, t=1}^n \mathbb{P}_t \right) V(s, a)$.  Then, $CB_k^P(s, a) \overset{\text{def}}{=} \sum_{j=1}^n CB_{k,Z_j^P}^P(s, a) + \eta_{k,h}(s, a)$, where $\eta_{k,h}(s, a)$ collects higher order factors that will be explicitly given later.

Finally, we define the confidence bonus as the summation of all confidence bonuses for rewards and transition: $CB_k(s, a) = CB_k^R(s, a) + CB_k^P(s, a)$.

## 4.2 VARIANCE OF FACTORED MARKOV CHAINS

After the analysis in Section 4.1, the remaining problem is how to define the confidence bonus $CB_{k,Z_i^R}^R(s, a)$ and $CB_{k,Z_j^P}^P(s, a)$. In this subsection, we tackle this problem by deriving the variance decomposition formula for factored MDP. To begin with, we consider Markov chains with stochastic factored transition and stochastic factored rewards, and deduce the Bellman equation of variance for factored Markov chains. The analysis shows how to define the empirical variance in the confidence bonus for factored MDP and gives an upper bound on the summation of per-step variance (Corollary 1.1).

In the Markov chain setting, the reward is defined to be a mapping from $\mathcal{S}$ to $\mathbb{R}$. Suppose $J_{t_1:t_2}(s)$ denotes the total rewards the agent obtains from step $t_1$ to step $t_2$ (inclusively), given that the agent starts from state $s$ in step $t_1$.  $J_{t_1:t_2}$ is a random variable depending on the randomness of the trajectory from step $t_1$ to $t_2$, and stochastic rewards therein. Following this definition of $J_{t_1:t_2}$, we define $J_{1:H}$ to be the total reward obtained during one episode. We use $s_t$ to denote the random state that the agent encounters at step $t$. We define $\omega_h^2(s) \overset{\text{def}}{=} \mathbb{E}\left[ (J_{h:H}(s_h) - V_h(s_h))^2 \mid s_h = s \right]$ to be the variance of the total gain after step $h$, given that $s_h = s$.

We define $\sigma_{R,i}^2(s) \overset{\text{def}}{=} \mathbb{V}\left[ R_i(\xi) \mid \xi = s \right]$ to be the variance of the $i$-th factored reward, given that the current state is $s$.  Given the current state $s$, we define the variance of the next-state value function w.r.t. the $i$-th factored transition as: $\sigma_{P,i,h}^2(s) \overset{\text{def}}{=} \mathbb{E}_{s_{h+1}[1:i-1]} \left[ \mathbb{V}_{s_{h+1}[i]} \left[ \mathbb{E}_{s_{h+1}[i+1:n]} \left[ V_{h+1}(s_{h+1}) \right] \right] \mid s_h = s \right]$. That is, for each given $s'[1 : i]$, we firstly take expectation over all possible values of $s'[i + 1 : n]$ w.r.t. $\mathbb{P}_{[i+1:n]}$. Then, we calculate the variance of transition $s'[i] \sim \mathbb{P}_i(\cdot | (s, a)[Z_i^P])$ given fixed $s'[1 : i - 1]$. Finally, we take the expectation of this variance w.r.t. $s'[1 : i - 1] \sim \mathbb{P}_{[1:i-1]}$.

**Theorem 1.** *For any horizon $h \in [H]$, we have $\omega_h^2(s) = \sum_{s'} \mathbb{P}(s' | s) \omega_{h+1}^2(s') + \sum_{i=1}^n \sigma_{P,i,h}^2(s) + \frac{1}{m^2} \sum_{i=1}^m \sigma_{R,i}^2(s)$.*

Theorem 1 generalizes the analysis of Munos & Moore (1999), which deals with non-factored MDPs and deterministic rewards. From the Bellman equation of variance, we can give an upper bound to the expected summation of per-step variance.

**Corollary 1.1.** *Suppose the agent takes policy $\pi$ during an episode. Let $w_h(s, a)$ denote the probability of entering state $s$ and taking action $a$ in step $h$. Then we have the following inequality:*

$$\sum_{h=1}^H \sum_{(s,a) \in \mathcal{X}} w_h(s, a) \left( \sum_{i=1}^n \sigma_{P,i}^2(V_{h+1}^\pi, s, a) + \frac{1}{m^2} \sum_{i=1}^m \sigma_{R,i}^2(s, a) \right) \leq H^2,$$

where $\sigma_{R,i}^2(s,a) \overset{def}{=} \mathbb{V}\left[r_i(\xi,\zeta)|\xi=s,\zeta=a\right]$ *is the variance of $i$-th factored reward given the current state-action pair* $(s,a)$, *and* $\sigma_{P,i}^2(V_{h+1}^\pi,s,a) = \mathbb{E}_{s_{h+1}[1:i-1]}\left[\mathbb{V}_{s_{h+1}[i]}\left[\mathbb{E}_{s_{h+1}[i+1:n]}\left[V_{h+1}^\pi(s_{h+1})\right]\right] \mid s_h=s\right]$ *is the variance of $i$-th factored transition given current state $s$.*

This corollary makes it possible to construct confidence bonus with variance for each factored rewards and transition separately. Please refer to Section F.2 for the detailed proof of Theorem 1 and Corollary 1.1.

## 4.3 ALGORITHM

Our algorithm is formally described in Alg. 1, with a more detailed explanation in Section C. Denote by $N_k((s,a)[Z])$ the number of steps that the agent encounters $(s,a)[Z]$ during the first $k$ episodes. In episode $k$, we estimate the mean value of each factored reward $R_i$ and each factored transition $\mathbb{P}_i$ with empirical mean value $\hat{R}_{k,i}$ and $\hat{\mathbb{P}}_{k,i}$ of the previous history data $\mathcal{L}$ respectively. After that, we construct the optimistic MDP $\hat{M}$ based on the estimated rewards and transition functions. For a certain $(s,a)$ pair, the transition function and reward function are defined as $\hat{R}_k(s,a) = \frac{1}{m}\sum_{i=1}^m \hat{R}_{k,i}((s,a)[Z_i^R])$ and $\hat{\mathbb{P}}_k(s' \mid s,a) = \prod_{j=1}^n \hat{\mathbb{P}}_{k,j}\left(s'[j] \mid (s,a)\left[Z_j^P\right]\right)$.

---

**Algorithm 1** FMDP-BF

    **Input**: $\delta$
    $\mathcal{L}=\emptyset$, initialize $N((s,a)[Z_i])=0$ for any factored set $Z_i$ and any $(s,a)[Z_i] \in \mathcal{X}[Z_i]$
    **for** episode $k=1,2,\cdots$ **do**
        Set $\overline{V}_{k,H+1}(s) = \underline{V}_{k,H+1}(s) = 0$ for all $s,a$.
5:       Estimate the empirical mean $\hat{R}_k$ and $\hat{\mathbb{P}}_k$ with history data $\mathcal{L}$.
        **for** horizon $h=H,H-1,...,1$ **do**
            **for** $s \in \mathcal{S}$ **do**
                **for** $a \in \mathcal{A}$ **do**
                    $\overline{Q}_{k,h}(s,a) = \min\{H,\hat{R}_k(s,a)+CB_k(s,a)+\hat{\mathbb{P}}_k\overline{V}_{k,h+1}(s,a)\}$
10:             **end for**
            $\pi_{k,h}(s) = \arg\max_a \overline{Q}_{k,h}(s,a)$
            $\overline{V}_{k,h}(s) = \max_{a\in\mathcal{A}}\overline{Q}_{k,h}(s,a)$
            $\underline{V}_{k,h}(s) = \max\left\{0,\hat{R}_k(s,\pi_{k,h}(s))-CB_k(s,\pi_{k,h}(s))+\hat{\mathbb{P}}_k\underline{V}_{k,h+1}(s,\pi_{k,h}(s))\right\}$
        **end for**
15:     **end for**
        Take action according to $\pi_{k,h}$ for $H$ steps in this episode.
        Update $\mathcal{L} = \mathcal{L}\bigcup\{s_{k,h},a_{k,h},r_{k,h},s_{k,h+1}\}_{h=1,2,...,H}$, and update counter $N_{k-1}((s,a)[Z_i])$.
    **end for**

---

Following the analysis in Section 4.1, we separately construct the confidence bonus of each factored reward $R_i$ with the empirical variance: $CB_{k,Z_i^R}^R(s,a) = \sqrt{\frac{2\hat{\sigma}_{R,k,i}^2(s,a)L_i^R}{N_{k-1}((s,a)[Z_i^R])} + \frac{8L_i^R}{3N_{k-1}((s,a)[Z_i^R])}}, \quad i \in [m]$, where $L_i^R \overset{\text{def}}{=} \log\left(18mT\left|\mathcal{X}[Z_i^R]\right|/\delta\right)$, and $\hat{\sigma}_{R,k,i}^2$ is the empirical variance of the $i$-th factored reward $R_i$, i.e. $\hat{\sigma}_{R,k,i}^2(s,a) = \frac{1}{N_{k-1}((s,a)[Z_i^P])}\sum_{t=1}^{(k-1)H}\mathbb{1}\left[(s_t,a_t)[Z_i^R]=(s,a)[Z_i^R]\right]\cdot r_{t,i}^2 - \left(\hat{R}_{k,i}((s,a)[Z_i^R])\right)^2$.

We define $L^P = \log\left(18nTSA/\delta\right)$ for short. Following the idea of Lemma 4.1, we separately construct the confidence bonus of scope $Z_i^P$ for transition estimation: $CB_{k,Z_i^P}^P(s,a) = \sqrt{\frac{4\hat{\sigma}_{P,k,i}^2(\overline{V}_{k,h+1},s,a)L^P}{N_{k-1}((s,a)[Z_i^P])}} + \sqrt{\frac{2u_{k,h,i}(s,a)L^P}{N_{k-1}((s,a)[Z_i^P])}} + \eta_{k,h,i}(s,a), \quad i \in [n]$, where $\hat{\sigma}_{P,k,i}(s,a)$ and $u_{k,h,i}(s,a)$ are defined later. $\eta_{k,h,i}(s,a)$ collects the additional bonus terms that do not affect the order of the final regret. The precise expression of $\eta_{k,h,i}(s,a)$ is deferred to Section C.

The definition of $\hat{\sigma}^2_{P,k,i}(\overline{V}_{k,h+1}, s, a)$ corresponds to $\sigma^2_{P,i}(V^\pi_h, s, a)$ in Corollary 1.1, which can be regarded as the empirical variance of transition $\mathbb{P}_{k,i}$: $\hat{\sigma}^2_{P,k,i}(\overline{V}_{k,h+1}, s, a) =$
$$\mathbb{E}_{s'[1:i-1]\sim\hat{\mathbb{P}}_{k,[1:i-1]}(\cdot|s,a)} \left[ \mathbb{V}_{s'[i]\sim\hat{\mathbb{P}}_{k,i}(\cdot|(s,a)[Z^P_i])} \left( \mathbb{E}_{s'[i+1:n]\sim\hat{\mathbb{P}}_{k,[i+1:n]}(\cdot|s,a)} \overline{V}_{k,h+1}(s') \right) \right].$$

To guarantee optimism, we need to use the empirical variance $\hat{\sigma}^2_{P,k,i}(V^*, s, a)$ to upper bound the estimation error in the proof. Since we do not know $V^*$ beforehand, we use $\hat{\sigma}^2_{P,k,i}(\overline{V}_{k,h+1}, s, a)$ as a surrogate in the confidence bonus. However, we cannot guarantee that $\hat{\sigma}^2_{P,k,i}(V^*, s, a)$ is upper bounded by $\hat{\sigma}^2_{P,k,i}(\overline{V}_{k,h+1}, s, a)$. To compensate for the error due to the difference between $V^*_{h+1}$ and $\overline{V}_{k,h+1}$, we add $\sqrt{\frac{2u_{k,h,i}(s,a)L^P}{N_{k-1}((s,a)[Z^P_i])}}$ to the confidence bonus, where $u_{k,h,i}(s,a)$ is defined as:
$$u_{k,h,i}(s,a) = \mathbb{E}_{s'_{[1:i]}\sim\hat{\mathbb{P}}_{k,[1:i]}(\cdot|s,a)} \left[ \left( \mathbb{E}_{s'_{[i+1:n]}\sim\hat{\mathbb{P}}_{k,[i+1:n]}(\cdot|s,a)} \left( \overline{V}_{k,h+1} - \underline{V}_{k,h+1} \right)(s') \right)^2 \right].$$

## 4.4 REGRET

**Theorem 2.** *Suppose $\left|\mathcal{X}[Z^R_i]\right| \le J^R, \left|\mathcal{X}[Z^P_j]\right| \le J^P$ for $i \in [m], j \in [n]$, then with prob. $1 - \delta$, the regret of Alg. 1 is $\mathcal{O}\left( \sqrt{J^R T \log(mT J^R/\delta)} \log T + \sqrt{nH J^P T \log(nTSA/\delta)} \log T \right)$.*

Note that the regret bound does not depend on the cardinalities of state and action spaces, but only has a square-root dependence on the cardinality of each factored subspace $\mathcal{X}[Z_i]$. By leveraging the structure of factored MDP, we achieve regret that scales exponentially smaller compared with that of UCBVI (Azar et al., 2017). The best previous regret bound for episodic factored MDP is achieved by Osband & Van Roy (2014b). When transformed to our setting, it becomes $\mathcal{O}\left( \sqrt{J^R T \log(mT J^R/\delta)} + nH\sqrt{\Gamma J^P T \log(nT J^P/\delta)} \right)$, where $\Gamma$ is an upper bound of $|\mathcal{S}_j|$. This is worse than our results by a factor of $\sqrt{nH\Gamma}$. Concurrent to our results, Tian et al. (2020) also propose efficient algorithms for episodic factored MDP. When transformed to our setting, their regret bound is $\tilde{\mathcal{O}}\left( \sqrt{J^R T} + n\sqrt{H J^P T} \right)$. Compared with their bounds, we further improve the regret by a factor of $\sqrt{n}$ with a more refined variance decomposition theorem (Theorem 1).

## 4.5 LOWER BOUND

In this subsection, we propose the regret lower bound for factored MDP. The proof of Theorem 3 is deferred to Section G.

**Theorem 3.** *Suppose $\log_2\left(|\mathcal{S}_i|\right) \le \frac{H}{2}$ for any $i \in [n]$, the regret of any algorithm on the factored MDP problem is lower bounded by $\Omega\left( \frac{1}{m} \sum_{i=1}^m \sqrt{\left|\mathcal{X}[Z^R_i]\right| T} + \frac{1}{n} \sum_{i=1}^n \sqrt{\left|\mathcal{X}[Z^P_i]\right| HT} \right)$.*

The lower bound of Tian et al. (2020) is $\Omega\left( \max\left\{ \max_i \sqrt{\left|\mathcal{X}[Z^R_i]\right| T}, \max_j \sqrt{\left|\mathcal{X}[Z^P_i]\right| HT} \right\} \right)$, which is derived from different hard instance construction. Their lower bound is of the same order with ours, while our bound measures the dependence on all the parameters including the number of the factored transition $n$ and the factored rewards $m$. If $\left|\mathcal{X}[Z^R_i]\right| = J^R$ and $\left|\mathcal{X}[Z^P_i]\right| = J^P$, the lower bound turns out to be $\Omega\left( \sqrt{J^R T} + \sqrt{H J^P T} \right)$, which matches the upper bound in Theorem 2 except for a factor of $\sqrt{n}$ and logarithmic factors.

## 5 RL WITH KNAPSACK CONSTRAINTS

In this section, we study RL with Knapsack constraints, or RLwK, as an application of FMDP-BF.

## 5.1 PRELIMINARIES

We generalize bandit with knapsack constraints or BwK (Badanidiyuru et al., 2013; Agrawal et al., 2016) to episodic MDPs. We consider the setting of tabular episodic Markov decision process,

$(\mathcal{S}, \mathcal{A}, H, \mathbb{P}, R, \boldsymbol{C})$, which adds to an episodic MDP with a $d$-dimensional stochastic cost vector, $\boldsymbol{C}(s, a)$. We use $\boldsymbol{C}_i(s, a)$ to denote the $i$-th cost in the cost vector $\boldsymbol{C}(s, a)$. If the agent takes action $a$ in state $s$, it receives reward $r$ sampled from $R(s, a)$, together with cost $\boldsymbol{c}$, before transitioning to the next state $s'$ with probability $\mathbb{P}(s'|s, a)$. In each episode, the agent's total budget is $\boldsymbol{B}$. We also use $\boldsymbol{B}_i$ to denote the total budget of $i$-th cost. Without loss of generality, we assume $\boldsymbol{B}_i \leq B$ for all $i$. An episode terminates after $H$ steps, or when the cumulative cost $\sum_h \boldsymbol{c}_{h,i}$ of any dimension $i$ exceeds the budget $\boldsymbol{B}_i$, whichever occurs first. The agent's goal is to maximize its cumulative reward $\sum_{k=1}^{K} \sum_{h=1}^{H} r_{k,h}$ in $K$ episodes.

## 5.2 COMPARISON WITH OTHER SETTINGS

While RLwK might appear similar to episodic constrained RL (Efroni et al., 2020; Brantley et al., 2020), it is fundamentally different, so those algorithms cannot be applied here.

As discussed in Section 3, the episodic constrained RL setting can be roughly divided into two categories. A line of works focus on soft constraints where the constraints are satisfied in expectation, i.e. $\sum_{h=1}^{H} \mathbb{E}[\boldsymbol{c}_{k,h}] \leq \boldsymbol{B}$. The expectation is taken over the randomness of the trajectories and the random sample of the costs. Another line of work focuses on hard constraints in $K$ episodes. To be more specific, they assume that the total costs in $K$ episodes cannot exceed a constant vector $\boldsymbol{B}$, i.e. $\sum_{k=1}^{K} \sum_{h=1}^{H} \boldsymbol{c}_{k,h} \leq \boldsymbol{B}$. Once this is violated before episode $K_1 < K$, the agent will not obtain any rewards in the remaining $K - K_1$ episodes. Though both settings are interesting and useful, they do not cover many common situations in constrained RL. For example, when playing games, the game is over once the total energy or health reduce to $0$. After that, the player may restart the game (starting a new episode) with full initial energy again. In robotics, a robot may episodically interact with the environment and learn a policy to carry out a certain task. The interaction in each episode is over once its energy is used up. In these two examples, we cannot just consider the expected cost or the cumulative cost across *all* episodes, but calculate the cumulative cost in every *individual* episode. Moreover, in many constrained RL applications, the agent's optimal action should depend on its remaining budget. For example, in robotics, the robot should do planning and take actions based on its remaining energy. However, previous results do not consider this issue, and use policies that map states to actions. Instead, in RLwK, we need to define the policy as a mapping from states and remaining budget to actions. Section H gives further details, including two examples for illustrating the difference between these settings.

## 5.3 ALGORITHM

We make the following assumptions about the cost function for simplicity. Both of them hold if all the stochastic costs are integers with an upper bound.

**Assumption 1.** *The budget $\boldsymbol{B}_i$ as well as the possible value of costs $\boldsymbol{C}_i(s, a)$ of any state $s$ and action $a$ is an integral multiple of the unit cost $\frac{1}{m}$.*

**Assumption 2.** *The stochastic cost $\boldsymbol{C}_i(s, a)$ has finite support. That is, the random variable $\boldsymbol{C}_i(s, a)$ can only take at most $n$ possible values.*

The reason for Assumption 2 is that we need to estimate the distribution of the cost, instead of just estimating its mean value. We discuss the necessity of the assumptions and the possible methods for continuous distribution in Section H.3.

From the above discussion, we know that we need to find a policy that is a mapping from state and budget to action. Therefore, it is natural to augment the state with the remaining budget. It follows that the size of augmented state space is $S \cdot (Bm)^d$. Directly applying UCBVI algorithm (Azar et al., 2017) will lead to a regret of order $O\left(\sqrt{HSAT(Bm)^d}\right)$. Our key observation is that the constructed state representation can be represented as a product of subspaces. Each subspace is relatively independent. For example, the transition matrix over the original state space $\mathcal{S}$ is independent of the remaining budget. Therefore, the constructed MDP can be formulated as a factored MDP, and the compact structure of the model can reduce the regret significantly.

By applying Alg. 1 and Theorem 2 to RLwK, we can reduce the regret to the order of $O\left(\sqrt{HSA(1 + dBm)T}\right)$ roughly, which is exponentially smaller. However, the regret still de-

pends on the total budget $B$ and the discretization precision $m$, which may be very large for continuous budget and cost. Another observation to tackle the problem is that the cost of taking action $a$ on state $s$ only depends on the current state-action pair $(s, a)$, but has no dependence on the remaining budget $\boldsymbol{B}$. To be more formal, we have $\boldsymbol{b}_{h+1} = \boldsymbol{b}_h - \boldsymbol{c}_h$, where $\boldsymbol{b}_h$ is the remaining budget at step $h$, and $\boldsymbol{c}_h$ is the cost suffered in step $h$. As a result, we can further reduce the regret to roughly $O\left(\sqrt{HdSAT}\right)$ by estimating the distribution of cost function. A similar model has been discussed in Brunskill et al. (2009), which is named as noisy offset model.

Our algorithm, which is called FMDP-BF for RLwK, follows the same basic idea of Alg. 1. We defer the detailed description to Section H to avoid redundance. The regret can be upper bounded by the following theorem:

**Theorem 4.** *With prob. at least $1 - \delta$, the regret of Alg. 4 is upper bounded by*

$$\mathcal{O}\left(\sqrt{dHSAT\left(\log(SAT) + d\log(Bm)\right)}\right)$$

Compared with the lower bound for non-factored tabular MDP (Jaksch et al., 2010), this regret bound matches the lower bound w.r.t. $S$, $A$, $H$ and $T$. There may still be a gap in the dependence of the number of constraints $d$, which is often much smaller than other quantities.

It should be noted that, though we achieve a near-optimal regret for RLwK, the computational complexity is high, scaling polynomially with the maximum budget $B$, and exponentially with the number of constraints $d$. This is a consequence of the NP-hardness of knapsack problem with multiple constraints (Martello, 1990; Kellerer et al., 2004). However, since the policy is defined on the state and budget space with cardinality $SB^d$, this computational complexity seems unavoidable. How to tackle this problem, such as with approximation algorithms, is an interesting future work.

## 6 CONCLUSION

We propose a novel RL algorithm for solving FMDPs with near optimal regret guarantee. It improves the best previous regret bound by a factor of $\sqrt{nH|\mathcal{S}_i|}$. We also derive a regret lower bound for FMDPs based on the minimax lower bound of multi-armed bandits and episodic tubular MDPs (Jaksch et al., 2010). Further, we formulate the RL with Knapsack constraints (RLwK) setting, and establish the connections between our results for FMDP and RLwK by providing a sample efficient algorithm based on FMDP-BF in this new setting.

A few problems remain open. The regret upper and lower bounds have a gap of approximately $\sqrt{n}$, where $n$ is the number of transition factors. For RLwK, it is important to develop a computationally efficient algorithm, or find a variant of the hard-constraint formulation. We hope to address these issues in the future work.

## ACKNOWLEDGEMENTS

This work was supported by Key-Area Research and Development Program of Guangdong Province (No. 2019B121204008)], National Key R&D Program of China (2018YFB1402600), BJNSF (L172037) and Beijing Academy of Artificial Intelligence.

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

# A    NOTATIONS

Before presenting the proof, we restate the definition of the following notations.

| Symbol | Explanation |
|---|---|
| $s_{k,h}, a_{k,h}$ | The state and action that the agent encounters in episode $k$ and step $h$ |
| $L^P$ | $\log\left(18nTSA/\delta\right)$ |
| $L_i^R$ | $\log\left(18mT\left\|\mathcal{X}[Z_i^R]\right\|/\delta\right)$ |
| $X_i^P$ | $\left\|\mathcal{X}[Z_i^P]\right\|$ |
| $X_i^R$ | $\left\|\mathcal{X}[Z_i^R]\right\|$ |
| $N_k((s,a)[Z])$ | the number of steps that the agent encounters $(s,a)[Z]$ during the first $k$ episodes |
| $\mathbb{P}V(s,a)$ | A shorthand of $\sum_{s'\in\mathcal{S}}\mathbb{P}(s'\|s,a)V(s')$ |
| $\phi_{k,i}(s,a)$ | $\sqrt{\frac{4\|\mathcal{S}_j\|L^P}{N_{k-1}((s,a)[Z_j^P])}}+\frac{4\|\mathcal{S}_j\|L^P}{3N_{k-1}((s,a)[Z_j^P])}$ |
| $\hat{\sigma}_{R,i}^2(s,a)$ | The empirical variance of reward $R_i$ |
| $\sigma_{P,i}^2(V,s,a)$ | the next state variance of $\mathbb{P}V$ for the transition $\mathbb{P}_i$, |
| | i.e. $\mathbb{E}_{s'[1:i-1]\sim\mathbb{P}_{[1:i]}(\cdot\|s,a)}\left[\mathbb{V}_{s'[i]\sim\mathbb{P}_i(\cdot\|(s,a)[Z_i^P])}\left(\mathbb{E}_{s'[i+1:n]\sim\mathbb{P}_{[i+1:n]}(\cdot\|s,a)}V(s')\right)\right]$ |
| $\hat{\sigma}_{P,k,i}(V,s,a)$ | the empirical next state variance of $\hat{\mathbb{P}}_kV$ for the transition $\hat{\mathbb{P}}_{k,i}$, |
| | i.e. $\mathbb{E}_{s'[1:i-1]\sim\hat{\mathbb{P}}_{k,[1:i]}(\cdot\|s,a)}\left[\mathbb{V}_{s'[i]\sim\hat{\mathbb{P}}_{k,i}(\cdot\|(s,a)[Z_i^P])}\left(\mathbb{E}_{s'[i+1:n]\sim\hat{\mathbb{P}}_{k,[i+1:n]}(\cdot\|s,a)}V(s')\right)\right]$ |
| $\Omega_{k,h}$ | The optimism and pessimism event for $k,h$: $\left\{\overline{V}_{k,h}\geq V_h^*\geq \underline{V}_{k,h}\right\}$ |
| $\Omega$ | The optimism and pessimism events for all $1\leq k\leq K, 1\leq h\leq H$, i.e. $\cup_{k,h}\Omega_{k,h}$ |
| $w_{k,h,Z}(s,a)$ | The probability of entering $(s,a)[Z]$ at step $h$ in episode $k$ |
| $w_{k,Z}(s,a)$ | $\sum_{h=1}^{H}w_{k,h,Z}(s,a)$ |
| $w_{k,h}(s,a)$ | The probability of entering $(s,a)$ at step $h$ in episode $k$, |
| | i.e. $w_{k,h,Z}(s,a)$ with $Z=\{1,2,...,d\}$ |
| $w_k(s,a)$ | $\sum_{h=1}^{H}w_{k,h}(s,a)$ |

# B    OMITTED DETAILS FOR FMDP-CH

In this section, we introduce our algorithm with Hoeffding-type confidence bonus and present the corresponding regret bound. Our algorithm, which is described in Algorithm 2, is related to UCBVI-CH algorithm (Azar et al., 2017), in the sense that Algorithm 2 reduces to UCBVI-CH if we consider a flat MDP with $m=n=d=1$.

Let $N_k((s,a)[Z])$ denote the number of steps that the agent encounters $(s,a)[Z]$ during the first $k$ episodes, and $N_k((s,a)[Z_j],s_j)$ denotes the number of steps that the agent transits to a state with $s[j]=s_j$ after encountering $(s,a)[Z_j]$ during the first $k$ episodes. In episode $k$, we estimate the mean value of each factored reward $R_i$ and each factored transition $\mathbb{P}_i$ with empirical mean value $\hat{R}_{k,i}$ and $\hat{\mathbb{P}}_{k,i}$ respectively. To be more specific, $\hat{R}_{k,i}((s,a)[Z_i^R])=\frac{\sum_{t\leq(k-1)H}\mathbb{1}\left[(s_t,a_t)[Z_i^R]=(s,a)[Z_i^R]\right]\cdot r_{t,i}}{N_{k-1}((s,a)[Z_i^R])}$, where $r_{t,i}$ denotes the reward $R_i$ sampled in step $t$, and $\hat{\mathbb{P}}_{k,j}\left(s[j]\|(s,a)[Z_j^P]\right)=\frac{N_{k-1}((s,a)[Z_j^P],s[j])}{N_{k-1}((s,a)[Z_j^P])}$. After that, we construct the optimistic MDP $\hat{M}$ based on the estimated rewards and transition functions. For a certain $(s,a)$ pair, the transition function and reward function are defined as $\hat{R}_k(s,a)=\frac{1}{m}\sum_{i=1}^{m}\hat{R}_{k,i}((s,a)[Z_i^R])$ and $\hat{\mathbb{P}}_k(s'\mid s,a)=\prod_{j=1}^{n}\hat{\mathbb{P}}_{k,j}\left(s'[j]\mid(s,a)[Z_j^P]\right)$.

We define $L_i^R=\log\left(18mT\left\|\mathcal{X}[Z_i^R]\right\|/\delta\right)$, $L^P=\log(18nTSA/\delta)$ and $\phi_{k,i}(s,a)=\sqrt{\frac{4\|\mathcal{S}_i\|L^P}{N_{k-1}((s,a)[Z_i^P])}}+\frac{4\|\mathcal{S}_i\|L}{3N_{k-1}((s,a)[Z_i^P])}$. We separately construct the confidence bonus of each fac-

---

**Algorithm 2** FMDP-CH

---

    **Input**: $\delta$,
    history data $\mathcal{L} = \emptyset$, initialize $N((s,a)[Z_i]) = 0$ for any factored set $Z_i$ and $(s,a)[Z_i] \in \mathcal{X}[Z_i]$
    **for** episode $k = 1, 2, ...$ **do**
        Set $\hat{V}_{k,H+1}(s) = 0$ for all $s$.
5:      Estimate $\hat{R}_{k,i}(s,a)$ with empirical mean value if $N_{k-1}((s,a)[Z_i^R]) > 0$, otherwise
        $\hat{R}_{k,i}(s,a) = 1$, then calculate $\hat{R}(s,a) = \frac{1}{m}\sum_{i=1}^m \hat{R}_i((s,a)[Z_i^R])$
        Let $\mathcal{K}_P = \left\{(s,a) \in \mathcal{S} \times \mathcal{A}, \cup_{i \in [n]} N_k((s,a)[Z_i^P]) > 0\right\}$
        Estimate $\hat{P}_k(\cdot|s,a)$ with empirical mean value for all $(s,a) \in \mathcal{K}_P$
        **for** horizon $h = H, H-1, ..., 1$ **do**
            **for** all $(s,a) \in \mathcal{S} \times \mathcal{A}$ **do**
10:             **if** $(s.a) \in \mathcal{K}_P$ **then**
                $\hat{Q}_{k,h}(s,a) = \min\{H, \hat{R}_k(s,a) + CB_k(s,a) + \hat{\mathbb{P}}_k \hat{V}_{k,h+1}(s,a)\}$
             **else**
                $\hat{Q}_{k,h}(s,a) = H$
             **end if**
15:             $\hat{V}_{k,h}(s) = \max_{a \in \mathcal{A}} \hat{Q}_{k,h}(s,a)$
            **end for**
        **end for**
        **for** step $h = 1, \cdots, H$ **do**
            Take action $a_{k,h} = \arg\max_a \hat{Q}_{k,h}(s_{k,h}, a)$
20:      **end for**
        Update history trajectory $\mathcal{L} = \mathcal{L} \bigcup \{s_{k,h}, a_{k,h}, r_{k,h}, s_{k,h+1}\}_{h=1,2,...,H}$, and update history
        counter $N_{k-1}((s,a)[Z_i])$.
    **end for**

---

tored reward $R_i$ and factored transition $\mathbb{P}_i$ in the following way:

$$CB_{k,Z_i^R}^R(s,a) = \sqrt{\frac{2L_i^R}{N_{k-1}((s,a)[Z_i^R])}}, \quad i \in [m] \tag{1}$$

$$CB_{k,Z_i^P}^P(s,a) = \sqrt{\frac{2H^2 L^P}{N_{k-1}((s,a)[Z_i^P])}} + H\phi_{k,i}(s,a) \sum_{j=1,j\neq i}^n \phi_{k,j}(s,a), \quad i \in [n] \tag{2}$$

We define the confidence bonus as the summation of all confidence bonus for rewards and transition, i.e. $CB_k(s,a) = \frac{1}{m}\sum_{i=1}^m CB_{k,Z_i^R}^R(s,a) + \sum_{j=1}^n CB_{k,Z_j^P}^P(s,a)$.

We propose the following regret upper bound for Alg. 2.

**Theorem 5.** *With prob. $1 - \delta$, the regret of Alg. 2 is upper bounded by*

$$\text{Reg}(K) = \mathcal{O}\left(\frac{1}{m}\sum_{i=1}^m \sqrt{|\mathcal{X}[Z_i^R]|\, T \log(mT|\mathcal{X}[Z_i^R]|/\delta)} + \sum_{j=1}^n H\sqrt{|\mathcal{X}[Z_j^P]|\, T \log(nTSA/\delta)}\right)$$

*Here $\mathcal{O}$ hides the lower order terms with respect to $T$.*

## C   OMITTED DETAILS IN SECTION 4

In this section, we clarify the omitted details in Section 4. The detailed algorithm is described in Alg. 3. we denote $N_k((s,a)[Z])$ as the number of steps that the agent encounters $(s,a)[Z]$ during the first $k$ episodes, and $N_k((s,a)[Z_j], s_j)$ as the number of steps that the agent transits to a state with $s[j] = s_j$ after encountering $(s,a)[Z_j]$ during the first $k$ episodes. In episode $k$, we estimate the mean value of each factored reward $R_i$ and each factored transition $\mathbb{P}_i$ with empirical mean value $\hat{R}_{k,i}$ and $\hat{\mathbb{P}}_{k,i}$ respectively. To be more specific, $\hat{R}_{k,i}((s,a)[Z_i^R]) =$

$\frac{\sum_{t \le (k-1)H} \mathbb{1}\left[(s_t, a_t)[Z_i^R] = (s,a)[Z_i^R]\right] \cdot r_{t,i}}{N_{k-1}((s,a)[Z_i^R])}$, where $r_{t,i}$ denotes the reward $R_i$ sampled in step $t$, and
$\hat{\mathbb{P}}_{k,j}\left(s[j] | (s,a)[Z_j^P]\right) = \frac{N_{k-1}((s,a)[Z_j^P], s[j])}{N_{k-1}((s,a)[Z_j^P])}$.

The formal definition of the confidence bonus for Alg. 1 is:

$$CB_{k,Z_i^R}^R(s,a) = \sqrt{\frac{2\hat{\sigma}_{R,k,i}^2(s,a)L_i^R}{N_{k-1}((s,a)[Z_i^R])}} + \frac{8L_i^R}{3N_{k-1}((s,a)[Z_i^R])} \tag{3}$$

$$CB_{k,Z_i^P}^P(s,a) = \sqrt{\frac{4\hat{\sigma}_{P,k,i}^2(\overline{V}_{k,h+1}, s, a)L^P}{N_{k-1}((s,a)[Z_i^P])}} + \sqrt{\frac{2u_{k,h,i}(s,a)L^P}{N_{k-1}((s,a)[Z_i^P])}} \tag{4}$$

$$+ \sqrt{\frac{16H^2L^P}{N_{k-1}((s,a)[Z_i^P])}} \sum_{j=1}^n \left( \left(\frac{4|\mathcal{S}_j|L^P}{N_{k-1}((s,a)[Z_j^P])}\right)^{\frac{1}{4}} + \sqrt{\frac{4|\mathcal{S}_j|L^P}{3N_{k-1}(s,a)[Z_j^P]}} \right) \tag{5}$$

$$+ \sum_{j=1}^n H\phi_{k,i}(s,a)\phi_{k,j}(s,a), \tag{6}$$

where $\phi_{k,i}(s,a) = \sqrt{\frac{4|\mathcal{S}_j|L^P}{N_{k-1}((s,a)[Z_j^P])}} + \frac{4|\mathcal{S}_j|L^P}{3N_{k-1}((s,a)[Z_j^P])}$. The definition of $\eta_{k,h,i}(s,a)$ is

$$\sqrt{\frac{16H^2L^P}{N_{k-1}((s,a)[Z_i^P])}} \sum_{j=1}^n \left( \left(\frac{4|\mathcal{S}_j|L^P}{N_{k-1}((s,a)[Z_j^P])}\right)^{\frac{1}{4}} + \sqrt{\frac{4|\mathcal{S}_j|L^P}{3N_{k-1}(s,a)[Z_j^P]}} \right) + \sum_{j=1}^n H\phi_{k,i}(s,a)\phi_{k,j}(s,a).$$

**Theorem 6.** *(Refined Statement of Theorem 2) With prob. at least $1 - \delta$, the regret of Alg. 1 is upper bounded by*

$$\mathcal{O}\left( \frac{1}{m} \sum_{i=1}^m \sqrt{\left|\mathcal{X}[Z_i^R]\right| T \log(mT \left|\mathcal{X}[Z_i^R]\right| / \delta) \log T} + \sqrt{\sum_{i=1}^n H \left|\mathcal{X}[Z_i^P]\right| T \log(nTSA/\delta) \log T} \right).$$

For clarity, we also present a cleaner single-term regret bound under a symmetric problem setting. Suppose $\mathcal{M}$ is a set of factored MDP with $m = n$, $|\mathcal{S}_i| = S_i$, $|\mathcal{X}_i| = S_iA_i$ and $|Z_i^R| = |Z_j^P| = \zeta$ for $i = 1, ..., m$ and $j = 1, ..., n$, we write $X_i = (S_iA_i)^\zeta$ and assume that $X_i \le J$ and $S_i \le \Gamma$.

**Corollary 6.1.** *Suppose $M^* \in \mathcal{M}$, with prob. $1 - \delta$, the regret of FMDP-BF is upper bounded by* $\mathcal{O}\left( \sqrt{nHJT \log(nTSA/\delta)} \right)$.

The minimax regret bound for non-factored MDP is $\mathcal{O}\left( \sqrt{HSAT \log(SAT/\delta)} \right)$. Compared with this result, our algorithm's regret is exponentially smaller when $n$ and $\zeta$ are relatively small. Under this problem setting, the regret of Osband & Van Roy (2014b) is $\mathcal{O}\left( nH\sqrt{\Gamma JT \log(nJT)} \right)$. Our results is better by a factor of $\sqrt{nH\Gamma}$.

# D  HIGH PROBABILITY EVENTS

In this section, we discuss the high-prob. events, and assume that these events happen during the proof.

---

**Algorithm 3** FMDP-BF (Detailed Description of Alg. 1)

---

    **Input**: $\delta$
    $\mathcal{L} = \emptyset$, initialize $N((s,a)[Z_i]) = 0$ for any factored set $Z_i$ and any $(s,a)[Z_i] \in \mathcal{X}[Z_i]$
    **for** episode $k = 1, 2, \cdots$ **do**
        Set $\overline{V}_{k,H+1}(s) = \underline{V}_{k,H+1}(s) = 0$ for all $s, a$.
5:      Let $\mathcal{K} = \left\{ (s,a) \in \mathcal{S} \times \mathcal{A} : \cap_{i=1,\dots,n} N_k((s,a)[Z_i^P]) > 0 \right\}$
        Estimate $\hat{R}_{k,i}(s,a)$ as the empirical mean if $N_{k-1}((s,a)[Z_i^R]) > 0$, and 1 otherwise
        $\hat{R}(s,a) = \frac{1}{m} \sum_{i=1}^{m} \hat{R}_i((s,a)[Z_i^R])$
        Estimate $\hat{P}_k(\cdot|s,a)$ with empirical mean value for all $(s,a) \in \mathcal{K}$
        **for** horizon $h = H, H-1, ..., 1$ **do**
10:        **for** $s \in \mathcal{S}$ **do**
            **for** $a \in \mathcal{A}$ **do**
                **if** $(s,a) \in \mathcal{K}$ **then**
                    $\overline{Q}_{k,h}(s,a) = \min\{H, \hat{R}_k(s,a) + CB_k(s,a) + \hat{\mathbb{P}}_k \overline{V}_{k,h+1}(s,a)\}$
                **else**
15:                  $\overline{Q}_{k,h}(s,a) = H$
                **end if**
            **end for**
            $\pi_{k,h}(s) = \arg\max_a \overline{Q}_{k,h}(s,a)$
            $\overline{V}_{k,h}(s) = \max_{a \in \mathcal{A}} \overline{Q}_{k,h}(s,a)$
20:            $\underline{V}_{k,h}(s) = \max\left\{ 0, \hat{R}_k(s, \pi_{k,h}(s)) - CB_k(s, \pi_{k,h}(s)) + \hat{\mathbb{P}}_k \underline{V}_{k,h+1}(s, \pi_{k,h}(s)) \right\}$
        **end for**
        **end for**
        **for** step $h = 1, \cdots, H$ **do**
            Take action $a_{k,h} = \arg\max_a \overline{Q}_{k,h}(s_{k,h}, a)$
25:      **end for**
        Update history trajectory $\mathcal{L} = \mathcal{L} \bigcup \{s_{k,h}, a_{k,h}, r_{k,h}, s_{k,h+1}\}_{h=1,2,\dots,H}$, and update history
        counter $N_{k-1}((s,a)[Z_i])$.
    **end for**

---

**Lemma D.1.** *(High prob. event) With prob. at least $1 - 2\delta/3$, the following events hold for any $k, h, s, a$:*

$$|\hat{R}_{k,i}((s,a)[Z_i^R]) - \bar{R}_i((s,a)[Z_i^R])| \leq \sqrt{\frac{2L_i^R}{N_{k-1}((s,a)[Z_i^R])}}, \quad i \in [m] \tag{7}$$

$$|\hat{\mathbb{P}}_{k,i} \prod_{j \neq i} \mathbb{P}_{k,j} V_h^*(s,a) - \prod_{j=1}^n \mathbb{P}_j V_h^*(s,a)| \leq \sqrt{\frac{2H^2 L^P}{N_{k-1}((s,a)[Z_i^P])}}, \quad i \in [n] \tag{8}$$

$$|(\hat{\mathbb{P}}_{k,i} - \mathbb{P}_{k,i})(\cdot|(s,a)[Z_i^P])|_1 \leq 2\sqrt{\frac{|\mathcal{S}_i| L^P}{N_{k-1}((s,a)[Z_i^P])}} + \frac{4|\mathcal{S}_i| L^P}{3N_{k-1}((s,a)[Z_i^P])} \quad i \in [n] \tag{9}$$

$$|(\hat{\mathbb{P}}_{k,i} - \mathbb{P}_{k,i})(s'|(s,a)[Z_i^P])| \leq \sqrt{\frac{2\mathbb{P}_i(s'[i]|(s,a)[Z_i^P]) L^P}{N_{k-1}((s,a)[Z_i^P])}} + \frac{L^P}{3N_{k-1}((s,a)[Z_i^P])} \quad i \in [n] \tag{10}$$

$$\sum_{k=1}^K \sum_{h=1}^H \left( \mathbb{P}\left( \hat{V}_{k,h+1} - V_{h+1}^{\pi_k} \right)(s_{k,h}, a_{k,h}) - \left( \hat{V}_{k,h+1} - V_{h+1}^{\pi_k} \right)(s_{k,h+1}) \right) \leq \sqrt{2HT \log(18SAT)} \tag{11}$$

$$\sum_{k=1}^K \sum_{h=1}^H \left( \mathbb{P}\left( \hat{V}_{k,h+1} - V_{h+1}^* \right)(s_{k,h}, a_{k,h}) - \left( \hat{V}_{k,h+1} - V_{h+1}^* \right)(s_{k,h+1}) \right) \leq \sqrt{2HT \log(18SAT)} \tag{12}$$

We define the above events as $\Lambda_1$, and assume it happens during the proof.

*Proof.* By Hoeffding's inequality and union bounds over all $i \in [m]$, step $k \in [K]$ and $(s,a) \in \mathcal{X}[Z_i^R]$, we know that Inq. 7 holds with prob. $1 - \frac{\delta}{9}$ for any $i \in [m], k \in [K], (s,a) \in \mathcal{X}[Z_i^P]$. Similarly, by Hoeffding's inequality and union bounds over all $i \in [n]$, step $t$ and $(s,a) \in \mathcal{X}$, Inq. 8 also holds with prob. $1 - \frac{\delta}{9}$ for any $i, s, a, k$. Inq. 9 is the high probability bound on the $L_1$ norm of the Maximum Likelihood Estimate, which is proved by Weissman et al. (2003). Inq. 10 can be proved with the use of Bernstein inequality and union bound (See Azar et al. (2017) for a similar derivation). Inq. 11 and Inq. 12 can be regarded as the summation of martingale difference sequences, which can be derived with the application of Azuma's inequality. Finally, we take union bounds over all these inequalities, which indicates that $\Lambda_1$ holds with prob. at least $1 - 2\delta/3$. $\qquad\square$

For the proof of Thm. 2, we also need to consider the following high-prob. events. We define the following events as $\Lambda_2$. During the proof of Thm. 2, we assume both $\Lambda_1$ and $\Lambda_2$ happen.

**Lemma D.2.** *With prob. at least $1 - \delta/3$, the following events hold for any $k, h, s, a$:*

$$\left| \hat{R}_{k,i}((s,a)[Z_i^R]) - \bar{R}_{k,i}((s,a)[Z_i^R]) \right| \leq \sqrt{\frac{2\hat{\sigma}_{R,i}^2(s,a)L_i^R}{N_{k-1}((s,a)[Z_i^R])}} + \frac{8L_i^R}{3N_{k-1}((s,a)[Z_i^R])}, \quad i \in [m]$$

(13)

$$\left| (\hat{\mathbb{P}}_{k,i} - \mathbb{P}_i) \prod_{j \neq i} \mathbb{P}_j V_{h+1}^*(s,a) \right| \leq \sqrt{\frac{2\sigma_{P,i}^2(V_{h+1}^*, s, a)L^P}{N_{k-1}((s,a)[Z_i^P]))}} + \frac{2HL^P}{3N_{k-1}((s,a)[Z_i^P])}, \quad i \in [n] \quad (14)$$

$$N_k((s,a)[Z_i^P]) \geq \frac{1}{2} \sum_{j < k} w_{j, Z_i^P}(s,a) - H \log(18nX_i^P H/\delta), i \in [n]$$

(15)

*Proof.* Inq. 13 can be proved directly by empirical Bernstein inequality. Now we mainly focus on Inq. 14. By Bernstein's inequality and union bounds over all $s, a, k, h$, we know that the following inequality holds with prob. at least $1 - \frac{\delta}{9}$.

$$\left| (\hat{\mathbb{P}}_{k,i} - \mathbb{P}_i) \prod_{j \neq i} \mathbb{P}_j V_{h+1}^*(s,a) \right|$$

$$= \sum_{s'[1:i-1] \in \mathcal{X}[1:i-1]} \mathbb{P}(s'[1:i-1]|s,a) \left| (\hat{\mathbb{P}}_{k,i} - \mathbb{P}_i) \prod_{j=i+1}^{n} \mathbb{P}_j V_{h+1}^*(s,a) \right|$$

$$\leq \sum_{s'[1:i-1] \in \mathcal{X}[1:i-1]} \mathbb{P}(s'[1:i-1]|s,a) \sqrt{\frac{2 \operatorname{Var}_{s'[i] \sim \mathbb{P}_i(\cdot|(s,a)[Z_i^P])} \left( \mathbb{E}_{s'[i+1:n] \sim \mathbb{P}_{[i+1:n]}(\cdot|s,a)} V_{h+1}(s') \mid s'[1:i-1] \right) L^P}{N_{k-1}((s,a)[Z_i^P])}}$$

$$+ \sum_{s'[1:i-1] \in \mathcal{X}[1:i-1]} \mathbb{P}(s'[1:i-1]|s,a) \frac{2HL^P}{3N_{k-1}((s,a)[Z_i^P])}$$

$$\leq \sqrt{\frac{2\sigma^2(V_{h+1}^*, s, a)L^P}{N_{k-1}((s,a)[Z_i^P]))}} + \frac{2HL^P}{3N_{k-1}((s,a)[Z_i^P])}$$

The last inequality is due to Jensen's inequality. That is,

$$\sum_{s'[1:i-1]} \mathbb{P}(s'[1:i-1]) \sqrt{\frac{C_1}{N_{k-1}((s,a)[Z_i^P])}} \leq \sqrt{\sum_{s'[1:i-1]} \frac{C_1 \mathbb{P}(s'[1:i-1])}{N_{k-1}((s,a)[Z_i^P])}}$$

$$= \sqrt{2\sigma^2(V_{h+1}^*, s, a)L^P \cdot \frac{1}{N_{k-1}((s,a)[Z_i^P])}}$$

where $\mathbb{P}(s'[1:i-1])$ is a shorthand of $\mathbb{P}(s'[1:i-1]|s,a)$, and $C_1$ here denotes

$$2 \operatorname{Var}_{s'[i] \sim \mathbb{P}_i(\cdot|(s,a)[Z_i^P])} \left( \mathbb{E}_{s'[i+1:n] \sim \mathbb{P}_{[i+1:n]}(\cdot|s,a)} V(s') \mid s'[1:i-1] \right) L^P.$$

Inq. 15 follows the same proof of the failure event $F^N$ in section B.1 of Dann et al. (2019). $\qquad\square$

# E  PROOF OF THEOREM 5

## E.1  ESTIMATION ERROR DECOMPOSITION

**Lemma E.1.** *The estimation error can be decomposed in the following way:*

$$|(\hat{\mathbb{P}}_k - \mathbb{P})(\cdot|s,a)|_1 \le \sum_{i=1}^n |(\hat{\mathbb{P}}_{k,i} - \mathbb{P}_i)(\cdot|(s,a)[Z_i^P])|_1 \tag{16}$$

$$|(\hat{\mathbb{P}}_k - \mathbb{P})V(s,a)| \le \sum_{i=1}^n \left| (\hat{\mathbb{P}}_{k,i} - \mathbb{P}_i) \left( \prod_{j \ne i, j=1}^n \mathbb{P}_j \right) V(s,a) \right|$$

$$+ \sum_{i=1}^n \sum_{j \ne i, j=1}^n |V|_\infty \left| \left( \hat{\mathbb{P}}_{k,i} - \mathbb{P}_i \right) (\cdot|(s,a)[Z_i^P]) \right|_1 \cdot \left| \left( \hat{\mathbb{P}}_{k,j} - \mathbb{P}_j \right) (\cdot|(s,a)[Z_j^P]) \right|_1, \tag{17}$$

*here $V$ denotes any value function mapping from $\mathcal{S}$ to $\mathbb{R}$, e.g. $V_{h+1}^*$ or $\hat{V}_{k,h+1} - V_{h+1}^*$.*

*Proof.* Inq. 16 has the same form of Lemma 32 in Li (2009) and Lemma 1 in Osband & Van Roy (2014b). We mainly focus on Inq. 17. We can decompose the difference in the following way:

$$\left| (\hat{\mathbb{P}}_k - \mathbb{P})V(s,a) \right|$$

$$\le \left| (\hat{\mathbb{P}}_{k,n} - \mathbb{P}_n) \prod_{i=1}^{n-1} \mathbb{P}_i V(s,a) \right| + \left| \mathbb{P}_n (\prod_{i=1}^{n-1} \hat{\mathbb{P}}_{k,i} - \prod_{i=1}^{n-1} \mathbb{P}_i) V(s,a) \right| + \left| (\hat{\mathbb{P}}_{k,n} - \mathbb{P}_n)(\prod_{i=1}^{n-1} \hat{\mathbb{P}}_{k,i} - \prod_{i=1}^{n-1} \mathbb{P}_i) V^*(s,a) \right| \tag{18}$$

For the last term of Inq. 18, we have

$$\left| (\hat{\mathbb{P}}_{k,n} - \mathbb{P}_n)(\prod_{i=1}^{n-1} \hat{\mathbb{P}}_{k,i} - \prod_{i=1}^{n-1} \mathbb{P}_i) V(s,a) \right|$$

$$\le \left| \left( \hat{\mathbb{P}}_{k,n} - \mathbb{P}_n \right) (\cdot|(s,a)[Z_n^P]) \right|_1 \cdot \left| \prod_{i=1}^{n-1} \hat{\mathbb{P}}_{k,i}(\cdot|s,a[Z_i^P]) - \prod_{i=1}^{n-1} \mathbb{P}_i(\cdot|s,a[Z_i^P]) \right|_1 \cdot |V|_\infty$$

$$\le \left| \left( \hat{\mathbb{P}}_{k,n} - \mathbb{P}_n \right) (\cdot|(s,a)[Z_n^P]) \right|_1 \sum_{i=1}^{n-1} \left| \left( \hat{\mathbb{P}}_{k,i} - \mathbb{P}_i \right) (\cdot|(s,a)[Z_i^P]) \right|_1 \cdot |V|_\infty,$$

Where the last inequality is due to Inq. 16.

For the second part of Inq. 18, we can further decompose the term as:

$$\left| \mathbb{P}_n \left( \prod_{i=1}^{n-1} \hat{\mathbb{P}}_{k,i} - \prod_{i=1}^{n-1} \mathbb{P}_i \right) V(s,a) \right|$$

$$\le \left| \mathbb{P}_n \left( \hat{\mathbb{P}}_{k,n-1} - \mathbb{P}_{n-1} \right) \prod_{i=1}^{n-2} \mathbb{P}_i V(s,a) \right| + \left| \mathbb{P}_n \mathbb{P}_{n-1} \left( \prod_{i=1}^{n-2} \hat{\mathbb{P}}_{k,i} - \prod_{i=1}^{n-2} \mathbb{P}_i \right) V(s,a) \right|$$

$$+ \left| \mathbb{P}_n \left( \hat{\mathbb{P}}_{k,n-1} - \mathbb{P}_{n-1} \right) \left( \prod_{i=1}^{n-2} \hat{\mathbb{P}}_{k,i} - \prod_{i=1}^{n-2} \mathbb{P}_i \right) V(s,a) \right| \tag{19}$$

Following the same decomposition technique, we can prove Inq. 17 by recursively decomposing the second term over all possible $n$:

$$|(\hat{\mathbb{P}}_k - \mathbb{P})V^*(s,a)| \le \sum_{i=1}^n \left| (\hat{\mathbb{P}}_{k,i} - \mathbb{P}_i) \left( \prod_{j \ne i, j=1}^n \mathbb{P}_j \right) V(s,a) \right|$$

$$+ \sum_{i=1}^n \sum_{j \ne i, j=1}^n \left| \left( \hat{\mathbb{P}}_{k,i} - \mathbb{P}_i \right) (\cdot|(s,a)[Z_i^P]) \right|_1 \cdot \left| \left( \hat{\mathbb{P}}_{k,j} - \mathbb{P}_j \right) (\cdot|(s,a)[Z_j^P]) \right|_1 \cdot |V|_\infty$$

$\square$

**Lemma E.2.** *Under event $\Lambda_1$, then the following Inequality holds:*

$$|\hat{R}_k(s,a) - \bar{R}(s,a)| \leq \frac{1}{m} \sum_{i=1}^{m} \sqrt{\frac{2L_i^R}{N_{k-1}((s,a)[Z_i^R])}} \tag{20}$$

$$|(\hat{\mathbb{P}}_k - \mathbb{P})(\cdot|s,a)|_1 \leq \sum_{i=1}^{n} \left( \sqrt{\frac{4|\mathcal{S}_i|L^P}{N_{k-1}((s,a)[Z_i^P])}} + \frac{4|\mathcal{S}_i|L^P}{3N_{k-1}((s,a)[Z_i^P])} \right) \tag{21}$$

$$|(\hat{\mathbb{P}}_k - \mathbb{P})V^*(s,a)| \leq \sum_{i=1}^{n} \sqrt{\frac{2H^2 L^P}{N_{k-1}((s,a)[Z_i^P])}}$$

$$+ \sum_{i=1}^{n} \sum_{j \neq i, j=1}^{n} H \left( \sqrt{\frac{4|\mathcal{S}_i|L^P}{N_{k-1}((s,a)[Z_i^P])}} + \frac{4|\mathcal{S}_i|L^P}{3N_{k-1}((s,a)[Z_i^P])} \right) \left( \sqrt{\frac{4|\mathcal{S}_j|L^P}{N_{k-1}((s,a)[Z_j^P])}} + \frac{4|\mathcal{S}_j|L^P}{3N_{k-1}((s,a)[Z_j^P])} \right) \tag{22}$$

*Proof.* Inq. 20 can be proved by Lemma D.1:

$$|\hat{R}_k(s,a) - \bar{R}(s,a)| \leq \frac{1}{m} \sum_{i=1}^{m} |\hat{R}_k(s,a) - \bar{R}(s,a)| \leq \frac{1}{m} \sum_{i=1}^{m} \sqrt{\frac{2L_i^R}{N_{k-1}((s,a)[Z_i^R])}}$$

Inq. 21 follows directly by applying Lemma D.1 to Lemma E.1.

$$|(\hat{\mathbb{P}}_k - \mathbb{P})(\cdot|s,a)|_1 \leq \sum_{i=1}^{n} |(\hat{\mathbb{P}}_{k,i} - \mathbb{P}_i)(\cdot|(s,a)[Z_i^P])|_1 \leq \sum_{i=1}^{n} \left( \sqrt{\frac{4|\mathcal{S}_i|L^P}{N_{k-1}((s,a)[Z_i^P])}} + \frac{4|\mathcal{S}_i|L^P}{3N_{k-1}((s,a)[Z_i^P])} \right)$$

Similarly, Inq 22 can be proved by:

$$|(\hat{\mathbb{P}}_k - \mathbb{P})V^*(s,a)|$$

$$\leq \sum_{i=1}^{n} \left| (\hat{\mathbb{P}}_{k,i} - \mathbb{P}_i) \left( \prod_{j \neq i, j=1}^{n} \mathbb{P}_j \right) V^*(s,a) \right|$$

$$+ \sum_{i=1}^{n} \sum_{j \neq i, j=1}^{n} H \left| \left( \hat{\mathbb{P}}_{k,i} - \mathbb{P}_i \right)(\cdot|(s,a)[Z_i^P]) \right|_1 \cdot \left| \left( \hat{\mathbb{P}}_{k,j} - \mathbb{P}_j \right)(\cdot|(s,a)[Z_j^P]) \right|_1$$

$$\leq \sum_{i=1}^{n} \sqrt{\frac{2H^2 L^P}{N_{k-1}((s,a)[Z_i^P])}}$$

$$+ \sum_{i=1}^{n} \sum_{j \neq i, j=1}^{n} H \left( \sqrt{\frac{4|\mathcal{S}_i|L^P}{N_{k-1}((s,a)[Z_i^P])}} + \frac{4|\mathcal{S}_i|L^P}{3N_{k-1}((s,a)[Z_i^P])} \right) \left( \sqrt{\frac{4|\mathcal{S}_j|L^P}{N_{k-1}((s,a)[Z_j^P])}} + \frac{4|\mathcal{S}_j|L^P}{3N_{k-1}((s,a)[Z_j^P])} \right)$$

$\square$

### E.2 OPTIMISM

**Lemma E.3.** *(Optimism) Under event $\Lambda_1$, $\hat{V}_{k,h}(s) \geq V_h^*(s)$ for any $k, h, s$.*

*Proof.* We prove the Lemma by induction. Firstly, for $h = H + 1$, the inequality holds trivially since $\hat{V}_{k,H+1}(s) = V_{H+1}^*(s) = 0$.

$$\hat{V}_{k,h}(s) - V_h^*(s)$$

$$\geq \hat{R}_k(s, \pi_h^*(s)) + CB_k(s, \pi_h^*(s)) + \hat{\mathbb{P}}_k \hat{V}_{k,h+1}(s, \pi_h^*(s)) - \bar{R}(s, \pi_h^*(s)) - \mathbb{P}V_{h+1}^*(s, \pi_h^*(s))$$

$$= \hat{R}_k(s, \pi_h^*(s)) - \bar{R}(s, \pi_h^*(s)) + CB_k(s, \pi_h^*(s)) + \hat{\mathbb{P}}_k(\hat{V}_{k,h+1} - V_{h+1}^*)(s, \pi_h^*(s)) + (\hat{\mathbb{P}}_k - \mathbb{P})V_{h+1}^*(s, \pi^*(s))$$

$$\geq \hat{R}_k(s, \pi_h^*(s)) - \bar{R}(s, \pi_h^*(s)) + CB_k(s, \pi_h^*(s)) + (\hat{\mathbb{P}}_k - \mathbb{P})V_{h+1}^*(s, \pi^*(s))$$

$$\geq 0$$

The first inequality is due to $\hat{V}_{k,h}(s) \geq \hat{Q}_{k,h}(s, \pi_h^*(s))$. The second inequality follows by induction condition that $\hat{V}_{k,h+1}(s) \geq V_{h+1}^*(s)$ for all $s$. The last inequality is due to Inq. 20 and Inq. 22 in Lemma E.2. □

### E.3 PROOF OF THEOREM 5

Now we are ready to prove Thm. 5.

*Proof.* (Proof of Thm. 5)

$$
\begin{aligned}
& V_h^*(s_{k,h}) - V_h^{\pi_k}(s_{k,h}) \\
\leq & \hat{V}_{k,h}(s_{k,h}) - V_h^{\pi_k}(s_{k,h}) \\
= & \hat{R}_k(s_{k,h}, \pi_{k,h}(s_{k,h})) + \hat{\mathbb{P}}_k \hat{V}_{k,h+1}(s_{k,h}, \pi_{k,h}(s_{k,h})) + CB_k(s_{k,h}, \pi_{k,h}(s_{k,h})) \\
& - \bar{R}(s_{k,h}, \pi_{k,h}(s_{k,h})) - \mathbb{P}V_{h+1}^{\pi_k}(s_{k,h}, \pi_{k,h}(s_{k,h})) \\
= & \hat{V}_{k,h+1}(s_{k,h+1}) - V_{h+1}^{\pi_k}(s_{k,h+1}) + \hat{R}_k(s, \pi_{k,h}(s_{k,h})) - \bar{R}(s, \pi_{k,h}(s_{k,h})) + CB_k(s_{k,h}, \pi_h^*(s)) \\
& + \mathbb{P}\left(\hat{V}_{k,h+1} - V_{h+1}^{\pi_k}\right)(s_{k,h}, \pi_{k,h}(s_{k,h})) - \left(\hat{V}_{k,h+1} - V_{h+1}^{\pi_k}\right)(s_{k,h+1}) \\
& + \left(\hat{\mathbb{P}}_k - \mathbb{P}\right)V_{h+1}^*(s_{k,h}, \pi_{k,h}(s_{k,h})) \\
& + \left(\hat{\mathbb{P}}_k - \mathbb{P}\right)\left(\hat{V}_{k,h+1} - V_{h+1}^*\right)(s_{k,h}, \pi_{k,h}(s_{k,h}))
\end{aligned}
$$

The first inequality is due to optimism $\hat{V}_{k,h}(s_{k,h}) \geq V_h^*(s_{k,h})$. The first equality is due to Bellman equation for $V_h^{\pi_k}$ and $\hat{V}_{k,h}$.

For notation simplicity, we define

$$
\begin{aligned}
\delta_{k,h}^1 &= \hat{R}_k(s, \pi_{k,h}(s_{k,h})) - \bar{R}(s, \pi_{k,h}(s_{k,h})) \\
\delta_{k,h}^2 &= \mathbb{P}\left(\hat{V}_{k,h+1} - V_{h+1}^{\pi_k}\right)(s_{k,h}, \pi_{k,h}(s_{k,h})) - \left(\hat{V}_{k,h+1} - V_{h+1}^{\pi_k}\right)(s_{k,h+1}) \\
\delta_{k,h}^3 &= \left(\hat{\mathbb{P}}_k - \mathbb{P}\right)V_{h+1}^*(s_{k,h}, \pi_{k,h}(s_{k,h}))
\end{aligned}
$$

Firstly we focus on the upper bound of $\left(\hat{\mathbb{P}}_{k,h} - \mathbb{P}\right)\left(\hat{V}_{k,h+1} - V_{h+1}^*\right)(s_{k,h}, \pi_{k,h}(s_{k,h}))$. We bound this term following the idea of Azar et al. (2017).

$$
\begin{aligned}
& \left(\hat{\mathbb{P}}_k - \mathbb{P}\right)\left(\hat{V}_{k,h+1} - V_{h+1}^*\right)(s_{k,h}, a_{k,h}) \\
\leq & \sum_{i=1}^n (\hat{\mathbb{P}}_i - \mathbb{P}_i) \prod_{j=1, j\neq i}^n \mathbb{P}_j \left(\hat{V}_{k,h+1} - V_{h+1}^*\right)(s_{k,h}, a_{k,h}) \\
& + \sum_{i=1}^n \sum_{j=1}^n H \left|\left(\hat{\mathbb{P}}_i - \mathbb{P}_i\right)(\cdot|(s_{k,h}, a_{k,h})[Z_i^P])\right|_1 \left|\left(\hat{\mathbb{P}}_j - \mathbb{P}_j\right)(\cdot|(s_{k,h}, a_{k,h})[Z_j^P])\right|_1 \\
\leq & \sum_{i=1}^n \left(\sum_{s'[i]\in\mathcal{S}[i]} \sqrt{2\frac{\mathbb{P}_i(s'[i]|\mathcal{X}[Z_i^P])L^P}{N_{k-1}((s,a)[Z_i^P])}} + \frac{L^P}{3N_{k-1}((s,a)[Z_i^P])}\right) \prod_{j=1, j\neq i}^n \mathbb{P}_j \left(\hat{V}_{k,h+1} - V_{h+1}^*\right)(s_{k,h}, a_{k,h}) \\
& + \sum_{i=1}^n \sum_{j\neq i, j=1}^n H \left(\sqrt{\frac{4|\mathcal{S}_i|L^P}{N_{k-1}((s,a)[Z_i^P])}} + \frac{4|\mathcal{S}_i|L^P}{3N_{k-1}((s,a)[Z_i^P])}\right)\left(\sqrt{\frac{4|\mathcal{S}_j|L^P}{N_{k-1}((s,a)[Z_j^P])}} + \frac{4|\mathcal{S}_j|L^P}{3N_{k-1}((s,a)[Z_j^P])}\right) \\
\leq & \sum_{i=1}^n \sum_{s'[i]\in\mathcal{S}[i]} \sqrt{2\frac{\mathbb{P}_i(s'[i]|\mathcal{X}[Z_i^P])L^P}{N_{k-1}((s,a)[Z_i^P])}} \prod_{j=1, j\neq i}^n \mathbb{P}_j \left(\hat{V}_{k,h+1} - V_{h+1}^*\right)(s_{k,h}, a_{k,h}) + \sum_{i=1}^n \frac{|\mathcal{S}_i|HL^P}{3N_{k-1}((s,a)[Z_i^P])} \\
& + \sum_{i=1}^n \sum_{j\neq i, j=1}^n H \left(\sqrt{\frac{4|\mathcal{S}_i|L^P}{N_{k-1}((s,a)[Z_i^P])}} + \frac{4|\mathcal{S}_i|L^P}{3N_{k-1}((s,a)[Z_i^P])}\right)\left(\sqrt{\frac{4|\mathcal{S}_j|L^P}{N_{k-1}((s,a)[Z_j^P])}} + \frac{4|\mathcal{S}_j|L^P}{3N_{k-1}((s,a)[Z_j^P])}\right)
\end{aligned}
$$

The first inequality is due to Lemma E.1. The second inequality is because of Lemma D.1, and the last inequality is due to the fact that $\left|\hat{V}_{k,h+1} - V_{h+1}^*\right|_\infty \leq H$.

For each $i \in [n]$, we consider those $s'[i]$ satisfying $N_{k-1}((s,a)[Z_i^P])\mathbb{P}_i(s'[i]|(s_{k,h}, a_{k,h})[Z_i^P]) \geq 2n^2 H^2 L^P$ and $N_{k-1}((s,a)[Z_i^P])\mathbb{P}_i(s'[i]|(s_{k,h}, a_{k,h})[Z_i^P]) \leq 2n^2 H^2 L^P$ separately.

For those $s'[i]$ satisfying $N_{k-1}((s,a)[Z_i^P])\mathbb{P}_i(s'[i]|s_{k,h}, a_{k,h}) \geq 2n^2 H^2 L^P$, the first term can be bounded by

$$\sum_{i=1}^n \sum_{s'[i] \in \mathcal{S}[i]} \sqrt{2\frac{\mathbb{P}_i(s'[i]|\mathcal{X}[Z_i^P])L^P}{N_{k-1}((s,a)[Z_i^P])}} \prod_{j=1, j\neq i}^n \mathbb{P}_j\left(\hat{V}_{k,h+1} - V_{h+1}^*\right)(s_{k,h}, a_{k,h})$$

$$=\sum_{i=1}^n \sum_{s'[i] \in \mathcal{S}[i]} \mathbb{P}_i(s'[i]|\mathcal{X}[Z_i^P])\sqrt{2\frac{L^P}{\mathbb{P}_i(s'[i]|\mathcal{X}[Z_i^P])N_{k-1}((s,a)[Z_i^P])}} \prod_{j=1, j\neq i}^n \mathbb{P}_j\left(\hat{V}_{k,h+1} - V_{h+1}^*\right)(s_{k,h}, a_{k,h})$$

$$\leq\frac{1}{H}\mathbb{P}\left(\hat{V}_{k,h+1} - V_{h+1}^*\right)(s_{k,h}, a_{k,h})$$

$$=\frac{1}{H}\left(\hat{V}_{k,h+1} - V_{h+1}^*\right)(s_{k,h+1}, a_{k,h+1})$$

$$+ \frac{1}{H}\left(\mathbb{P}\left(\hat{V}_{k,h+1} - V_{h+1}^*\right)(s_{k,h}, a_{k,h}) - \left(\hat{V}_{k,h+1} - V_{h+1}^*\right)(s_{k,h+1}, a_{k,h+1})\right)$$

where the second term can be regarded as a martingale difference sequence, and we denote it as $\delta_{k,h}^4$.

For those $s'[i]$ satisfying $N_{k-1}((s,a)[Z_i^P])\mathbb{P}_i(s'[i]|s_{k,h}, a_{k,h}) \leq 2n^2 H^2 L^P$, the summation can be bounded by

$$\sum_{i=1}^n \frac{nH^2|\mathcal{S}_i|L^P}{N_{k-1}((s,a)[Z_i^P])}$$

For notation simplicity, we define $\delta_{k,h}^5$ as:

$$\delta_{k,h}^5 = \sum_{i=1}^n \sum_{j\neq i, j=1}^n H\left(\sqrt{\frac{4|\mathcal{S}_i|L^P}{N_{k-1}((s,a)[Z_i^P])}} + \frac{4|\mathcal{S}_i|L^P}{3N_{k-1}((s,a)[Z_i^P])}\right)\left(\sqrt{\frac{4|\mathcal{S}_j|L^P}{N_{k-1}((s,a)[Z_j^P])}} + \frac{4|\mathcal{S}_j|L^P}{3N_{k-1}((s,a)[Z_j^P])}\right)$$

$$+ \sum_{i=1}^n \frac{2nH^2|\mathcal{S}_i|L^P}{N_{k-1}((s,a)[Z_i^P])}$$

To sum up, by the above analysis, we prove that

$$\left(\hat{\mathbb{P}}_{k,h} - \mathbb{P}\right)\left(\hat{V}_{k,h+1} - V_{h+1}^*\right)(s_{k,h}, \pi_{k,h}(s_{k,h})) \leq \frac{1}{H}\left(\hat{V}_{k,h+1} - V_{h+1}^*\right)(s_{k,h+1}, a_{k,h+1}) + \delta_{k,h}^4 + \delta_{k,h}^5$$

Now we are ready to summarize all the terms in the regret. Firstly, we recursively calculate the regret for all $h \in [H]$.

$$V_1^*(s_{k,1}, a_{k,1}) - V_1^{\pi_k}(s_{k,1}, a_{k,1}) \leq \hat{V}_{k,1}(s_{k,1}, a_{k,1}) - V^{\pi_k}(s_{k,1}, a_{k,1})$$

$$\leq CB(s_{k,1}, a_{k,1}) + \delta_{k,1}^1 + \delta_{k,1}^2 + \delta_{k,1}^3 + \delta_{k,1}^4 + \delta_{k,1}^5 + (1 + \frac{1}{H})(V_{k,2}(s_{k,2}, a_{k,2}) - V_2^{\pi_k}(s_{k,2}, a_{k,2}))$$

$$\cdots$$

$$\leq \sum_{h=1}^H \left(1 + \frac{1}{H}\right)^{h-1} \left(CB_k(s_h, a_h) + \delta_{k,h}^1 + \delta_{k,h}^2 + \delta_{k,h}^3 + \delta_{k,h}^4 + \delta_{k,h}^5\right)$$

$$\leq \sum_{h=1}^H e\left(CB_k(s_h, a_h) + \delta_{k,h}^1 + \delta_{k,h}^2 + \delta_{k,h}^3 + \delta_{k,h}^4 + \delta_{k,h}^5\right)$$

Then we sum up the regret over $k$ episodes,

$$\text{Reg}(K) \leq \sum_{k=1}^{K} (V_1^*(s_1, a_1) - V_1^{\pi_k}(s_1, a_1))$$

$$\leq \sum_{k=1}^{K} \sum_{h=1}^{H} e\left(CB_k(s_h, a_h) + \delta_{k,h}^1 + \delta_{k,h}^2 + \delta_{k,h}^3 + \delta_{k,h}^4 + \delta_{k,h}^5\right)$$

$\delta_{k,h}^2$ and $\delta_{k,h}^4$ can be regarded as martingale difference sequence, the summation of which can be bounded by $O(H\sqrt{T\log(T)})$ by Lemma E.2, while $\delta_{k,h}^1$ and $\delta_{k,h}^3$ can also be bounded by Lemma E.2. The summation of different terms in $\delta_{k,h}^1, \delta_{k,h}^3, \delta_{k,h}^4$ and $\delta_{k,h}^5$ can be separated into the following categories. In the following proof, we use $C$ to denote the dependence of other parameters except the counters $N_k((s_{k,h}, a_{k,h})[Z_i])$

For those terms of the form $\frac{C}{\sqrt{N_k((s_{k,h}, a_{k,h})[Z_i])}}$, we have

$$\sum_k \sum_h \frac{C}{\sqrt{N_k((s_{k,h}, a_{k,h})[Z_i])}} \leq HC + \sum_{x[Z_i] \in \mathcal{X}[Z_i]} \sum_{c=1}^{N_K(x[Z_i])} \frac{C}{\sqrt{c}}$$

$$= HC + \sum_{x[Z_i] \in \mathcal{X}[Z_i]} C\sqrt{N_K(x[Z_i])}$$

$$\leq HC + C\sqrt{|\mathcal{X}[Z_i]|T}$$

The last inequality is due to Cauchy-Schwarz inequality. This term influence the main factors in the final regret.

For those terms of the form $\frac{C}{N_k((s_{k,h}, a_{k,h})[Z_i])}$, we have

$$\sum_k \sum_h \frac{C}{N_k((s_{k,h}, a_{k,h})[Z_i])} \leq HC + \sum_{x[Z_i] \in \mathcal{X}[Z_i]} \sum_{c=1}^{N_K(x[Z_i])} \frac{C}{c}$$

$$\leq HC + \sum_{x[Z_i] \in \mathcal{X}[Z_i]} C\ln(N_K(x[Z_i]))$$

$$\leq HC + C|\mathcal{X}[Z_i]|\ln T,$$

which has only logarithmic dependence on $T$.

For those terms of the form $\frac{C}{\sqrt{N_k((s_{k,h}, a_{k,h})[Z_i])N_k((s_{k,h}, a_{k,h})[Z_j])}}$. we define $N_k((s,a)[Z_i], (s,a)[Z_j])$ as the number of times that agent has encountered $(s,a)[Z_i]$ and $(s,a)[Z_j]$ simultaneously for the first $k$ episodes. It is not hard to find that $N_k((s,a)[Z_i]) \geq N_k((s,a)[Z_i], (s,a)[Z_j])$ and $N_k((s,a)[Z_j]) \geq N_k((s,a)[Z_i], (s,a)[Z_j])$.

$$\sum_k \sum_h \frac{C}{\sqrt{N_k((s_{k,h}, a_{k,h})[Z_i])N_k((s_{k,h}, a_{k,h})[Z_j])}}$$

$$\leq \sum_k \sum_h \frac{C}{N_k((s_{k,h}, a_{k,h})[Z_i], (s_{k,h}, a_{k,h})[Z_j])}$$

$$\leq HC + \sum_{x[Z_i \cup Z_j] \in \mathcal{X}[Z_i \cup Z_j]} C\ln(N_k((s_{k,h}, a_{k,h})[Z_i], (s_{k,h}, a_{k,h})[Z_j]))$$

$$\leq HC + C|\mathcal{X}[Z_i \cup Z_j]|\ln T,$$

which also has only logarithmic dependence on $T$.

For other terms with the form of $\frac{C}{(N_k((s_{k,h}, a_{k,h})[Z_i]))^2}$ and $\frac{C}{N_k((s_{k,h}, a_{k,h})[Z_i])\sqrt{N_k((s_{k,h}, a_{k,h})[Z_j])}}$, the summation of these terms has no dependence on $T$, which is negligible since $T$ is the dominant factor.

By bounding these different kinds of terms with the above methods, we can finally show that

$$\text{Reg}(K) = \mathcal{O}\left(\frac{1}{m}\sum_{i=1}^{m}\sqrt{|\mathcal{X}[Z_i^R]|T\log(10mT|\mathcal{X}[Z_i^R]|/\delta)} + \sum_{j=1}^{n}H\sqrt{|\mathcal{X}[Z_j^P]|T\log(10nTSA/\delta)}\right)$$

Here $\mathcal{O}$ hides the lower-order factors w.r.t $T$. $\qquad\square$

## F   PROOF OF THEOREM 2

### F.1   ESTIMATION ERROR DECOMPOSITION

**Lemma F.1.** *Under event $\Lambda_1$ and $\Lambda_2$, we have*

$$\left|\hat{R}_k(s,a) - \bar{R}(s,a)\right| \le \frac{1}{m}\sum_{i=1}^{m}\sqrt{\frac{2\hat{\sigma}_{R,i}^2(s,a)L_i^R}{N_{k-1}((s,a)[Z_i^R])}} + \frac{1}{m}\sum_{i=1}^{m}\frac{8L_i^R}{3N_{k-1}((s,a)[Z_i^R])}$$

$$|(\hat{\mathbb{P}}_k - \mathbb{P})V_{h+1}^*(s,a)|$$
$$\le \sum_{i=1}^{n}\left(\sqrt{\frac{2\sigma_{P,i}^2(V_{h+1}^*,s,a)L^P}{N_{k-1}((s,a)[Z_i^P]))}} + \frac{2HL^P}{3N_{k-1}((s,a)[Z_i^P])}\right)$$
$$+ \sum_{i=1}^{n}\sum_{j\ne i,j=1}^{n}H\left(\sqrt{\frac{4|\mathcal{S}_i|L^P}{N_{k-1}((s,a)[Z_i^P])}} + \frac{4|\mathcal{S}_i|L^P}{3N_{k-1}((s,a)[Z_i^P])}\right)\left(\sqrt{\frac{4|\mathcal{S}_j|L^P}{N_{k-1}((s,a)[Z_j^P])}} + \frac{4|\mathcal{S}_j|L^P}{3N_{k-1}((s,a)[Z_j^P])}\right)$$

*Proof.* The first inequality follows directly by the definition that $R(s,a) = \frac{1}{m}\sum_{i=1}^{m}R_i(s,a)$ and Lemma D.2. We now prove the second inequality. By Lemma E.1, we have

$$|(\hat{\mathbb{P}}_k - \mathbb{P})V_{h+1}^*(s,a)| \le \sum_{i=1}^{n}\left|(\hat{\mathbb{P}}_{k,i} - \mathbb{P}_i)\left(\prod_{j\ne i,j=1}^{n}\mathbb{P}_j\right)V_{h+1}^*(s,a)\right|$$
$$+ \sum_{i=1}^{n}\sum_{j\ne i,j=1}^{n}H\left|\left(\hat{\mathbb{P}}_{k,i} - \mathbb{P}_i\right)(\cdot|(s,a)[Z_i^P])\right|_1 \cdot \left|\left(\hat{\mathbb{P}}_{k,j} - \mathbb{P}_j\right)(\cdot|(s,a)[Z_j^P])\right|_1,$$

By Inq. 9 in Lemma D.1 and Inq. 14 in Lemma D.2, we have

$$|(\hat{\mathbb{P}} - \mathbb{P})V_{h+1}^*(s,a)|$$
$$\le \sum_{i=1}^{n}\left(\sqrt{\frac{2\sigma_{P,i}^2(V_{h+1}^*,s,a)L^P}{N_{k-1}((s,a)[Z_i^P]))}} + \frac{2HL^P}{3N_{k-1}((s,a)[Z_i^P])}\right)$$
$$+ \sum_{i=1}^{n}\sum_{j\ne i,j=1}^{n}H\left(\sqrt{\frac{4|\mathcal{S}_i|L^P}{N_{k-1}((s,a)[Z_i^P])}} + \frac{4|\mathcal{S}_i|L^P}{3N_{k-1}((s,a)[Z_i^P])}\right)\left(\sqrt{\frac{4|\mathcal{S}_j|L^P}{N_{k-1}((s,a)[Z_j^P])}} + \frac{4|\mathcal{S}_j|L^P}{3N_{k-1}((s,a)[Z_j^P])}\right)$$
$$\qquad\qquad\qquad\qquad\qquad\qquad\qquad\qquad\qquad\qquad\qquad\qquad\qquad\qquad\qquad\qquad\square$$

### F.2   OMITTED PROOF IN SECTION 4.2

*Proof.* (Proof of Theorem. 1)
$$\omega_h^2(s)$$
$$= \mathbb{E}\left[(J_{h:H}(s_h) - V_h(s_h))^2 \mid s_h = s\right]$$
$$= \sum_{s'}\mathbb{P}(s'|s)\mathbb{E}\left[(J_{h+1:H}(s_{h+1}) + r_h - V_h(s_h))^2 \mid s_h = s, s_{h+1} = s'\right]$$
$$= \sum_{s'}\mathbb{P}(s'|s)\mathbb{E}\left[J_{h+1:H}^2(s_{h+1}) + r_h^2 + V_h^2(s_h) \mid s_h = s, s_{h+1} = s'\right]$$
$$+ \sum_{s'}\mathbb{P}(s'|s)\mathbb{E}\left[2r_h(J_{h+1:H}(s_{h+1}) - V_h(s_h)) - 2J_{h+1:H}(s_{h+1})V_h(s_h) \mid s_h = s, s_{h+1} = s'\right]$$

Given $s_h = s, s_{h+1} = s'$, $r_h$, $V_h(s_h)$ and $J_{h+1:H}(s_{h+1})$ are conditionally independent, thus we have

$$\mathbb{E}[2r_h(J_{h+1:H}(s_{h+1}) - V_h(s_h)) \mid s_h = s, s_{h+1} = s'] = \mathbb{E}[2\bar{R}(s_h)(V_{h+1}(s_{h+1}) - V_h(s_h)) \mid s_h = s, s_{h+1} = s']$$

$$\mathbb{E}[2J_{h+1:H}(s_{h+1})V_h(s_h) \mid s_h = s, s_{h+1} = s'] = \mathbb{E}[2V_{h+1}(s_{h+1})V_h(s_h) \mid s_h = s, s_{h+1} = s']$$

Therefore, we have

$$
\begin{aligned}
&\omega_h^2(s) \\
&= \sum_{s'} \mathbb{P}(s'|s)\mathbb{E}\left[J_{h+1:H}^2(s_{h+1}) + r_h^2 + V_h^2(s_h) \mid s_h = s, s_{h+1} = s'\right] \\
&\quad + \sum_{s'} \mathbb{P}(s'|s)\mathbb{E}\left[2\bar{R}(s_h)(V_{h+1}(s_{h+1}) - V_h(s_h)) - 2V_{h+1}(s_{h+1})V_h(s_h) \mid s_h = s, s_{h+1} = s'\right] \\
&= \mathbb{E}\left[r_h^2 - \bar{R}^2(s_h) \mid s_h = s\right] + \sum_{s'} \mathbb{P}(s'|s)\mathbb{E}\left[J_{h+1:H}^2(s_{h+1}) - (V_h(s_h) - \bar{R}(s_h))^2 \mid s_h = s, s_{h+1} = s'\right] \\
&= \mathbb{E}\left[r_h^2 - \bar{R}^2(s_h) \mid s_h = s\right] + \sum_{s'} \mathbb{P}(s'|s)\mathbb{E}\left[J_{h+1:H}^2(s_{h+1}) - \left(\sum_{s''}\mathbb{P}(s''|s)V_{h+1}(s'')\right)^2 \mid s_h = s, s_{h+1} = s'\right] \\
&= \mathbb{E}\left[r_h^2 - \bar{R}^2(s_h) \mid s_h = s\right] + \sum_{s'} \mathbb{P}(s'|s)V_{h+1}^2(s') - \left(\sum_{s''}\mathbb{P}(s''|s)V_{h+1}(s'')\right)^2 \\
&\quad + \sum_{s'} \mathbb{P}(s'|s)\mathbb{E}\left[J_{h+1:H}^2(s_{h+1}) - V_{h+1}^2(s_{h+1}) \mid s_{h+1} = s'\right]
\end{aligned}
$$
(23)

The second equality is due to the fact that $V_h(s) = \bar{R}(s) + \sum_{s'}\mathbb{P}(s'|s)V_{h+1}(s')$.

For the factored rewards, since the rewards $r_{h,i}$ are conditionally independent give $s$, we have

$$\mathbb{E}\left[r_h^2 - \bar{R}^2(s_h) \mid s_h = s\right] = \frac{1}{m^2}\sum_{i=1}^{m}\sigma_{R,i}^2(s)$$

For the factored transition, we decompose the variance in the following way:

$$
\begin{aligned}
&\sum_{s'}\mathbb{P}(s'|s)V_{h+1}^2(s') - \left(\sum_{s''}\mathbb{P}(s''|s)V_{h+1}(s'')\right)^2 - \sigma_{P,1,h}^2(s) \\
&= \sum_{s'}\mathbb{P}(s'|s)V_{h+1}^2(s') - \sum_{s''[1]}\mathbb{P}(s''[1] \mid s)\left(\sum_{s''[2:n]}\mathbb{P}_{2:n}(s''[2:n]|s)V_{h+1}(s'')\right)^2 \\
&= \sum_{s'[1]}\mathbb{P}(s'[1] \mid s)\left(\sum_{s'[2:n]}\mathbb{P}_{2:n}(s'[2:n]|s)V_{h+1}^2(s') - \left(\sum_{s''[2:n]}\mathbb{P}_{2:n}(s''[2:n]|s)V_{h+1}([s'[1], s''[2:n]])\right)^2\right)
\end{aligned}
$$

Here $([s'[1], s''[2:n]]$ denotes the vector $s''$ with $s''[1]$ replaced with $s'[1]$. By subtracting $\sigma_{P,i,h}^2(s)$ for $i = 2, ..., n$ in the above way, we can show that

$$\sum_{s'}\mathbb{P}(s'|s)V_{h+1}^2(s') - \left(\sum_{s''}\mathbb{P}(s''|s)V_{h+1}(s'')\right)^2 - \sum_{i=1}^{n}\sigma_{P,i,h}^2(s) = 0$$
(24)

Plugging Eqn. 24 back to Eqn. 23, we have

$$\omega_h^2(s) = \sum_{s'}\mathbb{P}(s'|s)\omega_{h+1}^2(s') + \sum_{i=1}^{n}\sigma_{P,i,h}^2(s) + \frac{1}{m^2}\sum_{i=1}^{m}\sigma_{R,i}^2(s),$$

$\square$

*Proof.* (Proof of Corollary 1.1) We can regard the MDP with given policy $\pi$ as a Markov chain. By Theorem 1, we have

$$\omega_h^2(s) = \sum_{s'} \mathbb{P}(s'|s, \pi(s))\omega_{h+1}^2(s') + \sum_{i=1}^n \sigma_{P,i}^2(V_h^\pi, s, a) + \frac{1}{m^2}\sum_{i=1}^m \sigma_{R,i}^2(s, a)$$

By recursively decomposing the variance until step $H$, we have:

$$\omega_h^2(s_1) = \sum_{h=1}^H \sum_{(s,a)\in\mathcal{X}} w_h(s, a)\left(\sum_{i=1}^n \sigma_{P,i}^2(V_h^\pi, s, a) + \frac{1}{m^2}\sum_{i=1}^m \sigma_{R,i}^2(s, a)\right)$$

Since $\omega_h^2(s_1) = \mathbb{E}\left[(J_{h:H}(s_h) - V_h(s))^2|s_h = s\right] \leq H^2$, we can immediately reach the conclusion. $\square$

### F.3 The "good" Set Construction

The construction of the "good" set is similar with that in Dann et al. (2017) and Zanette & Brunskill (2019), though we modify it to handle this more complicated factored setting. The idea is to partition each factored state-action subspace at each episode into two sets, the set of state-action pairs that have been visited sufficiently often (so that we can lower bound these visits by their expectations using standard concentration inequalities) and the set of $(s, a)$ that were not visited often enough to cause high regret. That is:

**Definition 4.** *(The Good Set) The set $L_{k,i}$ for factored transition $\mathbb{P}_i$ is defined as:*

$$L_{k,i} \overset{def}{=} \left\{(x[Z_i^P]) \in \mathcal{X}[Z_i^P] : \frac{1}{4}\sum_{j<k} w_{j,Z_i^P}(x) \geq H\log(18nX_i^P H/\delta) + H\right\}$$

The following two Lemmas follow the same idea of Lemma 6 and Lemma 7 in Zanette & Brunskill (2019).

**Lemma F.2.** *Under event $\Lambda_1$ and $\Lambda_2$, if $(s, a)[Z_i^P] \in L_{i,k}$, we have*

$$N_k((s, a)[Z_i^P]) \geq \frac{1}{4}\sum_{j<k} w_{j,Z_i^P}(s, a).$$

*Proof.* By Lemma D.2, we have

$$N_k((s, a)[Z_i^P]) \geq \frac{1}{2}\sum_{j<k} w_j((s, a)[Z_i^P]) - H\log(18nX_i^P H/\delta).$$

Since $(s, a)[Z_i^P] \in L_{i,k}$, we have $\frac{1}{4}\sum_{j<k} w_{j,Z_i^P}(x) \geq H\log(18nX_i^P H/\delta) + H$. That is,

$$\begin{aligned} N_k((s, a)[Z_i^P]) &\geq \frac{1}{2}\sum_{j<k} w_j((s, a)[Z_i^P]) - H\log(18nX_i^P H/\delta) \\ &\geq \frac{1}{2}\sum_{j<k} w_j((s, a)[Z_i^P]) - \frac{1}{4}\sum_{j<k} w_j((s, a)[Z_i^P]) \\ &= \frac{1}{4}\sum_{j<k} w_j((s, a)[Z_i^P]) \end{aligned}$$

$\square$

**Lemma F.3.** *It holds that*

$$\sum_{k=1}^K \sum_{h=1}^H \sum_{(s,a)[Z_i^P]\notin L_{k,i}} w_{k,h,Z_i^P}(s, a) \leq 8HX_i^P \log(10nX_i^P H/\delta).$$

*Proof.* For those $(s,a)[Z_i^P] \notin L_{k,i}$, we have $\frac{1}{4} \sum_{j<k} w_{j,Z_i^P}(x) \leq H \log(10nX_i^P H/\delta) + H \leq 2H \log(10nX_i^P H/\delta)$. That is,

$$\sum_{k=1}^{K} \sum_{h=1}^{H} \sum_{(s,a)[Z_i^P] \notin L_{k,i}} w_{k,h,Z_i^P}(s,a) \leq \sum_{(s,a)[Z_i^P]} 8H \log(10nX_i^P H/\delta) \leq 8H X_i^P \log(10nX_i^P H/\delta)$$

$\square$

Lemma F.2 shows that we can lower bound the visiting count of a certain $(s,a)[Z_i^P]$ if the visiting probability of $(s,a)[Z_i^P]$ is sufficient large. Lemma F.3 shows that those $(s,a)[Z_i^P]$ with little visiting probability cause little contribution to the final regret.

**Lemma F.4.**

$$\sum_{k=1}^{K} \sum_{h=1}^{H} \sum_{(s,a)[Z_i^P] \in L_{k,i}} \frac{w_{k,h,Z_i^P}(s,a)}{N_k((s,a)[Z_i^P])} \leq 4X_i^P \log T.$$

*Proof.* For those $(s,a)[Z_i^P] \in L_{k,i}$, we have $N_k((s,a)[Z_i^P]) \geq \frac{1}{4} \sum_{j<k} w_{j,Z_i^P}(s,a)$. Therefore, we have

$$\sum_{k=1}^{K} \sum_{h=1}^{H} \sum_{(s,a)[Z_i^P] \in L_{k,i}} \frac{w_{k,h,Z_i^P}(s,a)}{N_k((s,a)[Z_i^P])} \leq \sum_{k=1}^{K} \sum_{h=1}^{H} \sum_{(s,a)[Z_i^P] \in L_{k,i}} \frac{4w_{k,h,Z_i^P}(s,a)}{\sum_{j<k} w_{j,Z_i^P}(s,a)}$$

$$\leq \sum_{(s,a)[Z_i^P] \in \mathcal{X}_i^P} \sum_{k=1}^{K} \frac{w_{k,Z_i^P}(s,a)}{\sum_{j<k} w_{j,Z_i^P}(s,a)}$$

$$\leq 4X_i^P \log T$$

$\square$

**Lemma F.5.** *For factored set $Z_i^P$ of transition, we have:*

$$\sum_{k=1}^{K} \sum_{h=1}^{H} \sum_{(s,a)\in\mathcal{X}} \frac{w_{k,h}(s,a)}{N_{k-1}((s,a)[Z_i^P])} \leq 8X_i^P \log T \tag{25}$$

$$\sum_{k=1}^{K} \sum_{h=1}^{H} \sum_{(s,a)\in\mathcal{X}} \frac{w_{k,h}(s,a)}{\sqrt{N_{k-1}((s,a)[Z_i^P])N_{k-1}((s,a)[Z_j^P])}} \leq 8X_i^P \log T \tag{26}$$

$$\sum_{k=1}^{K} \sum_{h=1}^{H} \sum_{(s,a)\in\mathcal{X}} \frac{w_{k,h}(s,a)}{\sqrt{N_{k-1}((s,a)[Z_i^P])} \left(N_{k-1}((s,a)[Z_j^P])\right)^{\frac{1}{4}}} \leq 8\sqrt{X_{i,j}^P} T^{1/4} \log T \tag{27}$$

*where $X_{i,j}^P = |X[Z_i^P \cup Z_j^P]|$.*

*For factored set $Z_i^R$ of rewards, similarly we have:*

$$\sum_{k=1}^{K} \sum_{h=1}^{H} \sum_{(s,a)\in\mathcal{X}} \frac{w_{k,h}(s,a)}{N_{k-1}((s,a)[Z_i^R])} \leq 8X_i^R \log T \tag{28}$$

$$\sum_{k=1}^{K} \sum_{h=1}^{H} \sum_{(s,a)\in\mathcal{X}} \frac{w_{k,h}(s,a)}{\sqrt{N_{k-1}((s,a)[Z_i^R])N_{k-1}((s,a)[Z_j^R])}} \leq 8X_i^R \log T \tag{29}$$

$$\sum_{k=1}^{K} \sum_{h=1}^{H} \sum_{(s,a)\in\mathcal{X}} \frac{w_{k,h}(s,a)}{\sqrt{N_{k-1}((s,a)[Z_i^R])} \left(N_{k-1}((s,a)[Z_j^R])\right)^{\frac{1}{4}}} \leq 8\sqrt{X_{i,j}^R} T^{1/4} \log T \tag{30}$$

*where $X_{i,j}^R = |X[Z_i^R \cup Z_j^R]|$.*

*Proof.* We only prove the inequalities for the factored set of transition. The inequalities for the factored set of rewards can be proved in the same manner.

For Inq. 25, we define $\mathcal{X}_i((s,a)[Z_i^P]) = \{x \in \mathcal{X} \mid x[Z_i^P] = (s,a)[Z_i^P]\}$, then we have

$$\sum_{k=1}^{K}\sum_{h=1}^{H}\sum_{(s,a)\in\mathcal{X}}\frac{w_{k,h}(s,a)}{N_{k-1}((s,a)[Z_i^P])}$$

$$=\sum_{k,h}\sum_{(s,a)[Z_i^P]\in\mathcal{X}[Z_i^P]}\frac{w_{k,h,Z_i^P}(s,a)\sum_{(s_1,a_1)\in\mathcal{X}_i((s,a)[Z_i^P])}\frac{w_{k,h}(s_1,a_1)}{w_{k,h,Z_i^P}(s,a)}}{N_{k-1}((s,a)[Z_i^P])}$$

$$=\sum_{k,h}\sum_{(s,a)[Z_i^P]\in\mathcal{X}[Z_i^P]}\frac{w_{k,h,Z_i^P}(s,a)}{N_{k-1}((s,a)[Z_i^P])}$$

$$=\sum_{k,h}\sum_{(s,a)[Z_i^P]\in L_{k,i}}\frac{w_{k,h,Z_i^P}(s,a)}{N_{k-1}((s,a)[Z_i^P])} + \sum_{k,h}\sum_{(s,a)[Z_i^P]\notin L_{k,i}}\frac{w_{k,h,Z_i^P}(s,a)}{N_{k-1}((s,a)[Z_i^P])}$$

$$\leq\sum_{k,h}\sum_{(s,a)[Z_i^P]\in L_{k,i}}\frac{w_{k,h,Z_i^P}(s,a)}{N_{k-1}((s,a)[Z_i^P])} + \sqrt{\sum_{k,h}\sum_{(s,a)[Z_i^P]\notin L_{k,i}}w_{k,h,Z_i^P}(s,a)\sum_{k,h}\sum_{(s,a)[Z_i^P]\notin L_{k,i}}\frac{1}{N_{k-1}((s,a)[Z_i^P])}}$$

$$\leq 4X_i^P\log T + \sqrt{8HX_i^P\log(10nX_i^P H/\delta)}$$

$$\leq 8X_i^P\log T$$

In the first equality, we firstly categorize $(s,a)$ based on their value $(s,a)[Z_i^P]$ and sum up over all possible choice of $(s,a)[Z_i^P]$, then we sum up the value in each category in the inner summation. The second equality is due to $\sum_{(s_1,a_1)\in\mathcal{X}_i((s,a)[Z_i^P])}\frac{w_{k,h}(s_1,a_1)}{w_{k,h,Z_i^P}(s,a)} = 1$. The first inequality is due to Cauchy-Schwarz inequality. The second inequality is due to Lemma F.4 and Lemma F.3. The last inequality is due to the assumption that $X_i^P \geq H\log(10nX_i^P H/\delta)$.

For Inq. 26 and Inq. 27, we define $Z_{i,j}^P = Z_i^P \cup Z_j^P$. For the factored set $Z_{i,j}^P$, similarly we have

$$\sum_{k=1}^{K}\sum_{h=1}^{H}\sum_{(s,a)[Z_{i,j}^P]\in L_{k,i}}\frac{w_{k,h,Z_{i,j}^P}(s,a)}{N_k((s,a)[Z_{i,j}^P])} \leq 4X_{i,j}^P\log T$$

$$\sum_{k=1}^{K}\sum_{h=1}^{H}\sum_{(s,a)[Z_{i,j}^P]\notin L_{k,i}}w_{k,h,Z_{i,j}^P}(s,a) \leq 8HX_{i,j}^P\log(10nX_{i,j}^P H/\delta),$$

By the definition of $Z_{i,j}^P$, we know that $N_{k-1}((s,a)[Z_i^P]) \geq N_{k-1}((s,a)[Z_{i,j}^P])$ and $N_{k-1}((s,a)[Z_j^P]) \geq N_{k-1}((s,a)[Z_{i,j}^P])$. Therefore, we have

$$\sum_{k=1}^{K}\sum_{h=1}^{H}\sum_{(s,a)\in\mathcal{X}}\frac{w_{k,h}(s,a)}{\sqrt{N_{k-1}((s,a)[Z_i^P])N_{k-1}((s,a)[Z_j^P])}} \leq \sum_{k=1}^{K}\sum_{h=1}^{H}\sum_{(s,a)\in\mathcal{X}}\frac{w_{k,h}(s,a)}{N_{k-1}((s,a)[Z_{i,j}^P])}$$

$$\sum_{k=1}^{K}\sum_{h=1}^{H}\sum_{(s,a)\in\mathcal{X}}\frac{w_{k,h}(s,a)}{\sqrt{N_{k-1}((s,a)[Z_i^P])}\left(N_{k-1}((s,a)[Z_j^P])\right)^{\frac{1}{4}}} \leq \sum_{k=1}^{K}\sum_{h=1}^{H}\sum_{(s,a)\in\mathcal{X}}\frac{w_{k,h}(s,a)}{\left(N_{k-1}((s,a)[Z_{i,j}^P])\right)^{\frac{3}{4}}}$$

The following proof of Inq. 26 and Inq. 27 shares the same idea of the proof of Inq. 25. □

### F.4 Technical Lemmas about Variance

In this subsection, we prove several technical lemmas about variance. For notation simplicity, we use $\mathbb{E}_i$ and $\mathbb{E}_{[i:j]}$ as a shorthand of $\mathbb{E}_{s'[i]\sim\mathbb{P}_i(\cdot|(s,a)[Z_i^P])}$ and $\mathbb{E}_{s'[i:j]\sim\mathbb{P}_{[i:j]}(\cdot|(s,a)[Z_{[i:j]}^P])}$. Similarly, we

use $\mathbb{V}_i$ and $\mathbb{V}_{[i:j]}$ as a shorthand of $\mathbb{V}_{s'[i]\sim\mathbb{P}_i(\cdot|(s,a)[Z_i^P])}$ and $\mathbb{V}_{s'[i:j]\sim\mathbb{P}_{[i:j]}(\cdot|(s,a)[Z_{[i:j]}^P])}$. For those w.r.t the empirical transition $\hat{\mathbb{P}}_k$, we use $\hat{\mathbb{E}}_k$ and $\hat{\mathbb{V}}_k$ to denote the corresponding expectation and variance. For example, $\mathbb{E}_{s'[i]\sim\hat{\mathbb{P}}_{k,i}(\cdot|(s,a)[Z_i^P])}$ is denoted as $\hat{\mathbb{E}}_{k,i}$.

**Lemma F.6.** *Under event $\Lambda_1$, $\Lambda_2$, we have:*

$$\left|\hat{\sigma}_{P,k,i}^2(V,s,a) - \sigma_{P,i}^2(V,s,a)\right| \leq 4H^2 \sum_{j=1}^n \left(2\sqrt{\frac{|\mathcal{S}_j|L^P}{N_{k-1}((s,a)[Z_j^P])}} + \frac{4|\mathcal{S}_j|L^P}{3N_{k-1}((s,a)[Z_j^P])}\right),$$

*where $V$ denotes some given function mapping from $\mathcal{S}$ to $\mathbb{R}$.*

*Proof.*

$$\begin{aligned}
|\hat{\sigma}_{P,k,i}^2(V,s,a) - \sigma_{P,i}^2(V,s,a)| =& |\hat{\mathbb{E}}_{[1:i-1]}\hat{\mathbb{V}}_i\hat{\mathbb{E}}_{[i+1:n]}V(s') - \mathbb{E}_{[1:i-1]}\mathbb{V}_i\mathbb{E}_{[i+1:n]}V(s')| & (31) \\
\leq& |\hat{\mathbb{E}}_{[1:i-1]}\hat{\mathbb{V}}_i\hat{\mathbb{E}}_{[i+1:n]}V(s') - \mathbb{E}_{[1:i-1]}\hat{\mathbb{V}}_i\hat{\mathbb{E}}_{[i+1:n]}V(s')| & (32) \\
& + |\mathbb{E}_{[1:i-1]}\hat{\mathbb{V}}_i\hat{\mathbb{E}}_{[i+1:n]}V(s') - \mathbb{E}_{[1:i-1]}\mathbb{V}_i\mathbb{E}_{[i+1:n]}V(s')| & (33)
\end{aligned}$$

We bound Equ. 32 and 33 separately.

For equ. 32, we have

$$\begin{aligned}
&|\hat{\mathbb{E}}_{[1:i-1]}\hat{\mathbb{V}}_i\hat{\mathbb{E}}_{[i+1:n]}V(s') - \mathbb{E}_{[1:i-1]}\hat{\mathbb{V}}_i\hat{\mathbb{E}}_{[i+1:n]}V(s')| \\
=& \left|\sum_{s'[1:i-1]\in\mathcal{S}[1:i-1]} \left(\hat{\mathbb{P}}_{[1:i-1]} - \mathbb{P}_{[1:i-1]}\right)(s'[1:i-1]|s,a)\hat{\mathbb{V}}_i\hat{\mathbb{E}}_{[i+1:n]}V(s')\right| \\
\leq& \left|\hat{\mathbb{P}}_{[1:i-1]}(\cdot|s,a) - \mathbb{P}_{[1:i-1]}(\cdot|s,a)\right|_1 \cdot \left|\hat{\mathbb{V}}_i\hat{\mathbb{E}}_{i+1:n}V(s')\right|_\infty \\
\leq& H^2 \sum_{j=1}^{i-1} \left|\hat{\mathbb{P}}_j(\cdot|s,a) - \mathbb{P}_j(\cdot|s,a)\right|_1 \\
\leq& H^2 \sum_{j=1}^{i-1} \left(2\sqrt{\frac{|\mathcal{S}_j|L^P}{N_{k-1}((s,a)[Z_j^P])}} + \frac{4|\mathcal{S}_j|L^P}{3N_{k-1}((s,a)[Z_j^P])}\right) & (34)
\end{aligned}$$

The last inequality is due to Lemma D.1.

For equ. 33, given fixed $s'[1:i-1]$, we have

$$\begin{aligned}
\left|\hat{\mathbb{V}}_i\hat{\mathbb{E}}_{[i+1:n]}V(s') - \mathbb{V}_i\mathbb{E}_{[i+1:n]}V(s')\right| \leq& \left|\hat{\mathbb{E}}_i\left(\hat{\mathbb{E}}_{[i+1:n]}V(s')\right)^2 - \mathbb{E}_i\left(\mathbb{E}_{[i+1:n]}V(s')\right)^2\right| \\
& + \left|\left(\mathbb{E}_{[i:n]}V(s')\right)^2 - \left(\hat{\mathbb{E}}_{[i:n]}V(s')\right)^2\right| \\
\leq& \left|\hat{\mathbb{E}}_i\left(\hat{\mathbb{E}}_{[i+1:n]}V(s')\right)^2 - \mathbb{E}_i\left(\hat{\mathbb{E}}_{[i+1:n]}V(s')\right)^2\right| \\
& + \left|\mathbb{E}_i\left(\hat{\mathbb{E}}_{[i+1:n]}V(s')\right)^2 - \mathbb{E}_i\left(\mathbb{E}_{[i+1:n]}V(s')\right)^2\right| \\
& + \left|\left(\mathbb{E}_{[i:n]}V(s')\right)^2 - \left(\hat{\mathbb{E}}_{[i:n]}V(s')\right)^2\right| \\
\leq& \left|\hat{\mathbb{E}}_i\left(\hat{\mathbb{E}}_{[i+1:n]}V(s')\right)^2 - \mathbb{E}_i\left(\hat{\mathbb{E}}_{[i+1:n]}V(s')\right)^2\right| & (35) \\
& + \left|\mathbb{E}_i\left(\hat{\mathbb{E}}_{[i+1:n]}V(s')\right)^2 - \mathbb{E}_i\left(\mathbb{E}_{[i+1:n]}V(s')\right)^2\right| & (36) \\
& + 2H\left|\mathbb{E}_{[i:n]}V(s') - \hat{\mathbb{E}}_{[i:n]}V(s')\right| & (37)
\end{aligned}$$

The first inequality is due to the definition of variance. The last inequality is due to $\mathbb{E}_{[i:n]}V(s') + \hat{\mathbb{E}}_{[i:n]}V(s') \leq 2H$.

All Equ. 35, 36 and 37 can be bounded with the same manner of Equ. 32. That is, we first bound each term with the $L_1$-distance of transition probability multiplying the $L_\infty$-norm of the value function, then we upper bound the $L_1$-distance by Lemma D.1. This leads to the following results:

$$\left|\hat{\mathbb{V}}_i\hat{\mathbb{E}}_{[i+1:n]}V(s') - \mathbb{V}_i\mathbb{E}_{[i+1:n]}V(s')\right| \leq 4H^2\sum_{j=i}^n\left(2\sqrt{\frac{|\mathcal{S}_j|L^P}{N_{k-1}((s,a)[Z_j^P])}} + \frac{4|\mathcal{S}_j|L^P}{3N_{k-1}((s,a)[Z_j^P])}\right)$$

This bound doesn't depend on the given fixed $s'[1:i-1]$. By taking expectation over $s'[1:i-1] \sim \mathbb{P}_{[1:i-1]}(\cdot|s,a)$, we have

$$\mathbb{E}_{[1:i-1]}\left|\hat{\mathbb{V}}_i\hat{\mathbb{E}}_{[i+1:n]}V(s') - \mathbb{V}_i\mathbb{E}_{[i+1:n]}V(s')\right|$$

$$\leq\left|\hat{\mathbb{E}}_i\left(\hat{\mathbb{E}}_{[i+1:n]}V(s')\right)^2 - \mathbb{E}_i\left(\mathbb{E}_{[i+1:n]}V(s')\right)^2\right| \leq 4H^2\sum_{j=i}^n\left(2\sqrt{\frac{|\mathcal{S}_j|L^P}{N_{k-1}((s,a)[Z_j^P])}} + \frac{4|\mathcal{S}_j|L^P}{3N_{k-1}((s,a)[Z_j^P])}\right)$$

Combining with Equ. 34, we have

$$\left|\hat{\sigma}_{P,k,i}^2(V,s,a) - \sigma_{P,i}^2(V,s,a)\right| \leq 4H^2\sum_{j=1}^n\left(2\sqrt{\frac{|\mathcal{S}_j|L^P}{N_{k-1}((s,a)[Z_j^P])}} + \frac{4|\mathcal{S}_j|L^P}{3N_{k-1}((s,a)[Z_j^P])}\right)$$

$\square$

**Lemma F.7.** *Under event $\Lambda_1$, $\Lambda_2$ and $\Omega$, we have*

$$\sigma_{P,i}^2(V_{h+1}^*,s,a) - 2\hat{\sigma}_{P,k,i}^2(\overline{V}_{k,h+1},s,a) \leq u_{k,h,i}(s,a) + 4H^2\sum_{j=1}^n\left(2\sqrt{\frac{|\mathcal{S}_j|L^P}{N_{k-1}((s,a)[Z_j^P])}} + \frac{4|\mathcal{S}_j|L^P}{3N_{k-1}((s,a)[Z_j^P])}\right),$$

*where $u_{k,h,i}(s,a)$ is defined in Section 4.3:*

$$u_{k,h,i}(s,a) = \mathbb{E}_{s'_{[1:i]}\sim\hat{\mathbb{P}}_{k,[1:i]}(\cdot|s,a)}\left[\left(\mathbb{E}_{s'_{[i+1:n]}\sim\hat{\mathbb{P}}_{k,[i+1:n]}(\cdot|s,a)}\left(\overline{V}_{k,h+1} - \underline{V}_{k,h+1}\right)(s')\right)^2\right].$$

*Proof.* We can decompose the difference in the following way:

$$\sigma_{P,i}^2(V_{h+1}^*,s,a) - 2\hat{\sigma}_{P,k,i}^2(\overline{V}_{k,h+1},s,a)$$
$$\leq\hat{\sigma}_{P,k,i}^2(V_{h+1}^*,s,a) - 2\hat{\sigma}_{P,k,i}^2(\overline{V}_{k,h+1},s,a) + \sigma_{P,i}^2(V_{h+1}^*,s,a) - \hat{\sigma}_{P,k,i}^2(V_{h+1}^*,s,a)$$

By Lemma F.6, we know that

$$\left|\hat{\sigma}_{P,k,i}^2(V_{h+1}^*,s,a) - \sigma_{P,i}^2(V_{h+1}^*,s,a)\right| \leq 4H^2\sum_{j=1}^n\left(2\sqrt{\frac{|\mathcal{S}_j|L^P}{N_{k-1}((s,a)[Z_j^P])}} + \frac{4|\mathcal{S}_j|L^P}{3N_{k-1}((s,a)[Z_j^P])}\right)$$

Now we only need to bound $\hat{\sigma}_{P,k,i}^2(V_{h+1}^*,s,a) - 2\hat{\sigma}_{P,k,i}^2(\overline{V}_{k,h+1},s,a)$. By Lemma 2 of Azar et al. (2017), we know that for two random variables $X \in \mathbb{R}$ and $Y \in \mathbb{R}$, we have

$$\mathbb{V}(X) \leq 2[\mathbb{V}(Y) + \mathbb{V}(X-Y)] \tag{38}$$

That is,

$$\hat{\sigma}_{P,k,i}^2(V_{h+1}^*,s,a) - 2\hat{\sigma}_{P,k,i}^2(\overline{V}_{k,h+1},s,a)$$
$$=\hat{\mathbb{E}}_{[1:i-1]}\hat{\mathbb{V}}_i\hat{\mathbb{E}}_{[i+1:n]}V_{h+1}^*(s') - 2\hat{\mathbb{E}}_{[1:i-1]}\hat{\mathbb{V}}_i\hat{\mathbb{E}}_{[i+1:n]}\overline{V}_{k,h+1}(s')$$
$$\leq2\hat{\mathbb{E}}_{[1:i-1]}\hat{\mathbb{V}}_i\left(\hat{\mathbb{E}}_{[i+1:n]}V_{h+1}^*(s') - \hat{\mathbb{E}}_{[i+1:n]}\overline{V}_{k,h+1}(s')\right)$$
$$=2\hat{\mathbb{E}}_{[1:i-1]}\hat{\mathbb{V}}_i\hat{\mathbb{E}}_{[i+1:n]}\left(V_{h+1}^*(s') - \overline{V}_{k,h+1}(s')\right)$$
$$\leq2\hat{\mathbb{E}}_{[1:i]}\left[\left(\hat{\mathbb{E}}_{[i+1:n]}\left(V_{h+1}^*(s') - \overline{V}_{k,h+1}(s')\right)\right)^2\right]$$
$$=u_{k,h,i}(s,a)$$

The first inequality is due to Inq. 38, and the second inequality is due to $\mathbb{V}X \leq \mathbb{E}X^2$ for any random variable $X \in \mathbb{R}$.

To sum up, we have

$$\sigma_{P,i}(V_{h+1}^*, s, a) - 2\hat{\sigma}_{P,k,i}(\overline{V}_{k,h+1}, s, a) \leq u_{k,h,i}(s, a) + 8H^2 \sum_{j=1}^{n} \left( 2\sqrt{\frac{|\mathcal{S}_j|L^P}{N_{k-1}((s,a)[Z_j^P])}} + \frac{4|\mathcal{S}_j|L^P}{3N_{k-1}((s,a)[Z_j^P])} \right)$$

$\square$

**Lemma F.8.** *Under event $\Lambda_1$, $\Lambda_2$ and $\Omega$, suppose $\tilde{\text{Reg}}(K) = \sum_{k=1}^{K} \overline{V}_{k,1}(s_1) - V_1^{\pi_k}(s_1)$, we have:*

$$\sum_{k=1}^{K} \sum_{h=1}^{H} \sum_{(s,a) \in \mathcal{X}} w_{k,h}(s,a) \left( \sigma_{P,i}^2(V_{h+1}^*, s, a) - \sigma_{P,i}^2(V_{h+1}^{\pi_k}, s, a) \right) \leq 2H^2 \tilde{\text{Reg}}(K)$$

$$\sum_{k=1}^{K} \sum_{h=1}^{H} \sum_{(s,a) \in \mathcal{X}} w_{k,h}(s,a) \left( \sigma_{P,i}^2(\overline{V}_{k,h+1}, s, a) - \sigma_{P,i}^2(V_{h+1}^{\pi_k}, s, a) \right) \leq 2H^2 \tilde{\text{Reg}}(K)$$

*Proof.* We only prove the first inequality in detail. By replacing $V_{h+1}^*$ with $\overline{V}_{k,h+1}$, we can prove the second inequality in the same manner.

$$\sigma_{P,i}^2(V_{h+1}^*, s, a) - \sigma_{P,i}^2(V_{h+1}^{\pi_k}, s, a) = \mathbb{E}_{[1:i]} \left[ \mathbb{V}_i \left( \mathbb{E}_{[i+1:n]} V_{h+1}^*(s') \right) - \mathbb{V}_i \left( \mathbb{E}_{[i+1:n]} V_{h+1}^{\pi_k}(s') \right) \right]$$

Given fixed $s'[1 : i - 1]$, we bound the difference of the variances: $\mathbb{V}_i \left( \mathbb{E}_{[i+1:n]} V_{h+1}^*(s') \right) - \mathbb{V}_i \left( \mathbb{E}_{[i+1:n]} V_{h+1}^{\pi_k}(s') \right)$.

$$\mathbb{V}_i \left( \mathbb{E}_{[i+1:n]} V_{h+1}^*(s') \right) - \mathbb{V}_i \left( \mathbb{E}_{[i+1:n]} V_{h+1}^{\pi_k}(s') \right)$$
$$= \mathbb{E}_i \left[ \left( \mathbb{E}_{[i+1:n]} V_{h+1}^*(s') \right)^2 - \left( \mathbb{E}_{[i+1:n]} V_{h+1}^{\pi_k}(s') \right)^2 \right] - \left( \mathbb{E}_{[i:n]} \left[ V_{h+1}^*(s') \right] \right)^2 + \left( \mathbb{E}_{[i:n]} \left[ V_{h+1}^{\pi_k}(s') \right] \right)^2$$
$$\leq \mathbb{E}_i \left[ \left( \mathbb{E}_{[i+1:n]} V_{h+1}^*(s') \right)^2 - \left( \mathbb{E}_{[i+1:n]} V_{h+1}^{\pi_k}(s') \right)^2 \right]$$
$$\leq 2H \mathbb{E}_i \left[ \mathbb{E}_{[i+1:n]} V_{h+1}^*(s') - \mathbb{E}_{[i+1:n]} V_{h+1}^{\pi_k}(s') \right]$$
$$= 2H \mathbb{E}_{[i:n]} \left[ V_{h+1}^*(s') - V_{h+1}^{\pi_k}(s') \right]$$

The first inequality is due to $V_{h+1}^*(s') \geq V_{h+1}^{\pi_k}(s')$, and the second inequality is due to $\mathbb{E}_{[i+1:n]} V_{h+1}^*(s') + \mathbb{E}_{[i+1:n]} V_{h+1}^{\pi_k}(s') \leq 2H$.

We then take expectation over all $s'[1 : i - 1]$. that is

$$\sigma_{P,i}^2(V_{h+1}^*, s, a) - \sigma_{P,i}^2(V_{h+1}^{\pi_k}, s, a) \leq 2H \mathbb{E}_{[1:n]} \left[ V_{h+1}^*(s') - V_{h+1}^{\pi_k}(s') \right]$$

Plugging the inequality into the former equation, we have

$$\sum_{k=1}^{K} \sum_{h=1}^{H} \sum_{(s,a) \in \mathcal{X}} w_{k,h}(s,a) \left( \sigma_{P,i}^2(V_{h+1}^*, s, a) - \sigma_{P,i}^2(V_{h+1}^{\pi_k}, s, a) \right)$$
$$\leq \sum_{k=1}^{K} \sum_{h=1}^{H} \sum_{(s,a) \in \mathcal{X}} w_{k,h}(s,a) 2H \mathbb{E}_{s' \sim \mathbb{P}(\cdot|s,a)} \left[ V_{h+1}^*(s') - V_{h+1}^{\pi_k}(s') \right]$$
$$= \sum_{k=1}^{K} \sum_{h=2}^{H} \sum_{s \in \mathcal{S}} 2w_{k,h}(s) H \left[ V_h^*(s) - V_h^{\pi_k}(s) \right]$$
$$\leq \sum_{k=1}^{K} 2H^2 \left[ V_1^*(s_1) - V_1^{\pi_k}(s_1) \right]$$
$$\leq \sum_{k=1}^{K} 2H^2 \left[ \overline{V}_{k,1}(s_1) - V_1^{\pi_k}(s_1) \right]$$
$$= 2H^2 \tilde{\text{Reg}}(K)$$

For the second inequality, this is because that by lemma E.15 of Dann et al. (2017), we have

$$\sum_s w_{k,h}(s)\left[V_h^*(s) - V_h^{\pi_k}(s)\right]$$

$$= \sum_{h_1=h}^H \sum_s w_{k,h_1}(s)\left(\bar{R}(s,\pi^*(s)) - \bar{R}(s,\pi_k(s)) + \mathbb{P}V_{h_1+1}^*(s,\pi^*(s)) - \mathbb{P}V_{h_1+1}^*(s,\pi^k(s))\right)$$

$$V_1^*(s_1) - V_1^{\pi_k}(s_1) = \sum_{h_1=1}^H \sum_s w_{k,h_1}(s)\left(\bar{R}(s,\pi^*(s)) - \bar{R}(s,\pi_k(s)) + \mathbb{P}V_{h_1+1}^*(s,\pi^*(s)) - \mathbb{P}V_{h_1+1}^*(s,\pi^k(s))\right)$$

This means that $\sum_s w_{k,h}(s)\left[V_h^*(s) - V_h^{\pi_k}(s)\right] \le V_1^*(s_1) - V_1^{\pi_k}(s_1)$ for any $k, h$. □

### F.5 OPTIMISM AND PESSIMISM

**Lemma F.9.** *Suppose that $\Lambda_1$, $\Lambda_2$ and $\Omega_{k,h+1}$ happen, then we have the following inequalities hold for any $a \in \mathcal{A}$ and $s \in \mathcal{S}$:*

$$\left|\hat{R}(s,a) - \bar{R}(s,a)\right| \le \frac{1}{m}\sum_{i=1}^m CB_i^R(s,a) \tag{39}$$

$$\left|\hat{\mathbb{P}}_k V_{h+1}^*(s,a) - \mathbb{P}V_{h+1}^*(s.a)\right| \le \sum_{i=1}^n CB_i^P(s,a) \tag{40}$$

*Proof.* The first inequality follows directly by Lemma F.1 and the definition of $CB_i^R(s,a)$. For the second inequality, by Lemma F.1, we have

$$\left|\left(\hat{\mathbb{P}}_k - \mathbb{P}\right)V_{h+1}^*(s,a)\right|$$

$$\le \sum_{i=1}^n \left(\sqrt{\frac{2\sigma_{P,i}^2(V_{h+1}^*,s,a)L^P}{N_{k-1}((s,a)[Z_i^P])}} + \frac{2HL^P}{3N_{k-1}((s,a)[Z_i^P])}\right)$$

$$+ \sum_{i=1}^n \sum_{j\ne i, j=1}^n H\phi_{k,i}(s,a)\phi_{k,j}(s,a),$$

where $\phi_{k,i}(s,a) = \sqrt{\frac{4|\mathcal{S}_i|L^P}{N_{k-1}((s,a)[Z_i^P])}} + \frac{4|\mathcal{S}_i|L^P}{3N_{k-1}((s,a)[Z_i^P])}$. The first inequality is due to $\overline{V}_{k,h}(s) \ge \overline{Q}_{k,h}(s,\pi^*(s))$.

By the definition of $CB_i^P(s,a)$, we have

$$\sum_i CB_i^P(s,a) - \left|\left(\hat{\mathbb{P}}_k - \mathbb{P}\right)V_{h+1}^*(s,a)\right| \tag{41}$$

$$\ge \sum_{i=1}^n \left(\sqrt{\frac{4\hat{\sigma}_{P,k,i}^2(\overline{V}_{k,h+1},s,a)L^P}{N_{k-1}((s,a^*)[Z_i^P])}} - \sqrt{\frac{2\sigma_{P,i}^2(V_{h+1}^*,s,a^*)L^P}{N_{k-1}((s,a)[Z_i^P])}}\right) \tag{42}$$

$$+ \sum_{i=1}^n \sqrt{\frac{2u_{k,h,i}(s,a)L^P}{N_{k-1}((s,a)[Z_i^P])}} + \sum_{i=1}^n \sqrt{\frac{16H^2L^P}{N_{k-1}((s,a)[Z_i^P])}} \sum_{j=1}^n \left(\left(\frac{4|\mathcal{S}_j|L^P}{N_{k-1}((s,a)[Z_j^P])}\right)^{\frac{1}{4}} + \sqrt{\frac{4|\mathcal{S}_j|L^P}{3N_{k-1}(s,a)[Z_j^P]}}\right) \tag{43}$$

We mainly focus on the bound of Eqn 42.

$$\sqrt{\frac{4\hat{\sigma}^2_{P,k,i}(\overline{V}_{k,h+1},s,a)L^P}{N_{k-1}((s,a)[Z^P_i])}} - \sqrt{\frac{2\sigma^2_{P,i}(V^*_{h+1},s,a)L^P}{N_{k-1}((s,a)[Z^P_i])}} \tag{44}$$

$$\geq \begin{cases} -\sqrt{\frac{2\sigma^2_{P,i}(V^*_{h+1},s,a)L^P - 4\hat{\sigma}^2_{P,k,i}(\overline{V}_{k,h+1},s,a)L^P}{N_{k-1}((s,a)[Z^P_i]))}} & 2\hat{\sigma}^2_{P,k,i}(\overline{V}_{k,h+1},s,a) \leq \sigma^2_{P,i}(V^*_{h+1},s,a) \\ 0 & \text{otherwise} \end{cases}$$

$$\tag{45}$$

For those $2\hat{\sigma}^2_{P,k,i}(\overline{V}_{k,h+1},s,a) \leq \sigma^2_{P,i}(V^*_{h+1},s,a)$, by Lemma F.7, we have

$$\sigma^2_{P,i}(V^*_{h+1},s,a) - 2\hat{\sigma}^2_{P,k,i}(\overline{V}_{k,h+1},s,a)$$
$$\leq u_{k,h,i}(s,a) + 8H^2 \sum_{j=1}^n \left( 2\sqrt{\frac{|\mathcal{S}_j|L^P}{N_{k-1}((s,a)[Z^P_j])}} + \frac{4|\mathcal{S}_i|L^P}{3N_{k-1}((s,a)[Z^P_j])} \right)$$

That is,

$$\sqrt{\frac{4\hat{\sigma}^2_{P,k,i}(\overline{V}_{k,h+1},s,a)L^P}{N_{k-1}((s,a)[Z^P_i])}} - \sqrt{\frac{2\sigma^2_{P,i}(V^*_{h+1},s,a)L^P}{N_{k-1}((s,a^*)[Z^P_i])}}$$

$$\geq -\sqrt{\frac{2u_{k,h,i}(s,a)L^P + 16H^2L^P\sum_{j=1}^n\left(2\sqrt{\frac{|\mathcal{S}_j|L^P}{N_{k-1}((s,a)[Z^P_i])}} + \frac{4|\mathcal{S}_i|L^P}{3N_{k-1}((s,a^*)[Z^P_i])}\right)}{N_{k-1}((s,a^*)[Z^P_i])}}$$

$$\geq -\sqrt{\frac{2u_{k,h,i}(s,a)L^P}{N_{k-1}((s,a)[Z^P_i])}} - \sqrt{\frac{16H^2L^P}{N_{k-1}((s,a)[Z^P_i])}}\sum_{j=1}^n\left(\left(\frac{4|\mathcal{S}_j|L^P}{N_{k-1}((s,a)[Z^P_j])}\right)^{\frac{1}{4}} + \sqrt{\frac{4|\mathcal{S}_j|L^P}{3N_{k-1}(s,a)[Z^P_j]}}\right)$$

Combining with Eq. 41, we prove that

$$\left| \hat{\mathbb{P}}_k V^*_{h+1}(s,a) - \mathbb{P}V^*_{h+1}(s.a) \right| \leq \sum_{i=1}^n CB^P_i(s,a)$$

$\square$

The optimism is proved by induction.

**Lemma F.10.** *(Optimism) Suppose that $\Lambda_1$, $\Lambda_2$ and $\Omega_{k,h+1}$ happen, then we have $\overline{V}_{k,h} \geq V^*_h$.*

*Proof.*

$$\overline{V}_{k,h}(s) - V^*_h(s)$$
$$= CB_k(s,\pi^*(s)) + \hat{\mathbb{P}}_k\overline{V}_{k,h+1}(s,\pi^*(s)) - \mathbb{P}V^*_{h+1}(s,\pi^*(s)) + \hat{R}_k(s,\pi^*(s)) - \bar{R}_k(s,\pi^*(s))$$
$$\geq CB_k(s,\pi^*(s)) + \left(\hat{\mathbb{P}}_k - \mathbb{P}\right)V^*_{h+1}(s,\pi^*(s)) + \hat{R}_k(s,\pi^*(s)) - \bar{R}_k(s,\pi^*(s))$$
$$\geq 0$$

The first inequality is due to induction condition that $\Omega_{k,h+1}$ happens. The last inequality is due to Lemma F.9.

$\square$

**Lemma F.11.** *(Pessimism) Suppose that $\Lambda_1$, $\Lambda_2$ and $\Omega_{k,h+1}$ happen, then we have $\underline{V}_{k,h} \leq V^*_h$.*

*Proof.*

$$\underline{V}_{k,h}(s)$$
$$=\hat{R}_k(s, \pi_{k,h}(s)) - CB_k(s, \pi_{k,h}(s)) + \hat{\mathbb{P}}_k\underline{V}_{k,h+1}(s, \pi_{k,h}(s))$$
$$\leq\hat{R}(s, \pi_{k,h}(s)) - CB_k(s, \pi_{k,h}(s)) + \hat{\mathbb{P}}_k V_{h+1}^*(s, \pi_{k,h}(s))$$
$$=\bar{R}(s, \pi_{k,h}(s)) + \mathbb{P}V_{h+1}^*(s, \pi_{k,h}(s))$$
$$+ \left( \hat{R}(s, \pi_{k,h}(s)) - \bar{R}(s, \pi_{k,h}(s)) - \frac{1}{m}\sum_{i=1}^{m} CB_i^R(s, \pi_{k,h}(s)) \right)$$
$$+ \left( \hat{\mathbb{P}}_k V_{h+1}^*(s, \pi_{k,h}(s)) - \mathbb{P}V_{h+1}^*(s, \pi_{k,h}(s)) - \sum_{i=1}^{n} CB_i^P(s, \pi_{k,h}(s)) \right)$$

The inequality is due to $\underline{V}_{k,h+1}(s') \leq V_{h+1}^*(s')$ since event $\Omega_{k,h+1}$ happens.

By lemma F.9, we have

$$\left| \hat{R}_k(s, \pi_{k,h}(s)) - \bar{R}(s, \pi_k(s)) \right| \leq \frac{1}{m}\sum_{i=1}^{m} CB_i^R(s, \pi_{k,h}(s))$$

$$\left| \hat{P}_k V_{h+1}^*(s, \pi_{k,h}(s)) - \mathbb{P}V_{h+1}^*(s.\pi_{k,h}(s)) \right| \leq \sum_{i=1}^{n} CB_i^P(s, \pi_{k,h}(s))$$

Therefore, we have

$$\underline{V}_{k,h}(s, \pi_{k,h}(s)) \leq \bar{R}(s, \pi_{k,h}(s)) + \mathbb{P}V_{h+1}^*(s, \pi_{k,h}(s))$$
$$\leq \bar{R}(s, \pi_h^*(s)) + \mathbb{P}V_{h+1}^*(s, \pi_h^*(s))$$
$$\leq V_h^*(s, a)$$

$\square$

**Lemma F.12.** *(Optimism and pessimism) Under event $\Lambda_1$ and $\Lambda_2$, we have $\Omega_{k,h}$ holds for all $k$ and $h$.*

*Proof.* By Lemma F.10 and Lemma F.11, through induction over all possible $k, h$, we can prove the Lemma. $\square$

## F.6 PROOF OF THEOREM 2

*Proof.* We decompose $\tilde{\text{Reg}}(K) = \sum_{k=1}^{K} \left( \overline{V}_{k,1}(s_{k,1}, a_{k,1}) - V_1^{\pi_k}(s_{k_1}, a_{k,1}) \right)$ in the classical way (Azar et al., 2017; Zanette & Brunskill, 2019; Dann et al., 2019), that is

$$\sum_{k=1}^{K} \left( \overline{V}_{k,1}(s_1) - V_1^{\pi_k}(s_1) \right) \tag{46}$$

$$\leq \sum_{k,h}\sum_{s_h,a_h} w_{k,h}(s_h, a_h)CB_k(s_h, a_h) \tag{47}$$

$$+ \sum_{k,h}\sum_{s_h,a_h} w_{k,h}(s_h, a_h) \left( \hat{\mathbb{P}}_k - \mathbb{P} \right) V_{h+1}^*(s_h, a_h) \tag{48}$$

$$+ \sum_{k,h}\sum_{s_h,a_h} w_{k,h}(s_h, a_h) \left( \hat{\mathbb{P}}_k - \mathbb{P} \right) \left( \overline{V}_{k,h+1} - V_{h+1}^* \right)(s_h, a_h) \tag{49}$$

$$+ \sum_{k,h}\sum_{s_h,a_h} w_{k,h}(s_h, a_h) \left( \hat{R}_k(s_h, a_h) - \bar{R}(s_h, a_h) \right) \tag{50}$$

We bound Equ. 47, 48, 49 and 50 separately by Lemma F.13, Lemma F.14, Lemma F.15 and Lemma F.16. Combining the results of these Lemmas, we have

$$
\tilde{\mathrm{Reg}}(K) \leq C_1 \frac{1}{m} \sum_{i=1}^{m} \sqrt{X_i^R T L_i^R \log T} + C_2 \sqrt{\sum_{i=1}^{n} H T X_i^P L^P \log T} + C_3 \sqrt{n H^2 \tilde{\mathrm{Reg}}(K) \sum_{i=1}^{n} X_i^P L^P \log T}
$$
(51)

Here $C_1, C_2, C_3$ denote some constants. Solving the $\tilde{\mathrm{Reg}}(K)$ in Inq 51, we can show that

$$
\tilde{\mathrm{Reg}}(K) \leq O \left( \frac{1}{m} \sum_{i=1}^{m} \sqrt{X_i^R T L_i^R \log T} + \sqrt{\sum_{i=1}^{n} H T X_i^P L^P \log T} \right),
$$

where $O$ hides the lower order terms w.r.t $T$.

By the optimism principle (Lemma F.12), we have $V_1^*(s_{k,1}, a_{k,1}) \leq \overline{V}_{k,1}(s_{k,1}, a_{k,1})$. This leads to the final result:

$$
\sum_{k=1}^{K} \left( V_1^*(s_{k,1}, a_{k,1}) - V_1^{\pi_k}(s_{k_1}, a_{k,1}) \right) \leq O \left( \frac{1}{m} \sum_{i=1}^{m} \sqrt{X_i^R T L_i^R \log T} + \sqrt{\sum_{i=1}^{n} H T X_i^P L^P \log T} \right).
$$

$\square$

### F.7 BOUNDING THE MAIN TERMS

**Lemma F.13.** *Under event $\Lambda_1$, $\Lambda_2$ and $\Omega_{k,h}$, suppose $\tilde{\mathrm{Reg}}(K) = \sum_{k=1}^{K} \overline{V}_{k,1}(s_1) - V_1^{\pi_k}(s_1)$, we have*

$$
\sum_{k=1}^{K} \sum_{h=1}^{H} \sum_{s,a} w_{k,h}(s,a) CB_k(s,a)
$$

$$
\leq O \left( \frac{1}{m} \sum_{i=1}^{m} \sqrt{T X_i^R L_i^R \log T} + \sqrt{H T \sum_{i=1}^{n} X_i^P L^P \log T} + \sqrt{H^2 n \tilde{\mathrm{Reg}}(K) \sum_{i=1}^{n} X_i^P L^P \log T} \right)
$$

*Proof.* By the definition of $CB_k(s,a)$, we have

$$
\sum_{k,h,s,a} w_{k,h}(s,a) CB_k(s,a)
$$
(52)

$$
= \sum_{k,h,s,a} w_{k,h}(s,a) \left( \frac{1}{m} \sum_{i=1}^{m} CB_{k,i}^R(s,a) + \sum_{i=1}^{n} CB_{k,i}^P(s,a) \right)
$$
(53)

$$
= \sum_{k,h,s,a} w_{k,h}(s,a) \left( \frac{1}{m} \sum_{i=1}^{m} \sqrt{\frac{2 \hat{\sigma}_{R,k,i}^2(s,a) L_i^R}{N_{k-1}((s,a)[Z_i^R])}} + \sum_{i=1}^{n} \sqrt{\frac{4 \hat{\sigma}_{P,k,i}^2(\overline{V}_{k,h+1}, s, a) L^P}{N_{k-1}((s,a)[Z_i^P])}} \right)
$$
(54)

$$
+ \sum_{k,h,s,a} w_{k,h}(s,a) \frac{1}{m} \sum_{i=1}^{m} \frac{8 L_i^R}{3 N_{k-1}((s,a)[Z_i^R])}
$$
(55)

$$
+ \sum_{k,h,s,a} w_{k,h}(s,a) \sum_{i=1}^{n} \left( \sqrt{\frac{16 H^2 L^P}{N_{k-1}((s,a)[Z_i^P])}} \sum_{j=1}^{n} \left( \left( \frac{4|\mathcal{S}_j| L^P}{N_{k-1}((s,a)[Z_j^P])} \right)^{\frac{1}{4}} + \sqrt{\frac{4|\mathcal{S}_j| L^P}{3 N_{k-1}(s,a)[Z_j^P]}} \right) \right)
$$
(56)

$$
+ \sum_{k,h,s,a} w_{k,h}(s,a) \sum_{i=1}^{n} \sum_{j \neq i, j=1}^{n} \frac{36 H |\mathcal{S}_i| |\mathcal{S}_j| (L^P)^2}{\sqrt{N_{k-1}((s,a)[Z_i^P]) N_{k-1}((s,a)[Z_j^P])}}
$$
(57)

$$
+ \sum_{k,h,s,a} w_{k,h}(s,a) \sum_{i=1}^{n} \sqrt{\frac{2 u_{k,h,i}(s,a) L^P}{N_{k-1}((s,a)[Z_i^P])}}
$$
(58)

By Lemma F.5, the upper bound of Eqn. 55, 56 and 57 is $O(T^{\frac{1}{4}})$, which doesn't contribute to the main factor in the regret. We prove the upper bound of Eqn. 54 and Eqn. 58 in detail.

By Lemma F.6, we have

$$
\left|\hat{\sigma}^2_{P,k,i}(\overline{V}_{k,h+1},s,a) - \sigma^2_{P,i}(\overline{V}_{k,h+1},s,a)\right| \leq 4H^2 \sum_{j=1}^{n} \left(2\sqrt{\frac{|\mathcal{S}_j|L^P}{N_{k-1}((s,a)[Z_j^P])}} + \frac{4|\mathcal{S}_j|L^P}{3N_{k-1}((s,a)[Z_j^P])}\right),
$$

Then Eqn. 54 can be bounded as

$$
\frac{1}{m}\sum_{i=1}^{m}\sqrt{\frac{2\hat{\sigma}^2_{R,k,i}(s,a)L_i^R}{N_{k-1}((s,a)[Z_i^R])}} + \sum_{i=1}^{n}\sqrt{\frac{4\hat{\sigma}^2_{P,k,i}(\overline{V}_{k,h+1},s,a)L^P}{N_{k-1}((s,a)[Z_i^P])}}
$$

$$
\leq \frac{1}{m}\sum_{i=1}^{m}\sqrt{\frac{2\hat{\sigma}^2_{R,k,i}(s,a)L_i^R}{N_{k-1}((s,a)[Z_i^R])}} + \sum_{i=1}^{n}\sqrt{\frac{4\sigma^2_{P,i}(\overline{V}_{k,h+1},s,a)L^P}{N_{k-1}((s,a)[Z_i^P])}}
$$

$$
+ \sum_{i=1}^{n}\sqrt{\frac{4|\sigma^2_{P,i}(\overline{V}_{k,h+1},s,a) - \hat{\sigma}^2_{P,k,i}(\overline{V}_{k,h+1},s,a)|L^P}{N_{k-1}((s,a)[Z_i^P])}}
$$

$$
\leq \frac{1}{m}\sum_{i=1}^{m}\sqrt{\frac{2\hat{\sigma}^2_{R,k,i}(s,a)L_i^R}{N_{k-1}((s,a)[Z_i^R])}} \tag{59}
$$

$$
+ \sum_{i=1}^{n}\sqrt{\frac{4\sigma^2_{P,i}(\overline{V}_{k,h+1},s,a)L^P}{N_{k-1}((s,a)[Z_i^P])}} \tag{60}
$$

$$
+ 8H\sum_{i=1}^{n}\sqrt{\frac{L^P}{N_{k-1}((s,a)[Z_i^P])}}\sum_{j=1}^{n}\left(\left(\frac{|\mathcal{S}_j|L^P}{N_{k-1}((s,a)[Z_j^P])}\right)^{\frac{1}{4}} + \sqrt{\frac{4|\mathcal{S}_j|L^P}{3N_{k-1}((s,a)[Z_j^P])}}\right) \tag{61}
$$

Similar with Eqn. 61, the summation of Eqn. 61 is upper bounded by $O(T^{1/4})$ by Lemma F.5. For Eqn. 59, we have

$$
\sum_{k,h}\sum_{s,a}w_{k,h}(s,a)\frac{1}{m}\sum_{i=1}^{m}\sqrt{\frac{2\hat{\sigma}^2_{R,k,i}(s,a)L_i^R}{N_{k-1}((s,a)[Z_i^R])}}
$$

$$
\leq \sum_{k,h}\sum_{s,a}w_{k,h}(s,a)\frac{1}{m}\sum_{i=1}^{m}\sqrt{\frac{2L_i^R}{N_{k-1}((s,a)[Z_i^R])}}
$$

$$
\leq \frac{1}{m}\sum_{i=1}^{m}\sqrt{\sum_{k,h}\sum_{s,a}w_{k,h}(s,a)}\sqrt{\sum_{k,h}\sum_{s,a}\frac{2w_{k,h}(s,a)L_i^R}{N_{k-1}((s,a)[Z_i^R])}}
$$

$$
\leq \frac{1}{m}\sum_{i=1}^{m}\sqrt{T}\sqrt{\sum_{k,h}\sum_{s,a}\frac{2w_{k,h}(s,a)L_i^R}{N_{k-1}((s,a)[Z_i^R])}}
$$

The first inequality is due to $\hat{\sigma}^2_{R,k,i}(s,a) \leq 1$. The second inequality is due to Cauchy-Schwarz inequality. By Lemma F.5, the summation can be bounded by $\frac{1}{m}\sum_{i=1}^{m}\sqrt{X_i^R L_i^R T \log T}$.

For Eqn. 60, we have

$$\sum_{k,h}\sum_{s,a} w_{k,h}(s,a)\sum_{i=1}^{n}\sqrt{\frac{4\sigma_{P,i}^2(\overline{V}_{k,h+1},s,a)L^P}{N_{k-1}((s,a)[Z_i^P])}}$$

$$\leq\sqrt{\sum_{k,h,s,a} w_{k,h}(s,a)\sum_{i=1}^{n}\sigma_{P,i}^2(\overline{V}_{k,h+1},s,a)}\cdot\sqrt{\sum_{k,h,s,a} w_{k,h}(s,a)\sum_{i=1}^{n}\frac{4L^P}{N_{k-1}((s,a)[Z_i^P])}}$$

$$\leq\sqrt{\sum_{k,h,s,a} w_{k,h}(s,a)\sum_{i=1}^{n}\sigma_{P,i}^2(\overline{V}_{k,h+1},s,a)}\sqrt{\sum_{i=1}^{n}4X_i^P L^P\log T}$$

$$=\sqrt{\sum_{k,h,s,a} w_{k,h}(s,a)\sum_{i=1}^{n}\sigma_{P,i}^2(V_{h+1}^{\pi_k},s,a)}\sqrt{\sum_{i=1}^{n}4X_i^P L^P\log T}$$

$$+\sqrt{\sum_{k,h,s,a} w_{k,h}(s,a)\sum_{i=1}^{n}\left(\sigma_{P,i}^2(\overline{V}_{k,h+1},s,a)-\sigma_{P,i}^2(V_{h+1}^{\pi_k},s,a)\right)}\sqrt{\sum_{i=1}^{n}4X_i^P L^P\log T}$$

$$\leq\sqrt{\sum_{k,h,s,a} w_{k,h}(s,a)\sum_{i=1}^{n}\sigma_{P,i}^2(V_{h+1}^{\pi_k},s,a)}\sqrt{\sum_{i=1}^{n}4X_i^P L^P}+\sqrt{2H^2 n\tilde{\text{Reg}}(K)\sum_{i=1}^{n}X_i^P L^P\log T}$$

$$\leq\sqrt{\sum_{k,h,s,a} w_{k,h}(s,a)\left(\frac{1}{m^2}\sum_{i=1}^{m}\sigma_{R,i}^2(s,a)+\sum_{i=1}^{n}\sigma_{P,i}^2(V_{h+1}^{\pi_k},s,a)\right)}\sqrt{\sum_{i=1}^{n}4X_i^P L^P\log T}+\sqrt{2H^2 n\tilde{\text{Reg}}(K)\sum_{i=1}^{n}X_i^P L^P\log T}$$

$$\leq\sqrt{HT}\cdot\sqrt{\sum_{i=1}^{n}4X_i^P L^P\log T}+\sqrt{2H^2 n\tilde{\text{Reg}}(K)\sum_{i=1}^{n}X_i^P L^P\log T}$$

The first inequality is due to Cauchy-Schwarz inequality. The second inequality is due to Lemma F.5. The third inequality is due to Lemma F.8. The forth inequality is due to $\sigma_{R,k,i}^2(s,a)\geq 0$, and the last inequality is because of Corollary 1.1.

For Eqn. 58, we have

$$\sum_{k,h,s,a} w_{k,h}(s,a)\sum_{i=1}^{n}\sqrt{\frac{2u_{k,h,i}(s,a)L^P}{N_{k-1}((s,a)[Z_i^P])}}$$

$$\leq\sum_{i=1}^{n}\sqrt{2L^P\left(\sum_{k,h,s,a}\frac{4w_{k,h}(s,a)}{N_{k-1}((s,a)[Z_i^P])}\right)\left(\sum_{k,h,s,a} w_{k,h}(s,a)u_{k,h,i}(s,a)\right)}$$

$$\leq\sum_{i=1}^{n}\sqrt{64X_i^P L^P\log T\left(\sum_{k,h,s,a} w_{k,h}(s,a)u_{k,h,i}(s,a)\right)}\tag{62}$$

By Lemma F.20, we know that the summation $\sum_{k,h,s,a} w_{k,h}(s,a)u_{k,h,i}(s,a)$ is of order $O(T^{\frac{1}{2}})$. This means that Equ. 62 is of order $O(T^{\frac{1}{4}})$, which doesn't contribute to the main term ($O(\sqrt{T})$).

$\square$

**Lemma F.14.** *Under event* $\Lambda_1$, $\Lambda_2$ *and* $\Omega$, *suppose* $\tilde{\mathrm{Reg}}(K) = \sum_{k=1}^{K} \overline{V}_{k,1}(s_1) - V_1^{\pi_k}(s_1)$, *we have*

$$\sum_{k=1}^{K} \sum_{h=1}^{H} \sum_{(s,a) \in \mathcal{X}} w_{k,h}(s,a) \left( \hat{\mathbb{P}}_k - \mathbb{P} \right) V^*(s,a)$$

$$\leq O \left( \sqrt{ (HT + nH^2 \tilde{\mathrm{Reg}}(K)) \sum_{i=1}^{n} X_i^P L^P } \right)$$

*Proof.* By Lemma F.1, we have

$$\sum_{k=1}^{k_1} \sum_{h=1}^{H} \sum_{(s,a) \in \mathcal{X}} w_{k,h}(s,a) \left( \hat{\mathbb{P}}_k - \mathbb{P} \right) V^*(s,a)$$

$$\leq \sum_{k} \sum_{h} \sum_{i=1}^{n} \sum_{(s,a) \in \mathcal{X}} w_{k,h}(s,a) \sqrt{ \frac{2\sigma_{P,i}^2(V_{h+1}^*, s, a) L^P}{N_{k-1}((s,a)[Z_i^P])} }$$

$$+ \sum_{k} \sum_{h} \sum_{i=1}^{n} \sum_{(s,a) \in \mathcal{X}} w_{k,h}(s,a) \sum_{j \neq i, j=1}^{n} \frac{36 H |\mathcal{S}_i||\mathcal{S}_j|(L^P)^2}{\sqrt{N_{k-1}((s,a)[Z_i^P]) N_{k-1}((s,a)[Z_j^P])}}$$

By Lemma F.5, the second term has only logarithmic dependence on $T$, which is negligible compared with the main factor. We mainly focus on the first term.

$$\sum_{k} \sum_{h} \sum_{i=1}^{n} \sum_{(s,a) \in \mathcal{X}} w_{k,h}(s,a) \sqrt{ \frac{2\sigma_{P,i}^2(V_{h+1}^*, s, a) L^P}{N_{k-1}((s,a)[Z_i^P])} }$$

$$\leq \sqrt{ \sum_{k,h} \sum_{(s,a) \in \mathcal{X}} 2 w_{k,h}(s,a) \sum_{i=1}^{n} \sigma_{P,i}^2(V_{h+1}^*, s, a) } \cdot \sqrt{ \sum_{k,h} \sum_{(s,a) \in \mathcal{X}} \sum_{i=1}^{n} \frac{w_{k,h}(s,a) L^P}{N_{k-1}((s, \pi(s))[Z_i^P])} }$$

$$\leq \sqrt{ \sum_{k,h} \sum_{(s,a) \in \mathcal{X}} 2 w_{k,h}(s,a) \left( \frac{1}{m} \sum_{i=1}^{m} \sigma_{R,i}^2(s, a) + \sum_{i=1}^{n} \sigma_{P,i}^2(V_{h+1}^*, s, a) \right) } \cdot \sqrt{ \sum_{k,h} \sum_{(s,a) \in \mathcal{X}} \sum_{i=1}^{n} \frac{w_{k,h}(s,a) L^P}{N_{k-1}((s, \pi(s))[Z_i^P])} }$$

$$\leq \sqrt{ \left( HT + nH^2 \tilde{\mathrm{Reg}}(K) \right) } \sqrt{ 8 \sum_{i=1}^{n} X_i^P L^P \log T }$$

The first inequality is due to Cauchy-Schwarz inequality. The second inequality is due to $\sigma_{R,k,i}^2(s,a) \geq 0$. For the last inequality, the first part is the summation of the variance, which can be bounded by Lemma 1.1 and Lemma F.8, while the second part can be bounded as $\sum_i X_i^P L^P \log T$ by Lemma F.5. $\qquad \square$

**Lemma F.15.** *Under event* $\Lambda_1$, $\Lambda_2$ *and* $\Omega$, *suppose* $\tilde{\mathrm{Reg}}(K) = \sum_{k=1}^{K} \overline{V}_{k,1}(s_1) - V_1^{\pi_k}(s_1)$, *we have*

$$\sum_{k=1}^{K} \sum_{h=1}^{H} \sum_{(s,a) \in \mathcal{X}} w_{k,h}(s,a) \left( \hat{\mathbb{P}}_k - \mathbb{P} \right) \left( \overline{V}_{k,h+1} - V_{h+1}^* \right)(s,a) \leq O \left( H \sqrt{ \sum_{i=1}^{n} \sqrt{ T X_j^P |\mathcal{S}_j| L^P } } \sum_{i=1}^{n} 2 \sqrt{ 8 X_i^P L^P \log T } \right)$$

*Proof.* By Lemma E.1, we can prove that

$$\left(\hat{\mathbb{P}}_k - \mathbb{P}\right)\left(\overline{V}_{k,h+1} - V_{h+1}^*\right)(s,a)$$

$$\leq \sum_{i=1}^{n}(\hat{\mathbb{P}}_{k,i} - \mathbb{P}_i)\mathbb{P}_{[1:i-1]}\mathbb{P}_{[i+1:n]}\left(\overline{V}_{k,h}(s_{k,h}, a_{k,h}) - V_h^*(s_{k,h}, a_{k,h})\right)$$

$$+ \sum_{i=1}^{n}\sum_{j=1}^{n} H\left|\left(\hat{\mathbb{P}}_{k,i} - \mathbb{P}_i\right)(\cdot|(s,a)[Z_i^P])\right|_1\left|\left(\hat{\mathbb{P}}_{k,j} - \mathbb{P}_j\right)(\cdot|(s,a)[Z_j^P])\right|_1$$

$$\leq \sum_{i=1}^{n}\left(2\sum_{s'[i]\in\mathcal{S}_i}\sqrt{\frac{\mathbb{P}_i(s'[i]|\mathcal{X}[Z_i^P])L^P}{N_{k-1}((s,a)[Z_i^P])}}\right)\mathbb{P}_{[1:i-1]}\mathbb{P}_{[i+1:n]}\left(\overline{V}_{k,h}(s_{k,h}, a_{k,h}) - V_h^*(s_{k,h}, a_{k,h})\right)$$

$$+ \sum_{i=1}^{n}\sum_{j\neq i, j=1}^{n} 36H\frac{|\mathcal{S}_i||\mathcal{S}_j|(L^P)^2}{\sqrt{N_{k-1}((s,a)[Z_i^P])N_{k-1}((s,a)[Z_j^P])}} + \sum_{i=1}^{n}\frac{|\mathcal{S}_i|L^P}{3N_{k-1}((s,a)[Z_i^P])}$$

The second inequality is due to Lemma D.1.

We only focus on the summation of the first term, since the summation of other terms has only logarithmic dependence on $T$ by Lemma F.5.

$$\sum_{k,h}\sum_{s,a} w_{k,h}(s,a)\sum_{i=1}^{n}\left(2\sum_{s'[i]\in\mathcal{S}_i}\sqrt{\frac{\mathbb{P}_i(s'[i]|(s,a)[Z_i^P])L^P}{N_{k-1}((s,a)[Z_i^P])}}\right)\mathbb{P}_{[1:i-1]}\mathbb{P}_{i+1:n}\left(\overline{V}_{k,h}(s_{k,h}, a_{k,h}) - V_h^*(s_{k,h}, a_{k,h})\right)$$

$$= \sum_{i=1}^{n}\sum_{k,h}\sum_{s,a} w_{k,h}(s,a)\mathbb{P}_{1:i-1}\left(2\sum_{s'[i]\in\mathcal{S}_i}\sqrt{\frac{\mathbb{P}_i(s'[i]|\mathcal{X}[Z_i^P])L^P\left(\mathbb{P}_{[i+1:n]}\left(\overline{V}_{k,h}(s_{k,h}, a_{k,h}) - V_h^*(s_{k,h}, a_{k,h})\right)\right)^2}{N_{k-1}((s,a)[Z_i^P])}}\right)$$

$$\leq \sum_{i=1}^{n}\sum_{k,h}\mathbb{P}_{1:i-1}\sum_{s,a} w_{k,h}(s,a)\left(2\sum_{s'[i]\in\mathcal{S}_i}\sqrt{\frac{\mathbb{P}_i(s'[i]|(s,a)[Z_i^P])L^P\left(\mathbb{P}_{[i+1:n]}\left(\overline{V}_{k,h}(s_{k,h}, a_{k,h}) - \underline{V}_{k,h}(s_{k,h}, a_{k,h})\right)\right)^2}{N_{k-1}((s,a)[Z_i^P])}}\right)$$

$$\leq \sum_{i=1}^{n}\sum_{k,h}\sum_{s,a} w_{k,h}(s,a)\left(2\sqrt{\frac{|\mathcal{S}_i|L^P\mathbb{E}_{[1:i]}\left(\mathbb{E}_{[i+1:n]}\left(\overline{V}_{k,h}(s') - \underline{V}_{k,h}(s')\right)\right)^2}{N_{k-1}((s,a)[Z_i^P])}}\right)$$

$$\leq \sum_{i=1}^{n} 2\sqrt{\left(\sum_{k,h,s,a}\frac{w_{k,h}(s,a)}{N_{k-1}((s,a)[Z_i^P])}\right)\left(\sum_{k,h,s,a} w_{k,h}(s,a)\mathbb{E}_{[1:i]}\left(\mathbb{E}_{[i+1:n]}\left(\overline{V}_{k,h+1}(s') - \underline{V}_{k,h+1}(s')\right)\right)^2\right)}$$

$$\leq \sum_{i=1}^{n} 2\sqrt{8X_i^P L^P \log T}\sqrt{H^2\sum_{i=1}^{n}\sqrt{TX_j^P|\mathcal{S}_j|L^P}}$$

The first inequality is due to the fact that $\overline{V}_{k,h} \geq V_h^* \geq \underline{V}_{k,h}$. The second and the third inequality is due to Cauchy-Schwarz inequality. The last inequality is because of Lemma F.5 and Lemma F.20. □

**Lemma F.16.** *Under event $\Lambda_1$, $\Lambda_2$ and $\Omega$, we have*

$$\sum_{k,h}\sum_{s_h,a_h} w_{k,h}(s_h,a_h)\left(\hat{R}_k(s_h,a_h) - \bar{R}(s_h,a_h)\right) \leq O\left(\frac{1}{m}\sum_{i=1}^{m}\sqrt{TX_i^R L_i^R \log T}\right)$$

*Proof.* By Lemma F.1, we have

$$\sum_{k,h}\sum_{s_h,a_h} w_{k,h}(s_h,a_h)\left(\hat{R}_k(s_h,a_h)-\bar{R}(s_h,a_h)\right)$$

$$\leq\frac{1}{m}\sum_{i=1}^{n}\sum_{k,h}\sum_{s_h,a_h} w_{k,h}(s_h,a_h)\left(\sqrt{\frac{2\hat{\sigma}_{R,k,i}(s_h,a_h)L_i^R}{N_{k-1}((s_h,a_h)[Z_i^R])}}+\frac{8L_i^R}{3N_{k-1}((s_h,a_h)[Z_i^R])}\right)$$

$$\leq\frac{1}{m}\sum_{i=1}^{m}\sum_{k,h}\sum_{s_h,a_h} w_{k,h}(s_h,a_h)\left(\sqrt{\frac{2L_i^R}{N_{k-1}((s_h,a_h)[Z_i^R])}}+\frac{8L_i^R}{3N_{k-1}((s_h,a_h)[Z_i^R])}\right)$$

$$\leq\frac{1}{m}\sum_{i=1}^{m}\sqrt{\frac{\sum_{k,h}\sum_{s_h,a_h}2w_{k,h}(s_h,a_h)L_i^R}{N_{k-1}((s_h,a_h)[Z_i^R])}}+\frac{1}{m}\sum_{i=1}^{m}\sum_{k,h}\sum_{s_h,a_h} w_{k,h}(s_h,a_h)\frac{8L_i^R}{3N_{k-1}((s_h,a_h)[Z_i^R])}$$

The second inequality is due to $\hat{\sigma}_{R,k,i}(s,a)\leq 1$. The last inequality is due to Cauchy-Schwarz inequality. By Lemma F.5, we know that the summation is of order $O\left(\frac{1}{m}\sum_{i=1}^{m}\sqrt{TX_i^R L_i^R\log T}\right)$.
$\square$

**Lemma F.17.** *Under event $\Lambda_1$ and $\Lambda_2$, we have*

$$\left(\overline{V}_{k,h}-\underline{V}_{k,h}\right)(s)\leq\mathbb{E}_{traj_k}\left[\sum_{i=h}^{H}\left(2CB_k(s,\pi_{k,i}(s))+\sum_{j=1}^{n}\sqrt{\frac{2H^2\log(18nT|\mathcal{X}[Z_j^P]|/\delta)}{N_{k-1}((s,\pi_{k,j}(s))[Z_j^P])}}\right)\mid s_h=s,\pi_k\right]$$

*The expectation is over all possible trajectories in episode $k$ given $s_h=s$ following policy $\pi_k$.*

*Proof.*
$$\left(\overline{V}_{k,h}-\underline{V}_{k,h}\right)(s)$$
$$=2CB_k(s,\pi_{k,h}(s))+\hat{\mathbb{P}}_k(\overline{V}_{k,h+1}(s)-\underline{V}_{k,h+1}(s))$$
$$=2CB_k(s,\pi_{k,h}(s))+(\hat{\mathbb{P}}_k-\mathbb{P})(\overline{V}_{k,h+1}-\underline{V}_{k,h+1})(s,\pi_{k,h}(s))+\mathbb{P}(\overline{V}_{k,h+1}-\underline{V}_{k,h+1})(s,\pi_{k,h}(s))$$
$$\cdots$$
$$=\mathbb{E}_{traj_k}\left[\sum_{i=h}^{H}\left(2CB_k(s,\pi_{k,i}(s_i))+(\hat{\mathbb{P}}_k-\mathbb{P})(\overline{V}_{k,i+1}-\underline{V}_{k,i+1})(s_i,\pi_{k,i}(s_i))\right)\mid s_h=s,\pi_k\right]$$

The second term can be bounded as:
$$(\hat{\mathbb{P}}_k-\mathbb{P})(\overline{V}_{k,i+1}-\underline{V}_{k,i+1})(s_i,\pi_{k,i}(s_i))$$
$$\leq|\hat{\mathbb{P}}_k-\mathbb{P}|_1|\overline{V}_{k,i+1}-\underline{V}_{k,i+1}|_\infty(s_i,\pi_{k,i}(s))$$
$$\leq H\sum_{i=1}^{n}|\hat{\mathbb{P}}_{k,i}-\mathbb{P}_i|_1(\cdot|s_i,\pi_{k,i}(s_i))$$
$$\leq\sum_{j=1}^{n}\sqrt{\frac{2H^2L^P}{N_{k-1}((s,\pi_{k,j}(s))[Z_j^P])}}$$

$\square$

**Lemma F.18.** *Under event $\Lambda_1$ and $\Lambda_2$, we have*

$$\sum_{k=1}^{K}\sum_{h=1}^{H}\sum_{(s,a)\in\mathcal{X}} w_{k,h}(s,a)CB_k^2(s,a)$$

$$\leq\sum_{i=1}^{m}\frac{2(m+2)H^2L_i^R X_i^R\log T}{m^2}+\sum_{i=1}^{n}128n(m+n)H^2X_i^P L^P\sum_{j=1}^{n}|\mathcal{S}_j|L^P\log T,$$

*Which has only logarithmic dependence on $T$.*

Note that this bound is loose w.r.t parameters such as $H$, $|\mathcal{S}_j|$, $X_i^P$ and $X_i^R$. However, it is acceptable since we regard $T$ as the dominant parameter. This bound doesn't influence the dominant factor in the final regret.

*Proof.* By the definition of $CB_k(s,a)$, $CB_{k,i}^R(s,a)$ and $CB_{k,i}^P(s,a)$, we have

$$
\begin{aligned}
CB_k^2(s,a) \leq & (m+n)\left(\sum_{i=1}^m \frac{1}{m^2}\left(CB_{k,i}^R(s,a)\right)^2 + \sum_{i=1}^n \left(CB_{k,i}^P(s,a)\right)^2\right)\\
\leq & 2(m+n)\sum_{i=1}^m \frac{1}{m^2}\left(\frac{2H^2 L_i^R}{N_{k-1}((s,a)[Z_i^R])} + \frac{64(L_i^R)^2}{9(N_{k-1}((s,a)[Z_i^R]))^2}\right)\\
& + 4n(m+n)\sum_{i=1}^n \left(\frac{4H^2 L^P}{N_{k-1}((s,a)[Z_i^P])} + \frac{2HL^P}{N_{k-1}((s,a)[Z_i^P])}\right)\\
& + 4n(m+n)\sum_{i=1}^n \left(\frac{32H^2 L^P}{N_{k-1}((s,a)[Z_i^P])}\sum_{j=1}^n \left(\sqrt{\frac{4|\mathcal{S}_j|L^P}{N_{k-1}((s,a)[Z_j^P])}} + \frac{4|\mathcal{S}_j|L^P}{3N_{k-1}((s,a)[Z_j^P])}\right)\right)
\end{aligned}
$$

The second inequality is due to $\hat{\sigma}_{R,i}^2(s,a) \leq 1$, $\hat{\sigma}_{P,i}^2(\overline{V}_{k,h+1},s,a) \leq H^2$ and $u_{k,h,i}(s,a) \leq H$.

Now we are ready to bound $\sum_{k=1}^K \sum_{h=1}^H \sum_{(s,a)\in\mathcal{X}} w_{k,h}(s,a)CB_k^2(s,a)$:

$$
\begin{aligned}
& \sum_{k=1}^K \sum_{h=1}^H \sum_{(s,a)\in\mathcal{X}} w_{k,h}(s,a)CB_k^2(s,a)\\
\leq & \sum_{k=1}^K \sum_{h=1}^H \sum_{(s,a)\in\mathcal{X}} w_{k,h}(s,a)\left(\sum_{i=1}^m \frac{2(m+2)H^2 L_i^R}{m^2 N_{k-1}((s,a)[Z_i^R])} + \sum_{i=1}^n \frac{128n(m+n)H^2 L^P \sum_{j=1}^n |\mathcal{S}_j|L^P}{N_{k-1}((s,a)[Z_i^P])}\right)\\
\leq & \sum_{i=1}^m \frac{2(m+2)H^2 L_i^R X_i^R \log T}{m^2} + \sum_{i=1}^n 128n(m+n)H^2 X_i^P L^P \sum_{j=1}^n |\mathcal{S}_j|L^P \log T
\end{aligned}
$$

The last inequality is due to Lemma F.5. $\qquad\square$

**Lemma F.19.** *Under event $\Lambda_1$ and $\Lambda_2$, we have*

$$
\sum_{k=1}^K \sum_{h=1}^H \sum_{(s,a)\in\mathcal{X}} w_{k,h}(s,a)\mathbb{E}_{s'[1:i]\sim\mathbb{P}_{[1:i]}(\cdot|s,a)}\left(\mathbb{E}_{s'[i+1:n]\sim\mathbb{P}_{[i+1:n]}(\cdot|s,a)}\left(\overline{V}_{k,h+1} - \underline{V}_{k,h+1}\right)(s')\right)^2 \leq O(\log T),
$$

*Here $O$ hides the dependence on other parameters such as $H, X_i^P, X_i^R$ except $T$.*

*Proof.* For notation simplicity, we use $\mathbb{E}_i$ and $\mathbb{E}_{[i:j]}$ as a shorthand of $\mathbb{E}_{s'[i]\sim\mathbb{P}_i(\cdot|(s,a)[Z_i^P])}$ and $\mathbb{E}_{s'[i:j]\sim\mathbb{P}_{[i:j]}(\cdot|s,a)}$.

$$
\begin{aligned}
& \sum_{k,h,s,a} w_{k,h}(s,a)\mathbb{E}_{[1:i]}\left[\left(\mathbb{E}_{[i+1:n]}\left(\overline{V}_{k,h+1} - \underline{V}_{k,h+1}\right)(s')\right)^2\right]\\
\leq & \sum_{k,h,s,a} w_{k,h}(s,a)\mathbb{E}_{[1:i]}\mathbb{E}_{[i+1:n]}\left(\overline{V}_{k,h+1} - \underline{V}_{k,h+1}\right)^2(s,a)\\
= & \sum_{k,h,s,a} w_{k,h}(s,a)\mathbb{E}_{[1:n]}\left(\overline{V}_{k,h+1} - \underline{V}_{k,h+1}\right)^2(s,a)\\
= & \sum_{k=1}^K \sum_{h=1}^H \sum_{s,a} w_{k,h+1}(s,a)\left(\overline{V}_{k,h+1} - \underline{V}_{k,h+1}\right)^2(s,a)
\end{aligned}
$$

Define $U_k(s,a) = 2CB_k(s,a) + \sum_{j=1}^{n} \sqrt{\frac{2H^2 L^P}{N_{k-1}((s,a)[Z_j^P])}}$. By Lemma F.17, we have

$$\sum_{k=1}^{K}\sum_{h=1}^{H}\sum_{s,a} w_{k,h+1}(s,a)\left(\overline{V}_{k,h+1} - \underline{V}_{k,h+1}\right)^2(s,a)$$

$$\leq \sum_{k,h,s,a} w_{k,h+1}(s,a)\left(\sum_{h_1=h+1}^{H}\sum_{s_{h_1},a_{h_1}} \Pr(s_{h_1},a_{h_1}|s_{h+1}=s,a_{h+1}=a)U_k(s_{h_1},a_{h_1})\right)^2$$

$$\leq \sum_{k,h,s,a} w_{k,h+1}(s,a)H\sum_{h_1=h+1}^{H}\left(\sum_{s_{h_1},a_{h_1}} \Pr(s_{h_1},a_{h_1}|s_{h+1}=s,a_{h+1}=a)U_k(s_{h_1},a_{h_1})\right)^2$$

$$\leq \sum_{k,h,s,a} w_{k,h+1}(s,a)H\sum_{h_1=h+1}^{H}\sum_{s_{h_1},a_{h_1}} \Pr(s_{h_1},a_{h_1}|s_{h+1}=s,a_{h+1}=a)\left(U_k(s_{h_1},a_{h_1})\right)^2$$

$$= \sum_{k,h} H\sum_{h_1=h+1}^{H}\sum_{s_{h_1},a_{h_1}} w_{k,h_1}(s_{h_1},a_{h_1})\left(U_k(s_{h_1},a_{h_1})\right)^2$$

$$\leq \sum_{k,h} H^2 \sum_{s_h,a_h} w_{k,h}(s_h,a_h)\left(U_k(s_h,a_h)\right)^2 \tag{63}$$

Plugging the definition of $U_k(s,a)$ into Equ. 63, we have:

$$\sum_{k=1}^{K}\sum_{h=1}^{H}\sum_{s,a} w_{k,h+1}(s,a)\left(\overline{V}_{k,h+1} - \underline{V}_{k,h+1}\right)^2(s,a) \tag{64}$$

$$\leq \sum_{k,h} H^2 \sum_{s_h,a_h} w_{k,h}(s_h,a_h)\left(2CB_k(s,a) + \sum_{j=1}^{n}\sqrt{\frac{2H^2 L^P}{N_{k-1}((s,a)[Z_j^P])}}\right)^2 \tag{65}$$

$$\leq \sum_{k,h} 2nH^2 \sum_{s_h,a_h} w_{k,h}(s_h,a_h)\left(CB_k^2(s,a) + \sum_{j=1}^{n}\frac{2H^2 L^P}{N_{k-1}((s,a)[Z_j^P])}\right) \tag{66}$$

$$\leq \sum_{k,h} 2nH^2 \sum_{s_h,a_h} w_{k,h}(s_h,a_h)CB_k^2(s,a) + 16nH^4 X_i^P L^P \log T \tag{67}$$

The last inequality is due to Lemma F.5. We can bound $\sum_{k,h} 2nH^2 \sum_{s_h,a_h} w_{k,h}(s_h,a_h)CB_k^2(s,a)$ by Lemma F.18. Summing up over all terms, we can show that $\sum_{k=1}^{K}\sum_{h=1}^{H}\sum_{s,a} w_{k,h+1}(s,a)\left(\overline{V}_{k,h+1} - \underline{V}_{k,h+1}\right)^2(s,a)$ is of order $O(1)$. $\qquad\square$

**Lemma F.20.** *Under event $\Lambda_1$ and $\Lambda_2$, for any $i \in [n]$, we have*

$$\sum_{k=1}^{K}\sum_{h=1}^{H}\sum_{(s,a)\in\mathcal{X}} w_{k,h}(s,a)u_{k,h,i}(s,a) \leq O(H^2 \sum_{j=1}^{n}\sqrt{TX_j^P|\mathcal{S}_j|L^P}),$$

*Here $O$ hides the lower order terms w.r.t. $T$.*

*Proof.* For notation simplicity, we use $\mathbb{E}_i$ and $\mathbb{E}_{[i:j]}$ as a shorthand of $\mathbb{E}_{s'[i]\sim\mathbb{P}_i(\cdot|(s,a)[Z_i^P])}$ and $\mathbb{E}_{s'[i:j]\sim\mathbb{P}_{[i:j]}(\cdot|(s,a)[Z_i^P])}$. For those expectation w.r.t the empirical transition $\hat{\mathbb{P}}_k$, we use $\hat{\mathbb{E}}_k$ to denote the corresponding expectation.

$u_{k,h,i}(s,a)$ is defined as:

$$u_{k,h,i}(s,a) = \hat{\mathbb{E}}_{[1:i]}\left[\left(\hat{\mathbb{E}}_{[i+1:n]}\left(\overline{V}_{k,h+1} - \underline{V}_{k,h+1}\right)(s')\right)^2\right].$$

$$\sum_{k,h,s,a} w_{k,h}(s,a)\hat{\mathbb{E}}_{[1:i]}\left[\left(\hat{\mathbb{E}}_{[i+1:n]}\left(\overline{V}_{k,h+1}-\underline{V}_{k,h+1}\right)(s')\right)^2\right] \tag{68}$$

$$= \sum_{k,h,s,a} w_{k,h}(s,a)\mathbb{E}_{[1:i]}\left[\left(\mathbb{E}_{[i+1:n]}\left(\overline{V}_{k,h+1}-\underline{V}_{k,h+1}\right)(s')\right)^2\right] \tag{69}$$

$$+ \sum_{k,h,s,a} w_{k,h}(s,a)\hat{\mathbb{E}}_{[1:i]}\left[\left(\hat{\mathbb{E}}_{[i+1:n]}\left(\overline{V}_{k,h+1}-\underline{V}_{k,h+1}\right)(s')\right)^2\right]$$

$$- \sum_{k,h,s,a} w_{k,h}(s,a)\mathbb{E}_{[1:i]}\left[\left(\hat{\mathbb{E}}_{[i+1:n]}\left(\overline{V}_{k,h+1}-\underline{V}_{k,h+1}\right)(s')\right)^2\right] \tag{70}$$

$$+ \sum_{k,h,s,a} w_{k,h}(s,a)\mathbb{E}_{[1:i]}\left[\left(\hat{\mathbb{E}}_{[i+1:n]}\left(\overline{V}_{k,h+1}-\underline{V}_{k,h+1}\right)(s')\right)^2\right]$$

$$- \sum_{k,h,s,a} w_{k,h}(s,a)\mathbb{E}_{[1:i]}\left[\left(\mathbb{E}_{[i+1:n]}\left(\overline{V}_{k,h+1}-\underline{V}_{k,h+1}\right)(s')\right)^2\right] \tag{71}$$

That is

We can bound Eqn. 70 and Eqn. 71 by Lemma D.1. For Eqn. 70, we have

$$\sum_{k,h,s,a} w_{k,h}(s,a)\hat{\mathbb{E}}_{[1:i]}\left[\left(\hat{\mathbb{E}}_{[i+1:n]}\left(\overline{V}_{k,h+1}-\underline{V}_{k,h+1}\right)(s')\right)^2\right]$$

$$- \sum_{k,h,s,a} w_{k,h}(s,a)\mathbb{E}_{[1:i]}\left[\left(\hat{\mathbb{E}}_{[i+1:n]}\left(\overline{V}_{k,h+1}-\underline{V}_{k,h+1}\right)(s')\right)^2\right]$$

$$\leq \sum_{k,h,s,a} w_{k,h}(s,a)\left|\hat{\mathbb{P}}_{[1:i]}(\cdot|s,a)-\mathbb{P}_{[1:i]}(\cdot|s,a)\right|_1 H^2$$

$$\leq \sum_{k,h,s,a} w_{k,h}(s,a)\sum_{j=1}^{i}\sqrt{\frac{|\mathcal{S}_j|L^P}{N_{k-1}((s,a)[Z_j^P])}}H^2$$

$$\leq 8H^2 \sum_{j=1}^{i}\sqrt{TX_j^P|\mathcal{S}_j|L^P\log T}$$

The first inequality is due to $\left(\hat{\mathbb{E}}_{[i+1:n]}\left(\overline{V}_{k,h+1}-\underline{V}_{k,h+1}\right)(s')\right)^2 \leq H^2$ for any given $s'[1:i]$. The second inequality is due to Lemma D.1. The third inequality is due to Lemma F.5.

For Eqn. 71, similarly we have

$$
\sum_{k,h,s,a} w_{k,h}(s,a)\mathbb{E}_{[1:i]}\left[\left(\left(\hat{\mathbb{E}}_{[i+1:n]}\left(\overline{V}_{k,h+1}-\underline{V}_{k,h+1}\right)(s')\right)^2\right]\right.
$$
$$
-\sum_{k,h,s,a} w_{k,h}(s,a)\mathbb{E}_{[1:i]}\left[\left(\mathbb{E}_{[i+1:n]}\left(\overline{V}_{k,h+1}-\underline{V}_{k,h+1}\right)(s')\right)^2\right]
$$
$$
\leq 2H\sum_{k,h,s,a} w_{k,h}(s,a)\mathbb{E}_{[1:i]}\left[\left(\hat{\mathbb{E}}_{[i+1:n]}\left(\overline{V}_{k,h+1}-\underline{V}_{k,h+1}\right)(s')\right)-\left(\mathbb{E}_{[i+1:n]}\left(\overline{V}_{k,h+1}-\underline{V}_{k,h+1}\right)(s')\right)\right]
$$
$$
\leq 2H\sum_{k,h,s,a} w_{k,h}(s,a)\mathbb{E}_{[1:i]}\left[H\left|\hat{\mathbb{P}}_{[i+1:n]}(\cdot|s,a)-\mathbb{P}_{[i+1:n]}(\cdot|s,a)\right|_1\right]
$$
$$
\leq 2H^2\sum_{k,h,s,a} w_{k,h}(s,a)\mathbb{E}_{[1:i]}\left[\sum_{j=i+1}^{n}\sqrt{\frac{|\mathcal{S}_j|L^P}{N_{k-1}((s,a)[Z_j^P])}}\right]
$$
$$
= 2H^2\sum_{k,h,s,a} w_{k,h}(s,a)\sum_{j=i+1}^{n}\sqrt{\frac{|\mathcal{S}_j|L^P}{N_{k-1}((s,a)[Z_j^P])}}
$$
$$
\leq 16H^2\sum_{j=i+1}^{n}\sqrt{TX_j^P|\mathcal{S}_j|L^P\log T}
$$

Eqn. 69 can be bounded by Lemma F.19, which has only logarithmic dependence on $T$. □

## G   PROOF OF THEOREM 3

*Proof.* We consider the following two hard instances.

The first instance is an extension of the hard instance in Jaksch et al. (2010). They proposed a hard instance for non-factored weakly-communicating MDP, which indicates that the lower bound in that setting is $\Omega(\sqrt{DSAT})$. When transformed to the hard instance for non-factored episodic MDP, it shows a lower bound of order $\Omega(\sqrt{HSAT})$ in episodic setting Azar et al. (2017); Jin et al. (2018). Consider a factored MDP instance with $d=m=n$ and $\mathcal{X}[Z_i^R]=\mathcal{X}[Z_i^P]=\mathcal{X}_i=\mathcal{S}_i\times\mathcal{A}_i,\quad i=1,...,n$. This factored MDP can be decomposed into $n$ independent non-factored MDPs. By simply setting these $n$ non-factored MDPs to be the construction used in Jaksch et al. (2010), the regret for each MDP is $\Omega(\sqrt{H|\mathcal{X}[Z_i^P]|T})$. The total regret is $\Omega(\sum_{i=1}^{n}\sqrt{H|\mathcal{X}[Z_i^P]|T})$. Note that in our setting, the reward $R=\frac{1}{m}\sum_{i=1}^{m}R_i$ is $[0,1]$-bounded. Therefore, we need to normalize the reward function in the hard instance by a factor of $\frac{1}{m}$. This leads to a final lower bound of order $\Omega(\frac{1}{m}\sum_{i=1}^{n}\sqrt{H|\mathcal{X}[Z_i^P]|T})=\Omega(\frac{1}{n}\sum_{i=1}^{n}\sqrt{H|\mathcal{X}[Z_i^P]|T})$. Similar construction has been used to prove the lower bound for factored weakly-communicating MDP (Xu & Tewari, 2020).

The second hard instance is an extension of the hard instance for stochastic multi-armed bandits. The lower bound of stochastic multi-armed bandits shows that the regret of a MAB problem with $k_0$ arms in $T_0$ steps is lower bounded by $\Omega(\sqrt{k_0 T_0})$. Consider a factored MDP instance with $d=m=n$ and $\mathcal{X}[Z_i^R]=\mathcal{X}[Z_i^P]=\mathcal{X}_i=\mathcal{S}_i\times\mathcal{A}_i,\quad i=1,...,n$. There are $m$ independent reward functions, each associated with an independent deterministic transition. For reward function $i$, There are $\log_2(|\mathcal{S}_i|)$ levels of states, which form a binary tree of depth $\log_2(|\mathcal{S}_i|)$. There are $2^{h-1}$ states in level $h$, and thus $|\mathcal{S}_i|-1$ states in total. Only those states in level $\log_2(|\mathcal{S}_i|)$ have non-zero rewards, the number of which is $\frac{|\mathcal{S}_i|}{2}$. After taking actions at state $s'$ in level $\log_2(|\mathcal{S}_i|)$, the agent will transits back to state $s'$ in level $\log_2(|\mathcal{S}_i|)$. That is to say, the agent can enter "reward states" at least $H-\log_2(|\mathcal{S}_i|)\geq\frac{H}{2}$ times in one episodes. For each reward function $i$, the instance can be regarded as an MAB problem $\frac{|\mathcal{S}_i\mathcal{A}_i|}{2}$ arms running for $\frac{KH}{2}$ steps[1], thus the regret for reward $i$ is

---

[1] The instance is not exactly an MAB with $\frac{|\mathcal{S}_i\mathcal{A}_i|}{2}$ arms running for $\frac{KH}{2}$ steps, since in each episode the agent will choose a state $s'$, and then stay in $s'$ and choose different actions for $\frac{H}{2}$ steps. However, this is a mild difference and we can still follow the same proof idea of the lower bound for MAB (See e.g. Theorem 14.1 in Lattimore & Szepesvári (2020))

$\Omega(\sqrt{\frac{|\mathcal{S}_i \mathcal{A}_i|}{2} \frac{KH}{2}}) = \Omega(\sqrt{|\mathcal{X}[Z_i^R]|T})$. In this construction, the total reward function can be regarded as the average of $m$ independent reward functions of $m$ stochastic MDP. This indicates that the lower bound is $\Omega\left(\frac{1}{m}\sum_{i=1}^m \sqrt{|\mathcal{X}[Z_i^R]|\,T}\right) \geq \Omega\left(\frac{1}{m}\sum_{i=1}^m \sqrt{|\mathcal{X}[Z_i^R]|\,T}\right)$.

To sum up, the regret is lower bounded by

$$\Omega\left(\max\left\{\frac{1}{m}\sum_{i=1}^m \sqrt{|\mathcal{X}[Z_i^R]|T}, \frac{1}{n}\sum_{j=1}^n \sqrt{H|\mathcal{X}[Z_j^P]|T}\right\}\right),$$

which is of the same order as

$$\Omega\left(\frac{1}{m}\sum_{i=1}^n \sqrt{|\mathcal{X}[Z_i^R]|T} + \frac{1}{n}\sum_{j=1}^n \sqrt{H|\mathcal{X}[Z_j^P]|T}\right).$$

□

# H OMITTED DETAILS IN SECTION 5

## H.1 SPECIFIC INSTANCES

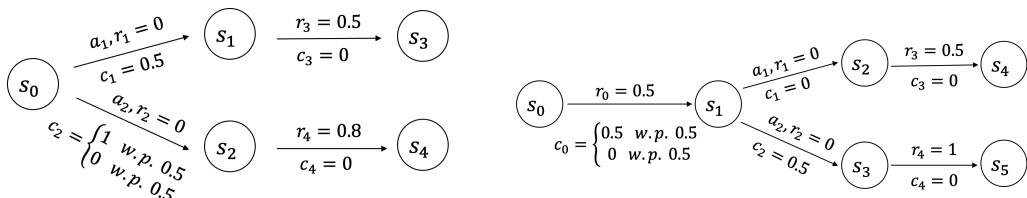

Figure 1: MDP Instances, Budget $B_0 = 0.5$

We further explain the difference with two specific examples in Fig. H.1. No matter which setting the previous work considers, the main idea of the algorithms in Efroni et al. (2020); Brantley et al. (2020) is to explore the MDP environment, and then find a near-optimal policy satisfying that the *expected* cumulative cost less than a constant vector $\boldsymbol{B_0}$, i.e. $\mathbb{E}[\sum_{h\in[H]}\boldsymbol{c}_h] \leq \boldsymbol{B_0}$. However, in our setting, the agent has to terminate the interaction once the total costs in this episode exceed budget $\boldsymbol{B}$. Because of this difference, their algorithm will converge to an sub-optimal policy with unbounded regret in our setting. In the first MDP instance (Fig. H.1), the agent starts from state $s_0$. After taking action $a_1$, it will transit to $s_1$ with a deterministic cost $c_1 = 0.5$. After taking action $a_2$, it will transit to $s_2$. The cost of taking $a_2$ is 0 with prob. 0.5, and 1 with prob 0.5. There are no rewards in state $s_0$. In state $s_1$ and $s_2$, the agent will not suffer any costs. The deterministic rewards are 0.5 and 0.8 respectively. $s_3$ and $s_4$ are termination states. The budget $B_0$ is 0.5. For this MDP instance, the optimal policy is to take action $a_1$ in $s_0$, since the agent can receive total rewards 0.5 by taking $a_1$. If taking action $a_2$, the agent will terminate at state $s_2$ with no rewards with prob. 0.5, which leads to an expected total rewards of 0.4. However, if we run the algorithm in Efroni et al. (2020); Brantley et al. (2020), the algorithm will converge to the policy that always selects action $a_2$ in $s_0$, since the expected cumulative cost of taking $a_2$ is $0.5 \leq B_0$.

We further show that the policies defined in the previous literature are not expressive enough in our setting. In the second instance, the agent starts in state $s_0$ with one action $a_0$. By taking $a_0$, the agent transits to $s_1$ with no rewards. The cost of taking $a_0$ is 0 with prob. 0.5, and 1 with prob 0.5. In $s_1$, the agent needs to decide to take $a_1$ or $a_2$, with deterministic costs of 0 and 0.5 respectively. After taking $a_1$, the agent will transits to $s_2$, in which it can obtain a reward $r_3 = 0.5$. While by taking $a_2$, the agent can transits to $s_3$, and obtain a reward $r_4 = 1$. The budget $B = 0.5$. In this instance, the action taken in $s_1$ depends on the remaining budget of the agent. That is to say, the policy is not expressive enough if it is defined as a mapping from *state* to *action*. Instead, we need to define it as a mapping from both *state* and *remaining budget* to *action*. However, previous literature only considers policies on the state space, which cannot deal with this problem.

## H.2 Algorithm and Regret

We denote $V_h^\pi(s, \boldsymbol{b})$ as the value function in state $s$ at horizon $h$ following policy $\pi$, and the agent's remaining budget is $\boldsymbol{b}$. For notation simplicity, we define $\mathbb{P}_S \mathbb{P}_C V(s, a) = \sum_{s'} \sum_{\boldsymbol{c}_0} \mathbb{P}(s'|s,a) \mathbb{P}(\boldsymbol{C}(s,a) = \boldsymbol{c}_0|s,a) V(s', \boldsymbol{b} - \boldsymbol{c}_0)$. We use $\mathbb{P}_{C,i}(c_0|s,a)$ to denote the "transition probability" of budget $i$, i.e. $\mathbb{P}(\boldsymbol{C}_i(s,a) = c_0|s,a)$. The Bellman Equation of our setting is written as:

$$V_h^\pi(s, \boldsymbol{b}) = \begin{cases} \bar{R}(s, \pi_h(s, \boldsymbol{b})) + \mathbb{P}_S \mathbb{P}_C V_{h+1}^\pi(s, \pi_h(s, \boldsymbol{b}), \boldsymbol{b}) & b > 0 \\ 0 & b \le 0 \end{cases} \tag{72}$$

Suppose $N_k(s,a)$ denotes the number of times $(s,a)$ has been encountered in the first $k$ episodes. We estimate the mean value of $r(s,a)$, the transition matrix $\mathbb{P}_S$ and $\mathbb{P}_C$ in the following way:

$$\hat{R}_k(s,a) = \frac{\sum_{k,h} \mathbb{1}[s_{k,h} = s, a_{k,h} = a] \cdot r_{k,h}}{N_{k-1}(s,a)}$$

$$\hat{\mathbb{P}}_{S,k}(s'|s,a) = \frac{N_{k-1}(s,a,s')}{N_{k-1}(s,a)}$$

$$\hat{\mathbb{P}}_{C,k,i}\left(\boldsymbol{C}_i(s,a) = c_0|s,a\right) = \frac{\sum_{k,h} \mathbb{1}[\boldsymbol{c}_{k,h,i} = c_0, s_{k,h} = s, a_{k,h} = a]}{N_{k-1}(s,a)}$$

Following the definition in the factored MDP setting, we define the confidence bonus for rewards and transition respectively (for $0 \le i \le d$):

$$CB_k^R(s,a) = \sqrt{\frac{2\hat{\sigma}_R^2(s,a)\log(2SAT)}{N_{k-1}(s,a)}} + \frac{8\log(2SAT)}{3N_{k-1}(s,a)} \tag{73}$$

$$CB_{k,i}^P(s,a,\boldsymbol{b}) = \sqrt{\frac{4\hat{\sigma}_{P,i}^2(\overline{V}_{k,h+1}, s, a, \boldsymbol{b})L}{N_{k-1}(s,a)}} + \sqrt{\frac{2u_{k,h,i}(s,a,\boldsymbol{b})L}{N_{k-1}(s,a)}} \tag{74}$$

$$+ \sqrt{\frac{32H^2 L}{N_{k-1}(s,a)}} \sum_{j=1}^n \left( \left(\frac{4nL}{N_{k-1}(s,a)}\right)^{\frac{1}{4}} + \sqrt{\frac{4nL}{3N_{k-1}(s,a)}} \right) \tag{75}$$

$$+ \sum_{j=1}^n H \left( \sqrt{\frac{4|\mathcal{S}_i|L^P}{N_{k-1}(s,a)}} + \frac{4|\mathcal{S}_i|L}{3N_{k-1}(s,a)} \right) \left( \sqrt{\frac{4|\mathcal{S}_j|L^P}{N_{k-1}(s,a)}} + \frac{4|\mathcal{S}_j|L}{3N_{k-1}(s,a)} \right), \tag{76}$$

where $L = \log(2dSAT) + d\log(mB)$ is the logarithmic factors because of union bounds. The additional $d\log(mB)$ is because that we need to take union bounds over all possible budget $\boldsymbol{b}$. This difference compared with factored MDP is mainly due to the noised offset model.

$CB_k^R(s,a)$ is the confidence bonus for rewards, and $\hat{\sigma}_R(s,a)$ denotes the empirical variance of reward $R(s,a)$, which is defined as:

$$\hat{\sigma}_R(s,a) = \frac{1}{N_{k-1}(s,a)} \sum_{k=1}^{k-1} \sum_{h=1}^H \mathbb{1}\left[(s,a)_{k,h} = (s,a)\right] \cdot (r_{k,h}(s_{k,h}, a_{k,h}))^2 - \left(\hat{R}_k(s,a)\right)^2$$

$CB_{k,0}^P(s,a)$ is the confidence bonus for state transition estimation $\hat{\mathbb{P}}_S$, and $\{CB_{k,i}^P(s,a)\}_{i=1,\dots,d}$ is the confidence bonus for budget transition estimation $\{\hat{\mathbb{P}}_{C,i}\}_{i=1,\dots,d}$. $\hat{\sigma}_{P,i}^2(\overline{V}_{k,h+1}, s, a)$ is the empirical variance of corresponding transition:

$$\hat{\sigma}_0^2(\overline{V}_{k,h+1}, s, a, \boldsymbol{b}) = \text{Var}_{s' \sim \hat{\mathbb{P}}_{S,k}(\cdot|(s,a)[Z_i^P])}\left( \mathbb{E}_{\boldsymbol{c} \sim \hat{\mathbb{P}}_{C,k}(\cdot|s,a)} \overline{V}_{k,h+1}(s', \boldsymbol{b} - \boldsymbol{c}) \right)$$

$$\hat{\sigma}_{P,i}^2(\overline{V}_{k,h+1}, s, a, \boldsymbol{b})$$

$$= \mathbb{E}_{s' \sim \hat{\mathbb{P}}_{S,k}(\cdot|s,a)} \mathbb{E}_{\boldsymbol{c}_{[1:i-1]} \sim \hat{\mathbb{P}}_{C,k,[1:i-1]}(\cdot|s,a)} \left[ \text{Var}_{\boldsymbol{c}_i \sim \hat{\mathbb{P}}_{C,k,i}(\cdot|s,a)} \left( \mathbb{E}_{\boldsymbol{c}_{[i+1:n]} \sim \hat{\mathbb{P}}_{C,k,[i+1:d]}(\cdot|s,a)} \overline{V}_{k,h+1}(s', \boldsymbol{b} - \boldsymbol{c}) \right) \right],$$

where $\hat{\mathbb{P}}_{C,k,[d_1:d_2]} = \prod_{i=d_1}^{d_2} \hat{\mathbb{P}}_{C,k,i}$.

$\sqrt{\frac{2u_{k,h,i}(s,a,\boldsymbol{b})}{N_{k-1}(s,a)}}$ is added to compensate the error due to the difference between $V_{h+1}^*$ and $\overline{V}_{k,h+1}$, where $u_{k,h,i}(s,a)$ is defined as:

$$u_{k,h,0}(s,a,\boldsymbol{b}) = \mathbb{E}_{s'\sim\hat{\mathbb{P}}_{S,k}(\cdot|s,a)}\left[\left(\mathbb{E}_{\boldsymbol{c}\sim\hat{\mathbb{P}}_{C,k}(\cdot|s,a)}\left(\overline{V}_{k,h+1} - \underline{V}_{k,h+1}\right)(s',\boldsymbol{b}-\boldsymbol{c})\right)^2\right]$$

$$u_{k,h,i}(s,a,\boldsymbol{b}) = \mathbb{E}_{s'\sim\hat{\mathbb{P}}_{S,k}(\cdot|s,a)}\mathbb{E}_{\boldsymbol{c}_{[1:i]}\sim\hat{\mathbb{P}}_{C,k,[1:i]}(\cdot|s,a)}\left[\left(\mathbb{E}_{\boldsymbol{c}_{[i+1:n]}\sim\hat{\mathbb{P}}_{C,k,[i+1:d]}(\cdot|s,a)}\left(\overline{V}_{k,h+1} - \underline{V}_{k,h+1}\right)(s',\boldsymbol{b}-\boldsymbol{c})\right)^2\right].$$

We calculate the optimistic value function and find the optimal policy $\pi$ via the following value iteration in our algorithm:

$$\overline{V}_h(s,\boldsymbol{b}) = \begin{cases} \max_a\left\{\left[\hat{R}(s,a) + CB(s,a) + \hat{\mathbb{P}}_s\hat{\mathbb{P}}_c\overline{V}_{h+1}(s,a,\boldsymbol{b})\right]\right\} & b > 0 \\ 0 & b \le 0 \end{cases} \tag{77}$$

---

**Algorithm 4** FMDP-BF for RLwK

---

**Input**: $\delta$
Initialize $N(s,a) = 0$ for any $(s,a) \in \mathcal{X}$
**for** episode $k = 1, 2, \cdots$ **do**
    Set $\overline{V}_{k,H+1}(s,\boldsymbol{b}) = \underline{V}_{k,H+1}(s,\boldsymbol{b}) = 0$ for all $s,a,\boldsymbol{b}$.
5:    Let $\mathcal{K} = \{(s,a) \in \mathcal{S} \times \mathcal{A} : N_k(s,a) > 0\}$
    **for** horizon $h = H, H-1, ..., 1$ **do**
        **for** $s \in \mathcal{S}$ and all possible budget $\boldsymbol{b}$ from $\boldsymbol{0}$ to $\boldsymbol{B}$ **do**
            **for** $a \in \mathcal{A}$ **do**
                **if** $(s,a) \in \mathcal{K}$ **then**
10:                  $\overline{Q}_{k,h}(s,a,\boldsymbol{b}) = \min\{H, \hat{R}_k(s,a) + CB_k(s,a) + \hat{\mathbb{P}}_{S,k}\hat{\mathbb{P}}_{C,k}\overline{V}_{k,h+1}(s,a,\boldsymbol{b})\}$
                **else**
                  $\overline{Q}_{k,h}(s,a,\boldsymbol{b}) = H$
                **end if**
            **end for**
15:            $\pi_{k,h}(s,\boldsymbol{b}) = \arg\max_a \overline{Q}_{k,h}(s,a,\boldsymbol{b})$
            $\overline{V}_{k,h}(s,\boldsymbol{b}) = \max_{a\in\mathcal{A}} Q_{k,h}(s,a,\boldsymbol{b})$
            $\underline{V}_{k,h}(s,\boldsymbol{b}) = \max\left\{0, \hat{R}_k(s,\pi_{k,h}) - CB_k(s,\pi_{k,h},,\boldsymbol{b}) + \hat{\mathbb{P}}_k\underline{V}_{k,h+1}(s,\pi_{k,h},\boldsymbol{b})\right\}$
        **end for**
    **end for**
20:    **for** step $h = 1, \cdots, H$ **do**
        Take action $a_{k,h} = \arg\max_a \hat{Q}_{k,h}(s_{k,h}, a)$
    **end for**
    Update history trajectory $\mathcal{L} = \mathcal{L}\bigcup\{s_i,a_i,r_i,s_{i+1}\}_{i=1,2,...,t_k}$
**end for**

---

*Proof.* (Theorem 4) The proof follows almost the same proof framework of Thm. 2. The term $\log(SAT) + d\log(Bm)$ is due to a union bound over all possible $(T,s,a)$ and budget $\boldsymbol{b}$. This difference is because of the additional union bounds over all budget $\boldsymbol{b}$. $\qquad\square$

### H.3 DISCUSSIONS ABOUT ASSUMPTION 1 AND ASSUMPTION 2

Assumptions 1 and 2 limit our Algorithm 4 to problems with discrete costs. One may wonder whether it is possible to construct $\epsilon$-net for budget $B$ and possible value of the costs when these assumptions don't hold. In that case, we only need to estimate the discrete cost distributions on the $\epsilon$-net, and then apply Algorithm 4 to tackle the problem. Unfortunately, we find that the $\epsilon$-net construction doesn't work for continuous cost distributions, and it is unlikely to achieve efficient

regret guarantee without further assumptions and the modification of the basic setting. If the cost distributions are continuous and the remaining budget can take any value in $\mathbb{R}$, a small perturbation on the remaining budget may totally change the policy in the following steps and the optimal value. To be more specific, suppose the agent enters a certain state with remaining budget $b$. There are three actions to choose, with the cost of $b - \epsilon$, $b$ and $b + \epsilon$, respectively. After suffering the cost, the agent can achieve reward of $0, 0.5$ and $1$ respectively. Note that we can construct such hard instances with $\epsilon$ extremely small. For these hard instances, we need to carefully estimate the value function of any remaining budget $b \in \mathbb{R}$ and the density functions of the costs, after which we can calculate the value through Bellman backup and find the optimal policy. However, estimating the density functions requires infinite number of samples and makes the problem intractable. In other words, the "non-smoothness" of the value and the policy w.r.t the remaining budget makes the problem difficult for continuous value distribution without further assumptions. This "non-smoothness" phenomenon also happens in the classical knapsack problem.

There are two possible ways to remove these assumptions and apply our algorithm to continuous cost distributions with $\epsilon$-net technique. The first idea is to allow the slight violation of the total budget constraints, with the maximum violation threshold $\delta$, or we assume that the value of any initial state is lipschiz w.r.t the total budget $B$ in a small neighborhood of $B$ (With maximum $L_\infty$ distance $\delta$). In that case, we can tolerate the estimation error of each cost function to be at most $\frac{\delta}{H}$. $\epsilon$-net technique with $\epsilon = \frac{\delta}{H}$ still works in this case and we can estimate the cost distribution with precision $\frac{\delta}{H}$. This modification is somewhat reasonable since the agent's policies are always "smooth" w.r.t the total budget in a small region near $B$ in many real applications such as games and robotics. The second idea is to consider soft constraints. That is, when the budget constraints are violated, the agent will suffer a loss that is linear w.r.t the violation of the constraints. We assume the linear coefficient is relatively large compared with other parameters. This is also a possible method to remove the non-smoothness w.r.t the total budget, which has wide applications in constrained optimization.

