# OpenReview forum: "Efficient Reinforcement Learning in Factored MDPs with Application to Constrained RL"
_ICLR.cc/2021/Conference — ICLR 2021 Poster_

### Official Review · AnonReviewer1 · 2020-10-28
**Official Blind Review #3**

**Rating:** 7
**Confidence:** 4

**Review:**

This paper focuses on episodic, factored Markov decision processes and proposes the FMDP-BF algorithm, which is computation efficient and improves the previous result of the FMDP algorithm. The author also provides a theoretical lower bound for the FMDP-BF algorithm and shows that the FMDP-BF algorithm's regret is near-optimal. Besides, this paper applies the FMDP-BF algorithm in RLwK and provides theoretic analyses of regret. However, I still have some suggestions about this paper.
Firstly, the definition of factor MDPs is complex and challenging to understand. It is better to add some examples for the factor MDP setting.
Secondly, there is no experimental support for the FMDP algorithm, and it is better to perform some experiments with FMDP-BF, FMDP-CH algorithm.
Finally, I have a concern about the FMDP algorithm. The UCBVI-CH algorithm, UCBVI-BF algorithm, and FMDP-CH algorithm only maintain one value function V in each episode. The only difference is the form of a bonus term. However, the FMDP-BF algorithm needs to keep two value function V to compute the bonus term, and I wondered whether it is necessary to keep two value functions in each episode.

---

> ### Author Response · Authors · 2020-11-19
> **Response to Reviewer 1**
>
> We thank Reviewer 1 for the insightful review and positive comments.
>
>  Regarding FMDP examples: An excellent example is given by Osband \& Van Roy (2014), about a large production line with $m$ machines in sequence with $K$ possible states each. Over a single time-step each machine can only be influenced by its direct neighbors. Another example  is in robotics: the transition dynamics of a robot's arms may be reasonably assumed to be independent of the transition dynamics of its legs. We have updated the paper and added these examples to better explain the concept of factored MDP (Paragraph 2 in page 3).
>
> Regarding the experiments: Thanks for your suggestions. Our main contribution is the provably efficient algorithms with regret guarantee. To the best of our knowledge, there is no well-accepted benchmark for factored MDPs. We would be interested in suggestions, and are open to testing our algorithms empirically on good benchmark problems.
>
> Regarding the use of two value functions in FMDP-BF algorithm:  Directly applying UCBVI-BF to our setting and only maintaining an optimistic value does not work due to technical issues. Roughly speaking, the problem comes from the additional bonus term $u_{k,h,i}(s,a)$, which is used to compensate for the error of variance estimation due to the difference between $V_{h+1}^*$ and $\bar{V_{k,h+1}}$. For non-factored MDP, Azar et al. (2017) construct this additional term with a clever induction method over all steps. This leaves a term of form $\frac{1}{N_{k,h+1}^{\prime}(y)}$ in the additional bonus term (the last term of Eqn. 4 in Azar et al. (2017)), where $N_{k,h+1}^{\prime}(y)$ is the counter of state $y$ in step $h+1$ before episode $k$. When calculating the regret, summing over all possible $y \in \mathcal{S}$ will lead to a linear dependence on $S$ (the term (h) on page 26 of Azar et al. (2017)) in the $O(T^{\frac{1}{4}})$ term. This linear dependence on $S$ in this lower order term w.r.t $T$ is okay for non-factored setting but **unacceptable for factored MDP** --- in this work we are instead aiming for regret bounds that avoid such $O(S)$ dependence by taking advantage of the factorization structure of FMDPs.  In other words, the construction technique of additional bonus term $u_{k,h,i}(s,a)$ in Azar et al. (2017) is too loose for our purpose, so we maintain both the optimistic and the pessimistic value estimation and calculate $u_{k,h,i}(s,a)$ in a more delicate way.

---

### Official Review · AnonReviewer2 · 2020-10-30
**Concerns about technical novelty over Osband & Van Roy (2014), Azar et al. 2017**

**Rating:** 6
**Confidence:** 3

**Review:**

The authors study the factor MDP problem in an online and episodic setting. They provide two main contributions on this question. First, they propose an OFU type algorithm which enjoys a better regret bound than Osband & Van Roy (2014) by a factor of  $\sqrt{nH \Gamma}$. The improvement is brought about by a refined consideration on the confidence radius' dependence on the variances of the rewards, which carries a similar idea to the design of UCBVI-CH by Azar et al. 2017 for the tabular case.  The second contribution is on the generalization to an episodic FMDP with knapsacks problem, where the authors generalize the approach in the first contribution to provide a regret bound.

My overall concern about the paper is about the technical novelty of the paper. While I understand that there are quite a lot of technical calculations involved, the improved regret bound compared to Osband & Van Roy (2014) appears to be a direct outcome of replacing the confidence bounds used in Osband & Van Roy (2014)  by the confidence bounds proposed in Azar et al. 2017 on each of the factors in the factored model. In other words, on top of the technical contributions from Osband & Van Roy (2014)  and Azar et al. 2017, the proposed paper's first technical contributions appear to be on how to merge those two papers together.

### Post Rebuttal ### Thanks for the clarification, I have adjusted my score accordingly

While the paper's second contributions allow the incorporation of a more refined resource constraints, it appears to me that the the proposed approach is unlikely to yield a computationally tractable algorithm even in the tabular setting, when the original state space is small. Indeed, the vector of remaining resource levels are embedded into the state, so that it will results in a curse in dimensionality even when the original space is of a manageable size. Overall, based on my evaluation on the theoretical and practical impact of the paper, I find that the contributions fall marginally below the acceptance threshold.  Finally, after the statement of the regret bound in Theorem 2, it is helpful to compare with the regret bound in the recent work by Tian et al. 2020, similarly to how the proposed algorithm is compared to Osband & Van Roy (2014).

---

> ### Author Response · Authors · 2020-11-19
> **Response to Reviewer 2**
>
> We thank Reviewer 2 for the insightful review and detailed comments.
>
>  Regarding technical novelty and contributions:
>    Directly merging the algorithms of Osband \& Van Roy (2014) and Azar et al. (2017), as suggested by the reviewer, does *not* result in our strong regret bounds. Roughly speaking, the problem comes from the additional bonus term $u_{k,h,i}(s,a)$, which is used to compensate for the error of variance estimation due to the difference between $V_{h+1}^*$ and $\bar{V_{k,h+1}}$.  The construction technique of the additional bonus term (the last term of Eqn. 4 in Azar et al. (2017)) is loose for factored MDPs. Directly modifying it to our setting will lead to a polynomial dependence on the cardinality of the whole state space $S$. To tackle this problem, we maintain both an optimistic and a pessimistic value estimates in our algorithm, and calculate the additional bonus term $u_{k,h,i}(s,a)$ in a sophisticated way. Moreover, we propose a decomposition formula for factored Markov chains (Theorem 1). It shows how to define the empirical variance in the confidence bonus, and helps further improve the regret bound by a factor of $\sqrt{nH}$ compared with FMDP-CH.  We also believe this theorem is of independent interest with potential use in other problems in factored MDPs.
>
>  Regarding the computational complexity of FMDP-BF for RLwK: Although the algorithm FMDP-BF for RLwK is statistically efficient, the computational complexity may be exponential in the dimension $d$, as the reviewer pointed out. This is the consequence of NP-hardness of knapsack problems with multiple constraints; please see the discussion at the end of Section 5. Luckily, in many real applications, the number of different kinds of constraints $d$ may be relatively small. For example, in robotics, the budget we are concerned about may only be the remaining energy of the robot (so $d=1$). In games, the player mainly cares about the constraints about health or the remaining time (so $d=2$). From a theoretical perspective, finding scenarios for which we can design both statistically and computationally efficient algorithms is an interesting future direction.
>
>  Regarding the comparison with Tian et al. (2020) : Thanks for the suggestion. Compared with their results, we further improve the regret by a factor of $\sqrt{n}$ with a more refined variance decomposition theorem (Theorem 1). We have added more discussion after our Theorem 2 in the paper (Page 7).

---

> > ### Comment · AnonReviewer2 · 2020-11-24
> > **Thanks for the clarification**
> >
> > After reading the paper again, I agree that Theorem 1 is a novel contributions on top of the two existing results highlighted, and this clarifies my query on the technical novelty. The score is adjusted accordingly.

---

### Official Review · AnonReviewer4 · 2020-11-02
**Tight regret bounds for episodic Factored MDPs**

**Rating:** 7
**Confidence:** 4

**Review:**

This paper proposes a reinforcement learning algorithm FMDP-BF for episodic Factored MDPs. Similar to previous works, FMDP-BF follows the principle of "optimism in the face of uncertainty" to efficiently explore to achieve low regret. Compared to algorithms for general MDPs, FMDP-BF leverages the factorization structure of FMDP and results in exponential regret reduction in the size of the sate space. Compared to previous results for FMDP, the proposed algorithm utilizes Bernstein-type confidence bounds and new bounds on the factored transitions to reduce the regret order. The paper also connects a class of constrained RL to FMDP and  provide a sample efficient for the constrained RL problem.

The proposed method improves the regret order for FMDP by using novel bounds for the estimation error of one-step values (Lemma 4.1) and their variances (Corollary 1.1). A regret lower bound is also provided which matches the upper bound except for a $\sqrt{n}$ and logarithmic factors. The paper is generally well written, and the proofs in the appendix are basically correct except some minor issues listed below.

Some suggestions:

- Corollary 1.1, likely typo: $V^{\pi}_{h}$ instead of $V^{\pi}_{h+1}$.
- Algorithm 1, line 13, likely typo: $\hat{R}_k(s, \pi_{k,h})$.
- Bottom of page 11, definition of $\phi_{k,i}$, likely typo in $N_{k-1}(s,a)$ .
- Through out the proofs, many terms could be replaced by proper notations like $\phi_{k,i}$ or $CB^R_{k, Z^R_i}(s, a)$. For example, the RHS of (9) is exactly $\phi_{k,i}$. By properly using the notations, the proofs would be much easier to read and the connection between lemmas would be much clearer.
- $\eta_{k, h, i}(s, a)$ is mentioned in Algorithm 1, but the notation is never explicitly defined. Also, it's a bit confusion to have two different sets of definitions for $CB^R, CB^P$ in (1)-(2) and in (3)-(6) for different algorithms. Maybe to use different notations and explicitly define the extra term $\eta$.
- Lemma D.1, missing $[Z^R_i]$ in (7). (9) seems like a direct application of (10).
- Lemma D.2, the failure probability of $F^N$ in Dann et al. is bounded by $SAH\delta$, which seems to suggests the log term in (15) is not correct.
- In the proof of Lemma D.2, $V$ should be $V_{h+1}$, and the sentence after the set of equations should be "The last inequality..."
- In (19), $\mathbb{P}_n$ instead of $\mathbb{P}_{k, n}$, and there seems to require an additional inequality for the last term to continue the recursion.
- For Lemma E.2, missing $|S_i|$ in (21), and missing $1/m$ for the first equation in the proof, and $\hat{\mathbb{P}}_{i, k}$ in after the first inequality of the second equation.
- Typo $\mathbb{P}_h$ in the proof of Lemma E.3.
- In the proof of Theorem 5, typo $\hat{\mathbb{P}}_{k, h}$ in the definition of $\delta^3_{k, h}$, missing $[Z^P_i]$ in the third paragraph of page 19, and extra a's in the value functions in the last equation of page 19. In page 20, $\delta^1_{k,h}$ and $\delta^3_{k,h}$ should be bounded by Lemma E.2. At the end of the proof, the doesn't the big-O notation also hides all lower order terms in $T$?
- In the proof of Lemma F.2, the constant in the log term is not consistent with Lemma D.2.
- In the proof of Lemma F.4, the first equality incorrectly swaps the two summation. The result seems to be correct by first applying the inequality.
- In the proof of Lemma F.5, the second inequality in the first equation seems to miss the last term which might be bounded by $log(T)$.
- In the proof of Lemma F.10, the two inequalities bounded by $CB^R$ and $CB^P$ terms may be better stated as a lemma.
- In the proof of Theorem 2, extra actions in the value functions in (44).

---

> ### Author Response · Authors · 2020-11-19
> **Response to Reviewer 4**
>
> We thank Reviewer 4 for the thorough review and positive comments. We have revised the paper accordingly.

---

### Official Review · AnonReviewer3 · 2020-11-04
**Review of Paper 718**

**Rating:** 7
**Confidence:** 4

**Review:**

This paper studies RL in episodic factored MDPs (FMDPs) in the regret setting. The paper introduces an algorithm called FMDP-BF, which is a model-based algorithm implementing the optimistic principle by maintaining upper and lower confidence bounds derived using empirical Bernstein-type confidence sets.
The paper presents a high-probability regret bound for FMDP-BF appearing to be superior than existing results for episodic FMDPs. It also reports a regret lower bound for FMDPs.

The paper further studies a variant of "RL under constraints" in episodic MDPs and the regret setting, and shows that the corresponding problem can be cast as an instance of RL in an associated FMDP. Applying FMDP-BF to this (originally non-factored) RL problem, the paper derives tighter regret bounds than what would be obtained otherwise.


Main Comments:

The paper studies an interesting and timely RL problem, which I believe to be significant in practice. In particular, the studied application to constrained RL is interesting and relevant. While the presented algorithm builds on well-known constructions that are already present in a series of recent works on episodic tabular MDPs, deriving sharp regret bounds like those in the present paper poses additional challenges making the problem non-trivial. To address such challenges, the paper introduces a couple of technical lemmas that sound interesting beyond the scope of this paper. These include in particular, Lemma 4.1 and Theorem 1 stating a Bellman-type result for the variance of factored Markov chains.

The paper is well-organized and states the results and the setup clearly. While the general writing quality is acceptable, I believe it still requires some polishing. In particular, there are many typos (see a list of them below), and in some cases there are unclear statements.
The review of the relevant literature seems adequate. In particular, the authors responsibly mention the parallel work of (Tian et al., 2020), though I urge the authors to provide a deeper technical comparison of their results to those of the latter paper.

Besides the writing quality that could be improved by clarifying some statements and fixing typos, I have the following comments, which I would like to be addressed in the rebuttal:

- While (Tian et al., 2020) is cited in the paper, the authors fail to adequately compare their results to those in (Tian et al., 2020)  -- except for a short discussion in page 3. Specifically, I believe they need to compare Theorem 2 to the regret bounds of (Tian et al., 2020) on page 6. I believe this is the right place for making such a comparison. Currently, there is only comparison against (Osband and Van Roy, 2014). The paper also lacks a comparison between the lower bound in Theorem 3 and those in (Tian et al., 2020). A precise and specific comparison is necessary here.

- The idea of reducing the constrained RL problem to an FMDP is very nice. However, I think Assumption 2 makes the application of such a reduction rather restrictive. Could you please explain whether this can be improved or not?

Minor Comments:

- In Section 2.1, as far as I know (Lu & Van Roy, 2019) is not relevant to factored MDPs.

- In page 7, the authors stated that introducing an $\epsilon$-net to relax Assumption 2 "will not influence the regret". This is not true. This will penalize the regret, but perhaps with the choice of $\epsilon=1/T$, the penalty does not influence the \emph{order} of regret bounds ignoring polylogarithmic factors. Please explain.

===== Post Rebuttal =====

Thanks for your response and for revising the paper.  I have increased my score accordingly.


Some Typos/Mistakes:

p. 1: In paragraph 3, introduce $T$.

p. 1: Introduce $\mathcal X[Z_i]$.

p. 2: there exists functions -> exist

p. 3 (and elsewhere): for factored Markov chain -> … chains

p. 3: Knapsack setting -> the knapsack setting

p. 3: … is more concise -> are

p. 3: We use … estimation of $V^\star_h$ -> …  estimation of $V^\star_h$, respectively.

p. 4: do not harm to the … -> remove "to"

p. 4 (and elsewhere): omits higher order factors -> I believe you mean here that $\eta$ collects some higher order terms in the expression.

p. 5: all possible value of -> values

p. 6: $X[Z_i]$ -> Did you mean $\mathcal X[Z_i]$

p. 7: for a $\sqrt{n}$ -> for a factor of $\sqrt{n}$

p. 7: focus -> focuses

p. 7: … but counts the cumulative cost … -> "counts" has made the statement rather unclear.

p. 8: from state to … -> from states and budget to actions (or state-space … to action-space)

p. 8: With prob. at least -> shortening probability to "prob." does not seem to provide any gain in the space.

p. 8: To be more formally -> formal

p. 8: in interesting future work -> an interesting …

---

> ### Author Response · Authors · 2020-11-19
> **Response to Reviewer 3**
>
> We thank Reviewer 3 for the insightful review and detailed comments.
>
>
> Regarding the assumptions for RLwK: Thanks for the valuable comments. We believe that it is unlikely to achieve efficient regret guarantee for continuous cost distributions without further assumptions. For our RLwK setting, we assume that the budget as well as the possible value of costs is an integral multiple of the unit cost $\frac{1}{m}$ in assumption 1, and assumption 2 states that the cost can only take at most $n$ possible values. A useful observation is that assumption 2 holds immediately with $n = mC$ if assumption 1 holds and the cost is bounded by a constant $C$. If the cost distributions are continuous and the remaining budget can take any value in $\mathbb{R}$, a small perturbation on the remaining budget may totally change the policy in the following steps and the optimal value.  To be more specific, suppose the agent enters a certain state with remaining budget $b$. There are three actions to choose, with the cost of $b-\epsilon$, $b$ and $b+\epsilon$, respectively. After suffering the cost, the agent can achieve reward of $0$, $0.5$ and $1$ respectively. Note that we can construct such hard instances with $\epsilon$ extremely small. For these hard instances, we need to carefully estimate  the value function of any remaining budget $b \in \mathbb{R}$ and the density functions of the costs, after which we can calculate the value through Bellman backup and find the optimal policy. However, estimating the density functions requires infinite number of samples and makes the problem intractable. In other words, the "non-smoothness" of the value and the policy w.r.t the remaining budget makes the problem difficult for continuous value distribution without further assumptions. This "non-smoothness"  phenomenon also happens in the classical knapsack problem. Based on these observations, we agree that a more careful construction is needed for the $\epsilon$-net argument, and thank the reviewer for suggesting it.
>
> There are two possible ways to remove these assumptions and apply our algorithm to continuous cost distributions. The first idea is to allow the slight violation of the total budget constraints, with the maximum violation threshold $\delta$, or we assume that the value of any initial state is lipschiz w.r.t the total budget $\boldsymbol{B}$ in a small neighborhood of $\boldsymbol{B}$ (With maximum $L_{\infty}$ distance $\delta$). In that case, we can tolerate the estimation error of each cost function to be at most $\frac{\delta}{H}$. $\epsilon$-net technique with $\epsilon = \frac{\delta}{H}$ still works and we can estimate the cost distribution with precision $\frac{\delta}{H}$. This modification is somewhat reasonable since the agent's policies are always "smooth" w.r.t the total budget in a small region near $\boldsymbol{B}$  in many real applications such as games and robotics. The second idea is to consider soft constraints. That is, when the budget constraints are violated, the agent will suffer a loss that is linear w.r.t the violation of the constraints. We assume the linear coefficient is relatively large compared with other parameters. This is also a possible method to remove the non-smoothness w.r.t the total budget, which has wide applications in constrained optimization. We have added these discussions in Appendix H.3.
>
>
> Regarding the comparison with Tian et al. (2020): Thanks for the suggestion. Compared with their results, we further improve the regret by a factor of $\sqrt{n}$ in the upper bound with a more refined variance decomposition theorem (Theorem 1). For the lower bounds, our lower bound is of the same order with theirs with different hard instance construction, and our lower bound measures the dependence on all the parameters including the number of the factored transition and the factored rewards. We have added the discussion after Theorem 2 and Theorem 3 in the paper (Page 7).
>
>
> Regarding the paper of Lu \& Van Roy (2019): The authors studied factored MDPs in section 5.3 as an application of their information-theoretical confidence bounds, though this is not the main contribution.  We will make the connection more explicit in the final version.
>
>
> Regarding the typos: Thanks for the comments. We have revised the paper accordingly.

---

### Decision · Program_Chairs · 2021-01-07
**Final Decision**

**Decision:**

Accept (Poster)

**Comment:**

This paper establishes the currently sharpest regret bounds for reinforcement learning in episodic factored MDP. The result improve the result by Osband and Van Roy 2014. The proposed FMDP-BF is a model-based algorithm that construct confidence sets of the transition distributions using Bernstein  and adapt policies by optimistic planning. The regret bounds holds with high probability. They also provide a lower bound for this class of problems. Reviewers all see merit in the theoretical results of the paper and reach a consensus that this is a good paper. We'd still like to request that the authors make all corrections and clarifications following the reviewers's suggestions, especially to improve the clarity of the formulation and proof sketches.

A separate suggestion: Model-based RL is a long existing approaches. For MDP belongs to a specific family, there exist regret bounds that depend on Eluder dimension of the the MDP family, see eg. https://arxiv.org/abs/1406.1853 and https://arxiv.org/abs/2006.01107. Can these results be applied to the factored MDP family and yield similar regret bounds? It would be necessary to add discussions about these papers, and explain why or why not these general regret bounds can apply to analyze HMDP.